# LAMDA: Two-Phase Multi-Fidelity HPO via Learning Promising Regions from Data

## Abstract

Multi-fidelity hyperparameter optimization (HPO) combines data from both high-fidelity (HF) and low-fidelity (LF) problems during the optimization process, aiding in effective sampling and preliminary screening. To enhance its performance, approaches that incorporate expert knowledge or transfer ability into the HPO algorithm have demonstrated their superiority, while such domain knowledge or abundant data from multiple similar tasks may not always be accessible. Observing that high-quality solutions in HPO exhibit some overlap between high- and low-fidelity problems, we propose a two-phase framework `Lamda` to streamline the multi-fidelity HPO. Specifically, in the first phase, it searches in the LF landscape to identify the promising regions of LF problem. In the second phase, we leverage such promising regions to construct reliable priors to navigate the HPO. We showcase how the `Lamda` framework can be integrated with various HPO algorithms to boost their performance, and further conduct theoretical analysis towards the integrated Bayesian optimization and bandit-based Hyperband. We demonstrate the effectiveness of our framework across 56 HPO tasks.

## 1. Introduction

The performance of machine learning models is highly dependent on their hyperparameters (Bischl et al., 2023), while hyperparameter optimization (HPO) has become a popular research area in both academia and industry (Li et al., 2022a). In practice, the cost of an HPO task can be prohibitively high when dealing with large models or datasets. For instance, the training time of a specified model on large datasets can take several hours or even days (Krizhevsky et al., 2012). Various HPO methods have been developed, ranging from the well-established random search (RS) (Bergstra & Bengio, 2012) to more data-efficient Bayesian optimization (BO) (Kandasamy et al., 2018; Bergstra et al., 2011; McLeod et al., 2017). Many of these methods find solutions from a uniform global perspective as shown in Figure 1(a). To avoid directionless search with potentially low returns, variants based on localized search strategies such as the trust region Bayesian optimization (TuRBO) (Eriksson et al., 2019) have been proposed with more focused search regions illustrated in Figure 1(b). Nevertheless, all of these methods do not scale satisfactorily with the increasingly complex and costly HPO tasks. In this context, especially with deep models and large-scale datasets, fidelity management becomes more important given limited budget.

Based on the hypothesis that low-fidelity (LF) evaluation reveals a reasonable approximation of the high-fidelity (HF) performance while consuming less budget, multi-fidelity HPO methods employ various techniques to actively manipulate the evaluation fidelity, such as using subsets of dataset, reducing feature space, and decreasing the number of training epochs (Klein et al., 2017; Falkner et al., 2018). The multi-fidelity Bayesian optimization (MFBO) (Swersky et al., 2013) and bandit-based methods (Li et al., 2017) are two representative multi-fidelity HPO methods. For MFBO,

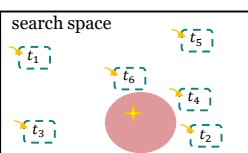 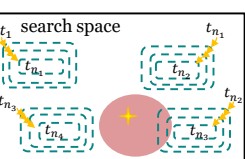 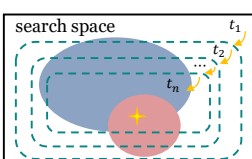 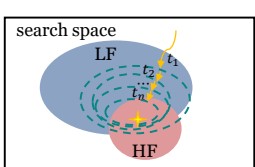

(a) Global search methods    (b) Local search methods: TuRBO    (c) Bandit based methods    (d) Methods using promising area

Figure 1: A conceptual demonstration of how different HPO methods explore the search space. The red and blue areas represents regions of high-quality solutions for an HF and LF problem, respectively, while the yellow stars denote the optimal solution for the HPO task. The dashed lines in panel (a) show the locations of sampled points at each iteration and represent the search space of the sampling function depicted in panels (b) through (d). $t$ represents the iteration number.

existing work primarily constructs an integrated surrogate model accommodating multi-fidelity evaluations for better acquiring candidate configurations (Poloczek et al., 2017; Kandasamy et al., 2019; Mikkola et al., 2022; Li et al., 2020b). Bandit-based methods, on the other hand, utilize data from LF problems to filter potentially good configurations for HF problems (Falkner et al., 2018; Li et al., 2022b; Awad et al., 2021). As both MFBO and bandit-based methods follow within the Bayesian framework, previous work unintentionally downplayed the role of priors. The random sampling strategy in bandit algorithms and uniform acquisition horizon of MFBO were consistently adopted for all HPO tasks, leading to uninformative priors leaving limited space to prevent performance degradation or further improve efficiency. Their trajectory routine is presented in Figure 1(c).

With the growth of data analysis techniques and the accumulation of more and more in-depth experience in HPO tasks, increasing effort has been put into the heuristics for better guidance of a single HPO task, functionally equivalent to proactively replacing the uninformative priors. The strategic search of recent HPO research has been proposed, either relying on the domain expert knowledge towards the incumbent HPO task, such as Priorband and BO with crafted prior (Souza et al., 2021; Mallik et al., 2023), or requiring the transfer similarity from multiple HPO tasks (Watanabe et al., 2023). As shown in Figure 1(d), these methods guide the search towards prior-determined promising areas to reduce budget consumption. Unfortunately, acquiring the correct expert knowledge for a specific HPO task is not often easy-to-play, and the transfer quality heavily relies on the hypothesis of task similarity and abundant meta sources. Although some work has demonstrated the optimization robustness regarding potentially misleading priors (Hvarfner et al., 2022), the additional cost for crafting the priors and unpredictable budget consumption discouraging practitioners from exhaustively determining a good prior by leveraging knowledge or transfer for their own HPO tasks.

In this paper, we endeavor to design priors for HPO algorithms with competitive heuristics and consistent budget management, without external cost or budget such as the expert cognitive load and other HPO task evaluation. This is achieved by further exploiting the relation between LF and HF landscapes of HPO tasks. We observe that in many HPO scenarios, promising regions containing good LF and HF solutions overlap to some extent (Sections E and F.3). This motivates us to construct a reasonable reliable prior from the LF evaluations. Our idea is orthogonal to that in the multi-fidelity HPO literature for two reasons. Strategically, we aim to identify the promising regions of LF landscape irrespective of the HF performance. Functionally, the identified LF regions will be used as prior for underlining HPO methods, including the multi-fidelity HPO ones. An additional advantage of our design is that budget for specifying prior can be explicitly integrated to the overall budget in multi-fidelity HPO. Table 1 shows the comparison of HPO methods in terms of budget management and search strategies. A preliminary HPO example is presented in Figure 2 considering two HPO

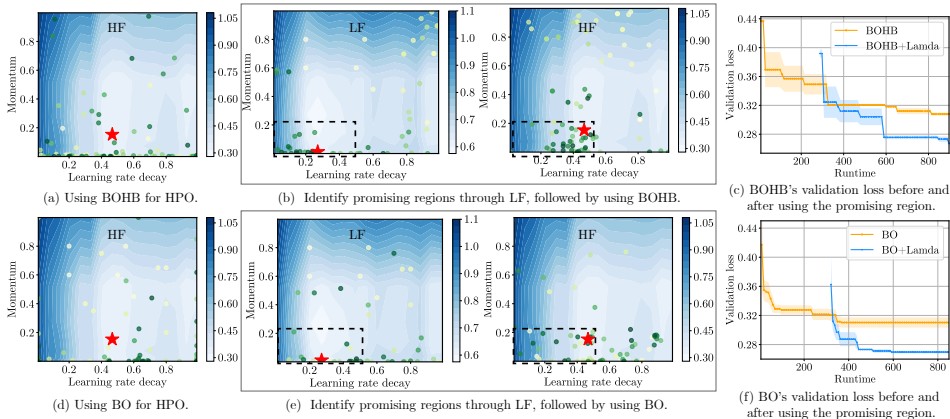

Figure 2: Using BOHB and BO for hyperparameter optimization of a WideResNet on CIFAR-100, before and after employing promising regions (denoted as with or without `Lamda`). The pentagrams mark the optimal solutions. In (a) and (b), the points represent sampled solutions by BOHB, while in (d) and (e), they represent sampled solutions by BO. The color gradient from yellow to green indicates the progression of sampling over time. The points represent samples from BOHB in (a) and (b), and from BO in (d) and (e), with colors transitioning from yellow to green indicating the progression of sampling over time.

Table 1: Existing methods in addressing the challenges of HPO. Methods that successfully address a challenge are marked with a checkmark (✓), while those that do not are marked with a cross (✗).

| Challenges | RS | BO | TuRBO | MFBO | Bandit-based | BO with prior | Transfer search space | Priorband | Ours |
|---|---|---|---|---|---|---|---|---|---|
| Strategic Search | ✗ | ✗ | ✓ | ✗ | ✗ | Use expert prior | Use similar tasks | Use expert prior | Use LF |
| Fidelity management | ✗ | ✗ | ✗ | ✓ | ✓ | ✗ | ✗ | ✓ | ✓ |
| Consistent budget | ✓ | ✓ | ✓ | ✓ | ✓ | ✗ | ✗ | ✗ | ✓ |

methods, BOHB (Falkner et al., 2018) and BO (Bergstra et al., 2011) with noninformative priors and LF-guided priors. In this case, the promising regions for both high- and low-fidelity problems were primarily concentrated within regions bounded by momentum values between $[0, 0.5]$ and learning rate decay values between $[0.2, 0.8]$, as shown in Figure 2(b) and (e). Algorithms with LF-guided prior can quickly explore the real promising regions (right panel of Figure 2(b) and (e)). Moreover, in Figure 2(c) and (f), while identifying the LF-guided prior consumes a certain amount of budget, the overall efficiency is significantly improved, which highlights our motivations.

Overall, we propose a two-phase multi-fidelity HPO framework, named Lamda (**L**e**a**rning pro**m**ising regions from **da**ta), which is algorithm-agnostic and serves as a booster for existing HPO algorithms within the Bayesian routine. The contributions are threefold:

- Building on the overlapping promising regions between LF and HF landscapes, we develop a framework that first introduces LF evaluations to identify the promising regions of LF problems, constructing a reasonably reliable prior for underlining HPO algorithms, and then leverages this prior to enhance the HPO algorithms. In addition, an overlapping coefficient is introduced to quantitatively measure the extent of overlapping.

- We integrate the learned prior with various existing HPO algorithms, ranging from prior- and bandit-based methods, as well as multi-fidelity BO, to augment their performance. The rational of this augmentation was demonstrated by showcasing theoretical analysis towards the prior-based Bayesian optimization and bandit-based Hyperband.

- We validate the competitiveness of our methods across diverse hyperparameter optimization tasks, including fully connected networks, transformers, ResNet, neural architecture search benchmarks, joint architecture and hyperparameter search cases, as well as fine-tuning pretrained image classification models.

## 2. MULTI-FIDELITY HPO BY LEARNING PROMISING REGIONS FROM DATA

The HPO problem is formulated as *minimizing* an expensive-to-evaluate objective function $f : \mathcal{X} \to \mathbb{R}$, where the goal is to find

$$\mathbf{x}^* \in \arg\min_{\mathbf{x} \in \mathcal{X}} f(\mathbf{x}). \tag{1}$$

The configuration $\mathbf{x}$ is selected from a search space $\mathcal{X}$ that may include any combination of continuous, discrete, and categorical variables. In the context of HPO, $f(\mathbf{x})$ represents the training or validation performance of a machine learning model given the hyperparameters defined by $\mathbf{x}$. In multi-fidelity HPO, $f_z(\cdot)$ with $z \in \{\ell, \ell+1, \ldots, h\}$ is introduced to denote a computation of $f(\cdot)$ at the fidelity level $z$, e.g., the validation loss of a model trained for $z$ epochs. Define $f_h$ and $f_\ell$ with $\ell < h$ as the HF and LF objectives, respectively.

In this paper, we propose a multi-fidelity HPO framework by exploiting promising regions from data (dubbed Lamda). It comprises a two-phase search strategy (as depicted in Algorithm 1): ▶ the *first-phase search* initially identifies promising regions in the LF landscape (Lamda-1 in Algorithm 2); ▶ the *second-phase search* leverages the learned information to guide the search in the HF landscape (Lamda-2 showcased in Appendix G). We will introduce the search strategies in different phases by addressing two key questions.

---

**Algorithm 1:** Pseudocode for Lamda

**Input:** Total budget $\Lambda$, maximum first-phase budget $B$, configuration parameters $l$.
1 $(\varphi_{\text{pro}}(\mathbf{x}), S, \Lambda_l) \leftarrow$ Lamda-1 $(B, l)$;
2 $\Lambda \leftarrow \Lambda - \Lambda_l$;
3 $x^* \leftarrow$ Lamda-2 $(\varphi_{\text{pro}}(\mathbf{x}), \Lambda_r)$;
4 **return** $x^*$

---

### 2.1 HOW TO IDENTIFY THE PROMISING REGIONS IN THE LOW-FIDELITY LANDSCAPE?

Our basic idea of the *first-phase search* is to divide the LF landscape into two parts: one consists of the promising regions while the other represents the inferior ones. This can be implemented as a binary classification problem. To train such classifier, we leverage the configurations visited so far during the HPO process in the LF landscape, denoted as $\mathcal{S} = \{\langle \mathbf{x}^i, f_\ell(\mathbf{x}^i) \rangle\}_{i=1}^t$ where $f_\ell(\cdot)$

is the LF objective function and $t$ is the current number of function evaluations. In particular, this paper adopts the classic tree-structured Parzen estimator (TPE) method (Bergstra et al., 2011; Gramacki, 2018) as the classifier, given the scalability and supports for both mixed continuous and discrete spaces. It uses the quantile of $\{f_\ell(\mathbf{x})|\mathbf{x} \in \mathcal{S}\}$ to determine the classification boundary. Specifically, we divide $\mathcal{S}$ into: $\mathcal{S}_{\text{pro}} = \{\mathbf{x} \mid f_\ell(\mathbf{x}) \le y^*, \mathbf{x} \in \mathcal{S}\}$ containing promising solutions, and $\mathcal{S}_{\text{inf}} = \{\mathbf{x} \mid f_\ell(\mathbf{x}) > y^*, \mathbf{x} \in \mathcal{S}\}$ containing inferior solutions, where $y^*$ is determined from $\alpha = \Pr(f_\ell(\mathbf{x}) < y^*)$ quantile of $\{f_\ell(\mathbf{x})|\forall \mathbf{x} \in \mathcal{S}\}$. Then we denote

$$\varphi_{\text{pro}}(\mathbf{x}) = p\left(\mathbf{x} \mid \mathcal{S}_{\text{pro}}\right), \quad \varphi_{\text{inf}}(\mathbf{x}) = p\left(\mathbf{x} \mid \mathcal{S}_{\text{inf}}\right), \quad (2)$$

where $\varphi_{\text{pro}}(\mathbf{x})$ is the probability density function (PDF) of the promising solutions, and $\varphi_{\text{inf}}(\mathbf{x})$ is the PDF of the inferior solutions. We will adopt the kernel density estimation for $\varphi_{\text{pro}}(\mathbf{x})$ and $\varphi_{\text{inf}}(\mathbf{x})$, given its non-parametric nature and applicability to complicated distributions (Chen, 2017).

Instead of searching for the optimal configurations in the LF landscape, the purpose of the *first-phase search* is to identify the promising regions. In practice, the targeted regions are relatively scattered at the beginning and will gradually become focused around the regions that potentially cover the optima (see an illustrative example in Figure 3). Based on this observation, we hypothesize that the *first-phase search* can be terminated when the distribution of promising regions becomes stable. To keep track of the progression of such distribution, we propose to use the overlapping coefficient (OVL) (Anderson et al., 2012) as a metric to quantify the similarity between two distributions.

---

**Algorithm 2:** Pseudocode for `Lamda-1`

**Input:** Maximum first-phase budget $B$, threshold $y^*$, $\Delta$, $\gamma$, fidelity level $l$, budget function $\lambda_z$.

1 Initialize $S \leftarrow \emptyset$, $\Lambda_l \leftarrow 0$, $t \leftarrow 0$, isStable $\leftarrow$ **False**;
2 **while** $\Lambda_l < B$ **and not** *isStable* **do**
3   $\mathbf{x}^t \leftarrow \arg\max_{\mathbf{x} \in \mathcal{X}} \text{AF}(\mathbf{x}, \mathcal{S})$ ;
4   $y^t \leftarrow f_\ell(\mathbf{x}^t) + \epsilon$;
5   $S \leftarrow S \cup \{(\mathbf{x}^t, y^t)\}$;
6   Update $\mathcal{S}_{\text{pro}}, \mathcal{S}_{\text{inf}}, \varphi_{\text{pro}}^t(\mathbf{x}), \varphi_{\text{inf}}^t(\mathbf{x})$;
7   **if** $1 - \rho\left(\varphi_{\text{pro}}^t(\mathbf{x}), \varphi_{\text{pro}}^{t+\triangle}(\mathbf{x})\right) \le \gamma$ **then**
8     isStable $\leftarrow$ **True**;
9   $\Lambda_l \leftarrow \Lambda_l + \lambda_z(\mathbf{x}^t, l), t \leftarrow t + 1$;
10 $\varphi_{\text{pro}}(\mathbf{x}) \leftarrow \varphi_{\text{pro}}^t(\mathbf{x})$;
11 **return** $(\varphi_{\text{pro}}(\mathbf{x}), S, \Lambda_l)$

---

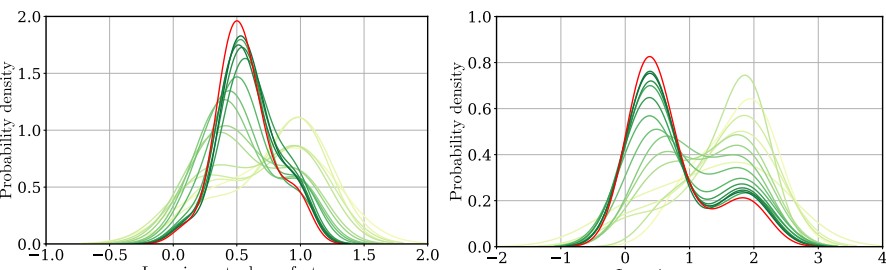

Figure 3: This figure shows the progression of PDFs for promising regions during hyperparameter optimization on a transformer model for the LM1B dataset, focusing on the LF problem. We display PDFs for two out of four hyperparameters, with colors changing from yellow to green to indicate iteration progress. The red line represents the true PDF of the promising solutions in the LF problem.

**Definition 1** (Overlapping coefficient). *Let $\varphi_1(\mathbf{x})$ and $\varphi_2(\mathbf{x})$ be two PDFs on the search space $\mathcal{X}$. The overlapping coefficient $\rho$ of the two functions is defined as:*

$$\rho\left(\varphi_1(\mathbf{x}), \varphi_2(\mathbf{x})\right) = \int_{\mathbf{x} \in \mathcal{X}} \min\left\{\varphi_1(\mathbf{x}), \varphi_2(\mathbf{x})\right\} d\mathbf{x}. \quad (3)$$

Note that $\rho\left(\varphi_1(\mathbf{x}), \varphi_2(\mathbf{x})\right)$ ranges from 0 to 1, where $\rho = 1$ if and only if the two distributions are fully overlapped, and $\rho = 0$ if there is no intersection at all. The *first-phase search* is terminated either if the allocated computational budget is exhausted or the OVL of estimated distributions between $\Delta \in \mathbb{N}$ iterations is close enough:

$$1 - \rho\left(\varphi_{\text{pro}}^t(\mathbf{x}), \varphi_{\text{pro}}^{t+\triangle}(\mathbf{x})\right) \le \gamma, \quad (4)$$

where $\gamma$ denotes the threshold. The calculation of $\rho$ involves a multidimensional integral, which can be numerically intractable. In practice, we employ the Monte Carlo method to estimate $\rho$ as

$$
\begin{aligned}
\rho\left(\varphi_1(\mathbf{x}), \varphi_2(\mathbf{x})\right) &= \int_{\mathbf{x} \in \mathcal{X}} \min\left\{\varphi_1(\mathbf{x}), \varphi_2(\mathbf{x})\right\} d\mathbf{x} = \int_{\mathbf{x} \in \mathcal{X}} \min\left\{1, \frac{\varphi_2(\mathbf{x})}{\varphi_1(\mathbf{x})}\right\} \varphi_1(\mathbf{x}) d\mathbf{x} \\
&= \mathbb{E}\left[\min\left\{1, \frac{\varphi_2(\mathbf{x})}{\varphi_1(\mathbf{x})}\right\}\right] \approx \frac{1}{N}\sum_{i=1}^{N} \min\left\{1, \frac{\hat{\varphi}_2(\mathbf{x})}{\hat{\varphi}_1(\mathbf{x})}\right\},
\end{aligned}
\tag{5}
$$

where $N$ is the number of samples used in the Markov Chain Monte Carlo sampling, and $\hat{\varphi}(\cdot)$ is an approximation of $\varphi(\cdot)$ such as using the kernel density estimation.

## 2.2 How to Leverage LF Promising Regions in the High-fidelity Landscape?

With the identified promising regions in the LF landscape, we hypothesize that such information can be used to define the promising regions in the HF landscape. Instead of searching among the entire search space, the *second-phase search* is more focused within the regions defined below:

$$
\tilde{\varphi}_{\text{pro}}(\mathbf{x}) = (1 - w) \cdot \varphi(\mathbf{x}) + w \cdot \varphi_{\text{pro}}(\mathbf{x}), \tag{6}
$$

where $\varphi(\mathbf{x})$ is the probability distribution used to guide the HPO process in the HF landscape, $\varphi_{\text{pro}}(\mathbf{x})$ is the probability distribution of the promising regions identified from the *first-phase search* in the LF landscape, and $w \in [0, 1)$ with is a hyperparameter that controls the trade-off between the importance of $\varphi(\mathbf{x})$ learned *on-the-fly* and $\varphi_{\text{pro}}(\mathbf{x})$ learned in the *first-phase search*. Figure 4 provides a conceptual visualization of leveraging equation (6) during the second-phase search. The redefined

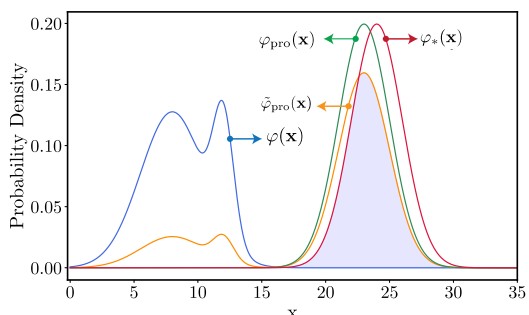

Figure 4: Conceptual visualization of leveraging promising regions: $\varphi(\mathbf{x})$, $\varphi_{\text{pro}}(\mathbf{x})$, $\tilde{\varphi}_{\text{pro}}(\mathbf{x})$, and $\varphi_*(\mathbf{x})$ represent the original sampling distribution, the density function of the promising regions, the modified density function incorporating the promising regions with a weight of $w = 0.8$, and the density function of the real optimum.

promising regions will be closer to the true optimal solution if the promising regions learned in the first phase are closer to the optimum than those before the redefinition, as proven in Proposition 1. Note that since $\varphi(\mathbf{x})$ is progressively updated during the HPO process with new configurations evaluated and added to the dataset, it is expected that $\tilde{\varphi}_{\text{pro}}(\mathbf{x})$ will experience a similar trend as $\varphi_{\text{pro}}(\mathbf{x})$ in the *first-phase search*.

## 2.3 Integration and Comparison with Current Multi-fidelity HPO Methods

Instead of a standalone algorithm, `Lamda` plays as a booster that can be integrated with any existing multi-fidelity HPO methods with minor adaptation and thus augmenting the performance of the baseline optimizer. In a nutshell, we only need to replace the sampling strategies of the baseline optimizer with $\tilde{\varphi}_{\text{pro}}(\mathbf{x})$. We justify this by comparing with current multi-fidelity HPO methods.

- **Prior-Based Methods:** By using `Lamda`, $\tilde{\varphi}_{\text{pro}}(\mathbf{x})$ serves as *a priori* knowledge that represents a reasonable estimation of promising regions in the *second-phase search*. Unlike prior-based methods, which depend on prescribed knowledge or experience from domain experts, $\tilde{\varphi}_{\text{pro}}(\mathbf{x})$ is adaptively learned from data during the *first-phase search* through the HPO process in the LF landscape. As a result, our approach is resilient to 'pathological priors'—whether misleading, lacking in informative values, or potentially adversarial—which are not uncommon when tackling new, unseen real-life black-box applications. Additionally, we expect a scenario between our data-driven priors with those elicited from experts can offer consolidated performance enhancement.

- **Bandit-Based Methods:** By using `Lamda`, $\tilde{\varphi}_{\text{pro}}(\mathbf{x})$ serves as an effective alternative to the sampling distributions used in bandit-based methods. This restricts the HPO process to focus exclusively on the learned promising regions. In contrast, bandit-based methods begin with LF assessments to identify candidates for HF evaluation and then gradually shift the search focus towards these identified areas. This process, which alternates between exploration and exploitation throughout the entire search space, often leads to inefficient use of computational resources by exploring less promising regions.

- **MFBO Methods:** Similar to bandit-based methods, MFBO methods use an acquisition function learned from data collected across multiple fidelities to explore the entire search space. This can lead to unnecessary exploration of less promising regions. By using `Lamda`, $\tilde{\varphi}_{\text{pro}}(\mathbf{x})$ restricts the search space within the learned promising regions. Note that this strategy can be applied to the other BO variants.

For proof-of-concept purposes, we choose `PriorBand`, `BOHB`, `MUMBO` as representatives of the prior-, bandit-based and MFBO methods, respectively. By augmenting with `Lamda`, we have `Lamda+PriorBand`, `Lamda+BOHB`, and `Lamda+MUMBO`. In addition, it is also interesting to see whether `Lamda` can be useful for vanilla BO and even random search. To this end, we derive two other variants `Lamda+BO` and `Lamda+RS`. We provide multiple pseudocode to demonstrate how `Lamda-2` can be adapted to various algorithms, including `PriorBand`, `BOHB`, `MUMBO`, vanilla `BO`, and `random search`. The details for each integration are in Appendix G. Furthermore, we provide theoretical analysis under both the prior-based BO framework and bandit-based Hyperband framework, indicating the rational of this augmentation. For `Lamda+BO`, it incorporates prior knowledge in the acquisition function:

$$\mathbf{x}^{n+1} = \arg\max \tilde{\varphi}_{\text{pro}}(\mathbf{x})\text{AF}(\mathbf{x}, \mathcal{D}), \tag{7}$$

where AF is the acquisition function in vanilla BO such as the expected improvement (EI) considered in this paper. $\mathcal{D} = \{\langle \mathbf{x}^i, f_h(\mathbf{x}^i)\rangle\}_{i=1}^n$ where $f_h(\cdot)$ is the HF objective function. The Gaussian process regression is employed as the surrogate model of $f_h(\cdot)$. For a solution $\tilde{\mathbf{x}}$, the predicted mean and variance of the value distribution of $f_h(\tilde{\mathbf{x}})$ are $\mu_f(\tilde{\mathbf{x}})$ and $\sigma_f^2(\tilde{\mathbf{x}})$.

In `Lamda+BO`, we apply the widely used EI as the acquisition function in equation (7) given as

$$\text{EI}(\tilde{\mathbf{x}}|\mathcal{D}) = \sigma_f(\tilde{\mathbf{x}})\big(z\Phi_f(z) + \phi_f(z)\big), \tag{8}$$

where $z = \frac{f_{\mathcal{D}}^* - \mu_f(\tilde{\mathbf{x}})}{\sigma_f(\tilde{\mathbf{x}})}$, $f_{\mathcal{D}}^* = \min_{\langle \mathbf{x}, f_h(\mathbf{x})\rangle \in \mathcal{D}} f_h(\mathbf{x})$, $\Phi_f$ and $\phi_f$ denote the cumulative distribution function and probability density function, respectively.

**Theorem 1.** *Given $\mathcal{D}_n$, $\tilde{\varphi}_{\text{pro}}(\mathbf{x})$, and applying the EI into equation (7), assume the GP models are non-degenerated. Let $\mathcal{D}$ be the collected observations with $\langle \mathbf{x}^1, f_h(\mathbf{x}^1)\rangle$ fixed while $\left\{\langle \mathbf{x}^i, f_h(\mathbf{x}^i)\rangle\right\}_{i=2}^n$ are sequentially chosen by*

$$\mathbf{x}^{n+1} = \arg\max \tilde{\varphi}_{\text{pro}}(\mathbf{x})\text{EI}(\tilde{\mathbf{x}}|\mathcal{D}). \tag{9}$$

*Then, as $n \to \infty$, almost surely: the acquisition function converges to zero; and the evaluated best objective $f_{\mathcal{D}}^* \to f_h^*$, where $f_h^*$ represents the global optimum of $f_h(\cdot)$.*

The proof is given in Appendix 11A.2. Unlike the theory in (Hvarfner et al., 2022), the convergence property of equation (9) does not need a decaying factor to force exponentially decreasing of the priors of promising LF regions. Based on the dynamic update of $\tilde{\varphi}_{\text{pro}}(\mathbf{x})$, it is sufficient to capture the promising regions in the HF landscape according to the overlap.

For bandit-based method, we study the `Lamda+PriorBand` in the theoretical routine of Hyperband. The algorithm involves two loops. In the outer loop, at the $k$-th round, the algorithm allocates $B_{k,s} = 2^k + \text{poly}(k)$ budgets and $n_{k,s} = 2^s$ configurations randomly sampled from $\tilde{\varphi}_{\text{pro}}(x)$, for $s = 0, 1, \ldots, s_{\max}$, subject to $s_{\max} + \log_2(s_{\max}) < k$, where $\text{poly}(k)$ is some polynomial function w.r.t $k$. In the inner loop, the successive halving algorithm is leveraged to find the best arm among the $n_{k,s}$ arms with $B_{k,s}$ budget. In the context of multi-arm bandits, each configuration $\mathbf{x}^i$ corresponds to an independent arm to pull, whose reward with the $j$-th pull is denoted by $l_{i,j} = f_j(\mathbf{x}^i)$. We assume there exists $\lim_{k\to\infty} l_{i,k} = \nu_i$ for all $\mathbf{x}^i \in \mathcal{X}$, and denote $\nu_* = \inf_{x\in\mathcal{X}} \nu_i$. Denote also that the distribution of $v$ as $F$ satisfying $P(\nu - \nu_* \le \epsilon) = F(\nu_* + \epsilon)$ for any $\epsilon$. The inverse function is defined by $F^{-1}(y) = \inf\{x : F(x) \le y\}$. In addition, there exists a monotonically decreasing function $\gamma(t) : N \to R$ satisfying $\sup_i |l_{i,t} - \nu_i| \le \gamma(t)$.

**Theorem 2.** *For fixed $\delta \in (0, 1)$. Let $\hat{\nu}_B$ be the empirically best-performing arm output from successive halving of round $k_B = \log_2(B)$ of the outer loop, and let $s_B < k_B$. Then, there is:*

$$\hat{\nu}_B - \nu_* \le 3\left(F^{-1}\left(\frac{\log(4k_B^3/\delta)}{2^{s_B}} - \nu_*\right) + \gamma\left(\frac{2^{k_B-1}}{k_B}\right)\right), \tag{10}$$

*where $s_B$ satisfies $2^{k_B} + \text{poly}(k_B) > 4s_B\mathbf{H}(F, \gamma, 2^{s_B}, 2k_B^3/\gamma)$ with $\mathbf{H}(F, \gamma, n, \delta) = 2n\int_{p_n}^1 \gamma^{-1}(\frac{F^{-1}(t)-\nu^*}{4})dt + \frac{10}{3}\log(2/\delta)\gamma^{-1}\left(\frac{F^{-1}(p_n)-\nu_*}{4}\right)$ and $p_n = \frac{\log_2(2/\delta)}{n}$.*

Since all configurations for successive halving tasks are sampled randomly from a probability distribution described by $\tilde{\varphi}_{\text{pro}}$, the theoretical results, specifically Corollary 3 in (Li et al., 2017), still hold in this case. Different from Hyperband that relied on non-adaptive grid search exhausting $c\log_2(B)$ overall budgets with some constant $c$, we sample configurations and allocate budget through both grid search and adaptive design based on $\tilde{\varphi}_{\text{pro}}(\mathbf{x})$. Theoretically, this requires the same order of budgets as Hyperband. It will be a quite interesting question to ask how the fact of overlapping can help avoiding the grid search of Hyperband, which will be our future work.

## 3. EXPERIMENT SETUP

### 3.1 BENCHMARK SUITES

Our experiments consider 56 benchmarks that cover various search spaces including mixed types and log-scaled hyperparameters. Further, they involve a wide range of downstream tasks including image classification, language modeling, tabular data processing, medical applications, and translation.

They are selected from four sources. ① Tabular benchmarks include ▶ four cases from `FCNet` (Pfisterer et al., 2022), each with 6 hyperparameters; ▶ one from `NAS-Bench-301` with 34 hyperparameters (Pfisterer et al., 2022); ▶ three from `NAS-Bench-201`, each with 6 hyperparameters (Eggensperger et al., 2021); and ▶ twenty benchmarks from `rpart` on decision tree, `glmnet` on elastic net, `ranger` on random forest, and `XGBoost` (Eggensperger et al., 2021). ② Surrogate benchmarks include ▶ four problems from `PD1 benchmarks` with 4 hyperparameters (Mallik et al., 2023; Wang et al., 2021); ▶ three problems from `JAHSBench` (Mallik et al., 2023; Bansal et al., 2022) with 14 mixed-type hyperparameters for tuning both the neural networks architecture and training hyperparameters. ③ Training two deep neural networks include LeNet on CIFAR-10, and ResNet-18 on CIFAR-10 and CIFAR-100 with 5 hyperparameters. ④ Two synthetic Hartmann functions (Mallik et al., 2023) with three and six variables respectively. ⑤ 20 tasks for fine-tuning pretrained image classification models (Pineda-Arango et al., 2024).

For the tabular and surrogate benchmarks, we use the number of epochs as the parameter to set the fidelity level. As for the training of deep neural networks, the size of the dataset is used as the parameter to control the fidelity level, as shown in Table 5. Our experiments have considered different scenarios where the promising regions of LF and HF landscape have varying levels of overlaps (Table 5 in Appendix C.4 provides statistics of the overlapping rates). Further detail about all benchmarks are provided in Appendix C.

### 3.2 PEER ALGORITHMS

We choose nine peer algorithms as the baselines to validate the effectiveness of proposed approach. They are ▶ `PriorBand` (Mallik et al., 2023) and `PFNs4BO` (Müller et al., 2023) as prior-based methods; ▶ `HyperBand` (Li et al., 2017), `BOHB` (Falkner et al., 2018), and `Hyper-Tune` (Li et al., 2022b) as bandit-based methods; ▶ `MUMBO` (Li et al., 2021a) and `DPL` (Kadra et al., 2023) as MFBO methods; and ▶ random search (`RS`), `BO` and `TuRBO` (Eriksson et al., 2019) as other popular HPO methods. During our experiments, we used the default values for these algorithms. For our algorithm, the parameter settings are as follows: $\gamma = 0.1$, $\Delta = 5$, $\alpha = 15$ and $w = 0.5$. Additionally, we allocate a computational budget of $B = 5D$ high-fidelity resources in the first-phase search, where $D$ denotes the problem's dimensionality.

## 4. EXPERIMENTAL RESULTS

### 4.1 EFFECTIVE OF USING PROMISING REGIONS

**Results on tabular, surrogate and synthetic benchmarks**: This experiment demonstrates how our algorithm framework improves five commonly used optimizers: `PriorBand`, `BOHB`, `MUMBO`, `BO` and `RS`. For `BOHB`, `MUMBO`, `BO` and `RS`, 33 tasks from the above tabular, surrogate and synthetic benchmarks are used. For `PriorBand`, the evaluation included eight tasks: four from the original paper (PD1-LM1B, PD1-WMT, MFH3, and MFH6) and four additional tasks in FCNet. In addition, good prior are used at `PriorBand` for PD1-LM1B, PD1-WMT, MFH3, and MFH6. Table 2 presents the numbers of win/lose/tie obtained by using the Wilcoxon signed-rank test and Figure 5 shows the average rank over the HPO tasks. According to Figure 5, it can seen that using the strategy of promising regions can obtain better results on all the five algorithms. The results of the Wilcoxon signed-rank test in Table 2 also validate the efficiency of using promising regions. Using promising regions can achieve significantly better results than the baseline in more than half of the tasks. For the remaining problems, they obtain results equal to those of the baseline. Regarding `PriorBand`, for the four tasks with prior information, `Lamda+PriorBand` achieves results comparable to those of `PriorBand`. However, for the four tasks without prior information,

`Lamda+PriorBand` achieves better results. For `BOHB`, using the strategy of promising regions improves in rank as the consumed resources grow. The main reason for such slow-starting may caused by the resources needed for finding promising regions. However, it quickly takes the top rank after using about 15 HF resources. `MUMBO`, `BO` and `RS` present similar performances in the condition of using promising regions.

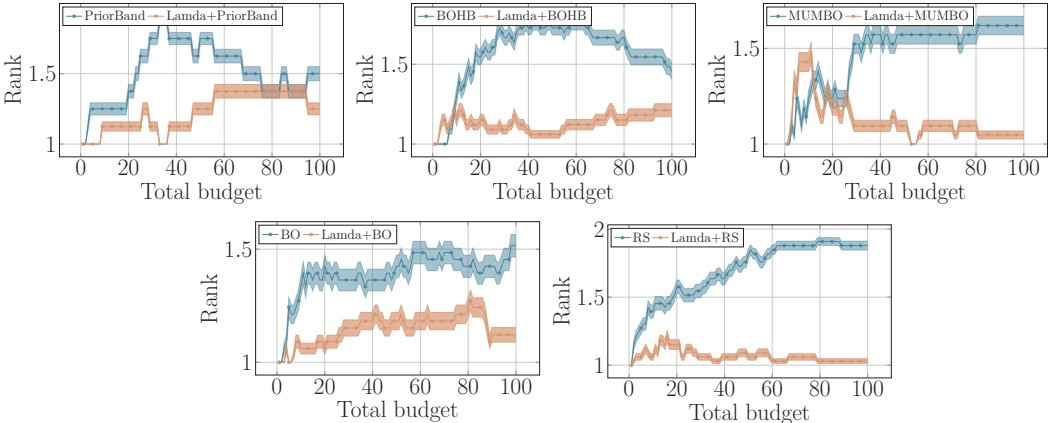

Figure 5: Comparing average relative ranks of `PriorBand`, `BOHB`, `MUMBO`, `BO` and `RS` under the proposed framework across 33 HPO tasks.

Table 2: Performance comparison (**Win**/Lose/Tie) of `Lamda+PriorBand`, `Lamda+BOHB`, `Lamda+MUMBO`, `Lamda+BO` and `Lamda+RS` against their baselines over 100 HF evaluations.

| Lamda+BOHB | Lamda+MUMBO | Lamda+BO | Lamda+RS | Lamda+PriorBand |
|---|---|---|---|---|
| **24**/0/9 | **19**/0/14 | **17**/1/15 | **29**/1/3 | **3**/0/5 |

Figure 6: Validation error observed in tuning 8 HPO tasks, using `PriorBand` as the baseline.

Figure 6 and Figure 18 to Figure 25 shows the performance curves for each benchmark under the framework of `Priorband`, `BOHB`, `MUMBO`, `BO` and `RS`. Within the framework of `Priorband`, `Lamda+Priorband` converge faster on all the 8 tasks as shown in Figure 6. For other four algorithms, using the promising regions accelerates the discovery of effective solutions compared to the baseline on most of the tasks. In particular, we would like highlight the results on JAHS-CIFAR-10, JAHS-Colorectal-Histology, and JAHS-Fashion-MNIST, whose overlaps are low. Lamda consistently enhance the baseline algorithms.

To better understand the results, we sample 10,000 hyperparameter configurations for each benchmark and evaluate their performance at high and low fidelities. The configurations are mapped into 2D space. We visualize the good solutions ( Figure 14 to Figure 17) and landscape (Figure 33 to Figure 36) of the benchmarks at low and high fidelities. For FCNet, PD1, and NAS-Bench-201, the gap between `Lamda+BOHB` and the naïve `BOHB` is caused by the great overlapping between the high and low fidelities as shown in Figure 14. Additionally, Figure 12 and Figure 13 shows the found good solutions by the LF. It can be seen that these solutions are close to the good solutions

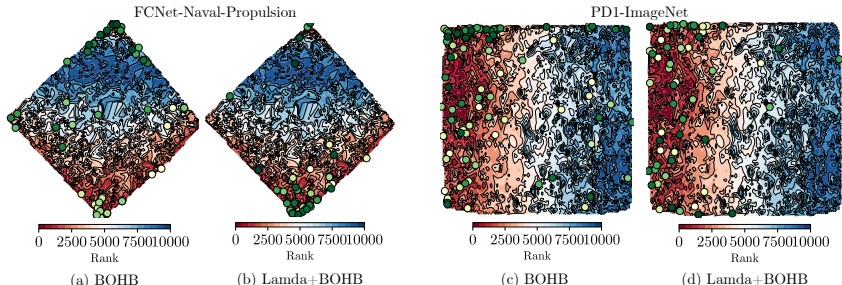

(a) BOHB     (b) Lamda+BOHB     (c) BOHB     (d) Lamda+BOHB

Figure 7: 2D visualization of the sampling points during the optimization process of `Lamda+BOHB` and `BOHB` on FCNet-Naval-Propulsion and PD1-ImageNet. The contour represents the dimensionality-reduced landscape of the HF problem, while the points indicate the HF samples collected during optimization. The color gradient from green to yellow indicates the order of sampling, with yellow representing later sampling stages.

of HF. Additionally, we provide a 2D visualization of the sampling points during the optimization process of `Lamda+BOHB`, `BOHB`, `Lamda+MUMBO`, and `MUMBO` on FCNet-Naval-Propulsion and PD1-ImageNet, as shown in Figure 7 and Figure 37. It can be observed that `Lamda`-based methods tend to focus sampling in regions with better fitness values, whereas `BOHB` and `MUMBO` allocate relatively more resources to exploring areas with moderate fitness values.

**Results on raw problems**: In this part, we evaluate `BOHB` and `Lamda+BOHB` on three raw HPO tasks. Figure 8 shows the the performance curves on three vision problems. We observe that the performance of `Lamda+BOHB` is worse than `BOHB` at the initial iteration. However, it quickly outperforms `BOHB` after some resources. The main reason is that the promising regions at the high and low fidelity have great overlapping as shown Figure 9.

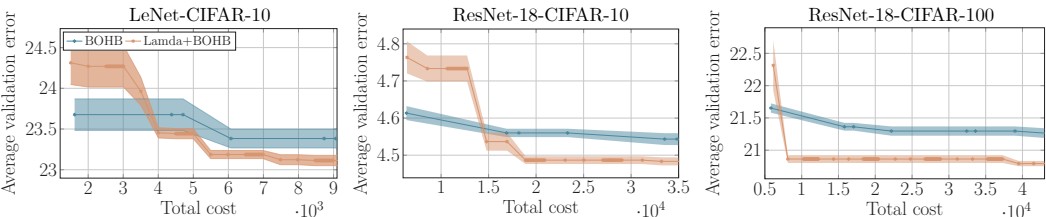

Figure 8: Validation error observed in tuning raw problems.

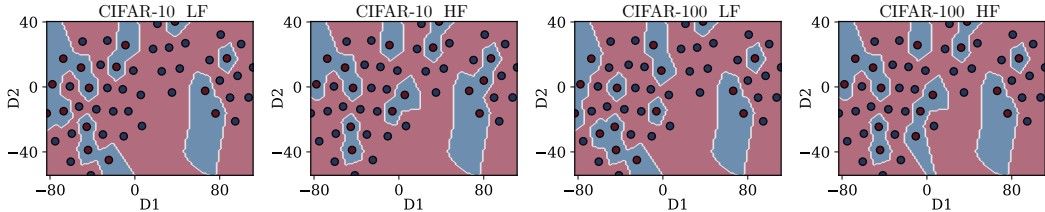

Figure 9: Visualizing the top 30% of solutions (represented by blue regions) in both high and low fidelity while optimizing the hyperparameters of ResNet-18 on CIFAR-10 and CIFAR-100 datasets.

## 4.2 PEER COMPARISON

**Results on tabular and surrogate benchmarks**: This experiment demonstrates how the performance of the proposed algorithm framework compared to the other methods over the number of evaluations. Figure 10 shows the average rank over the 33 tasks. According to Figure 10, we observe that `Lamda+BO` and `Lamda+MUMBO` consistently quickly take the top and keep the rank until the end. It also can be seen that `Hyperband` and `RS` are the two worst algorithms, which may be due to the random sampling strategy.

We have conducted 20 HPO tasks for fine-tuning pretrained image classification models. Experimental results (see Appendix F.4) have also demonstrated that Lamda consistently enhance the baseline algorithms.

## 4.3 PARAMETER ANALYSIS

We investigate the impact of parameters in `Lamda` within the `BOHB` framework, including different budgets ($B$) and thresholds ($\gamma$) for stopping the first phase, the interval for calculating the overlapping coefficient ($\Delta$), the quantile ($\alpha$) used in the TPE, and the initial weight ($w_0$). Four tasks at the XGBoost benchmark, each involving a 10-dimensional hyperparameter optimization problem, are used for the experiments. The budgets are varied as $B = D, 4D, 8D, 10D$, and $20D$, and $\gamma$ values are tested at $0.5, 0.2$, and $0.1$. As shown

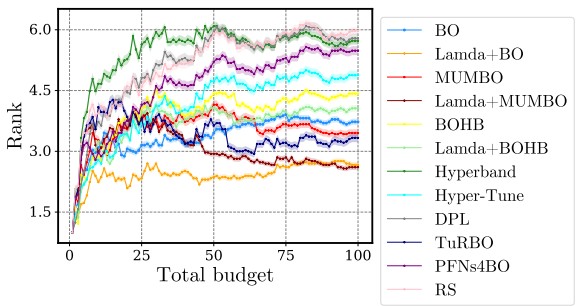

Figure 10: Comparing average relative ranks of peer algorithms across 33 HPO tasks.

in Figure 28, the choice of $B$ influences algorithm performance, where too high a budget ($B > 8D$) leads to excessive resource consumption in the first phase. The impact of $\gamma$ on the algorithm is relatively minor as shown in Figure 29, likely due to the constraints imposed by the maximum budget $B$. Additionally, we investigate the influence of the parameter $\Delta$ on the algorithm's performance, as illustrated in Figure 30. The settings for $\Delta$ are $3, 5, 15, 35$, and $50$. The results indicate that both excessively large and excessively small values of $\Delta$ slightly affect the algorithm's performance. Specifically, values of $\Delta$ that are too low may cause the algorithm to erroneously determine that promising regions have stabilized, whereas excessively high values of $\Delta$ can lead to considerable delays in the algorithm's capacity to verify the stability of these regions. We also analyze the influence of parameter $\alpha$ using the values 5, 15, and 25. The results in Figure 31 indicate no significant impact on the algorithm's performance.

Regarding the initial weight $w_0$, we examine settings of $1, 0.8, 0.5, 0.3$, and $0.1$. The results, illustrated in Figure 32, suggest that while the optimal $w_0$ slightly vary across tasks, its overall impact on the algorithm's performance is minimal. In addition, values above $0.1$ consistently outperform the original `BOHB`. Further analysis of rankings show that settings with $w > 0.1$ achieve better results compared to $w = 0.1$. Notably, a setting of $w = 1$ shows superior performance during the early stages of the optimization. These results also indicate the efficiency of using the promising regions.

## 5. LIMITATIONS AND FUTURE WORK

In this work, we only consider two fidelities for a proof-of-concept purpose. Our next step is to extend the current `Lamda` framework for tackling multiple fidelity levels. In addition, there is a gap on theoretical underpinnings about how the involved parameters, such as the quantile threshold in LF problems and the overlapping coefficient between LF and HF landscape, impact the convergence rate or regret of the HBO methods augmented with `Lamda`. Last but not the least, this paper is mainly designed for multi-fidelity hyperparameter optimization. It will be interesting to explore its applications to a broader range of black-box optimization problems where multi-fidelity experiments and data are prevalent (e.g., computational fluid dynamics optimization in engineering design (Barrett et al., 2006; Liu et al., 2017), and new material (Goldfeld et al., 2005; Khatamsaz et al., 2021), or drug design (Fare et al., 2022; Greenman et al., 2021) in scientific discovery).

## 6. CONCLUSIONS

This paper highlights a common limitation in existing HPO algorithms, which often searches across the whole search space. While some methods leverage prior knowledge to constrain the search space, accessibility to the knowledge is not always guaranteed. To address this challenge, we have developed an algorithmic framework that enables algorithms to autonomously identify promising regions from the LF to accelerate the HPO process, based on the the potential overlap of promising regions between high- and low-fidelity HPO landscape. This framework is integrated with a variety of existing HPO techniques, including prior- and bandit-based methods, as well as multi-fidelity BO, to enhance their efficacy. We support the rationale behind this augmentation through theoretical analysis focused on prior-based Bayesian optimization and bandit-based Hyperband. Our empirical evaluations across diverse hyperparameter optimization tasks—such as fully connected networks, transformers, ResNets, and neural architecture search benchmarks, including joint architecture and hyperparameter searches—demonstrate the competitiveness of our methods.

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

## A. THEORETICAL ANALYSIS

**Proposition 1.** *Assume the OVL between $\varphi(\mathbf{x})$ and the PDF of the true promising solutions $\varphi_*(\mathbf{x})$ is less than the overlapping between the PDF of promising region $\varphi_{\mathrm{pro}}(\mathbf{x})$ and $\varphi_*(\mathbf{x})$. Then, the modified sampling function $\tilde{\varphi}_{\mathrm{pro}}(\mathbf{x}) = w_1 \cdot \varphi(\mathbf{x}) + w_2 \cdot \varphi_{\mathrm{pro}}(\mathbf{x})$, where $w_1 + w_2 = 1$, will have a greater or equal overlapping with $\varphi_*(\mathbf{x})$ compared to the overlapping of $\varphi(\mathbf{x})$ with $\varphi_*(\mathbf{x})$.*

Proposition 1 suggests that incorporating the promising regions into the sampling distribution enhances its alignment with the distribution of the real optimal solutions.

*Proof.* Using the definition of $\rho(\varphi_1, \varphi_2)$ in equation (3), the OVL between $\varphi(\mathbf{x}), \varphi_{\mathrm{pro}}(\mathbf{x}), \tilde{\varphi}_{\mathrm{pro}}(\mathbf{x})$ and $\varphi_*(\mathbf{x})$ are computed as follows:

$$\rho(\varphi(\mathbf{x}), \varphi_*(\mathbf{x})) = 1 - \frac{1}{2} \int_{\mathbf{x} \in \mathcal{X}} |\varphi(\mathbf{x}) - \varphi_*(\mathbf{x})| \, d\mathbf{x},$$

$$\rho(\varphi_{\mathrm{pro}}(\mathbf{x}), \varphi_*(\mathbf{x})) = 1 - \frac{1}{2} \int_{\mathbf{x} \in \mathcal{X}} |\varphi_{\mathrm{pro}}(\mathbf{x}) - \varphi_*(\mathbf{x})| \, d\mathbf{x}, \tag{11}$$

$$\rho(\tilde{\varphi}_{\mathrm{pro}}(\mathbf{x}), \varphi_*(\mathbf{x})) = 1 - \frac{1}{2} \int_{\mathbf{x} \in \mathcal{X}} |\tilde{\varphi}_{\mathrm{pro}}(\mathbf{x}) - \varphi_*(\mathbf{x})| \, d\mathbf{x}.$$

where

$$\rho(\varphi_1(\mathbf{x}), \varphi_2(\mathbf{x})) = \int_{\mathbf{x} \in \mathcal{X}} \min\{\varphi_1(\mathbf{x}), \varphi_2(\mathbf{x})\} \, d\mathbf{x} = 1 - \frac{1}{2} \int_{\mathbf{x} \in \mathcal{X}} |\varphi_1(\mathbf{x}) - \varphi_2(\mathbf{x})| \, d\mathbf{x}. \tag{12}$$

Given that thus $\varphi(\mathbf{x})$ has a smaller overlap with $\varphi_*(\mathbf{x})$ than $\varphi_{\mathrm{pro}}(\mathbf{x})$, it follows that:

$$\rho(\varphi(\mathbf{x}), \varphi_*(\mathbf{x})) - \rho(\varphi_{\mathrm{pro}}(\mathbf{x}), \varphi_*(\mathbf{x}))$$
$$= 1 - \frac{1}{2} \int_{\mathbf{x} \in \mathcal{X}} |\varphi(\mathbf{x}) - \varphi_*(\mathbf{x})| \, d\mathbf{x} - \left( 1 - \frac{1}{2} \int_{\mathbf{x} \in \mathcal{X}} |\varphi_{\mathrm{pro}}(\mathbf{x}) - \varphi_*(\mathbf{x})| \, d\mathbf{x} \right)$$
$$= \frac{1}{2} \int_{\mathbf{x} \in \mathcal{X}} |\varphi_{\mathrm{pro}}(\mathbf{x}) - \varphi_*(\mathbf{x})| \, d\mathbf{x} - \frac{1}{2} \int_{\mathbf{x} \in \mathcal{X}} |\varphi(\mathbf{x}) - \varphi_*(\mathbf{x})| \, d\mathbf{x} \tag{13}$$
$$= \frac{1}{2} \int_{\mathbf{x} \in \mathcal{X}} |\varphi_{\mathrm{pro}}(\mathbf{x}) - \varphi_*(\mathbf{x})| - |\varphi(\mathbf{x}) - \varphi_*(\mathbf{x})| \, d\mathbf{x} \le 0$$

Given the above, if $\rho(\varphi(\mathbf{x}), \varphi_*(\mathbf{x})) < \rho(\tilde{\varphi}_{\mathrm{pro}}(\mathbf{x}), \varphi_*(\mathbf{x}))$, it implies that $\varphi(\mathbf{x})$ has a smaller overlap with $\varphi_*(\mathbf{x})$ than $\tilde{\varphi}_{\mathrm{pro}}(\mathbf{x})$. This can be further analyzed as:

$$\rho(\varphi(\mathbf{x}), \varphi_*(\mathbf{x})) - \rho(\tilde{\varphi}_{\mathrm{pro}}(\mathbf{x}), \varphi_*(\mathbf{x}))$$
$$= 1 - \frac{1}{2} \int_{\mathbf{x} \in \mathcal{X}} |\varphi(\mathbf{x}) - \varphi_*(\mathbf{x})| \, d\mathbf{x} - \left( 1 - \frac{1}{2} \int_{\mathbf{x} \in \mathcal{X}} |w_1 \cdot \varphi(\mathbf{x}) + w_2 \cdot \varphi_{\mathrm{pro}}(\mathbf{x}) - \varphi_*(\mathbf{x})| \, d\mathbf{x} \right)$$
$$= \frac{1}{2} \int_{\mathbf{x} \in \mathcal{X}} |w_1 \cdot \varphi(\mathbf{x}) + w_2 \cdot \varphi_{\mathrm{pro}}(\mathbf{x}) - \varphi_*(\mathbf{x})| \, d\mathbf{x} - \frac{1}{2} \int_{\mathbf{x} \in \mathcal{X}} |\varphi(\mathbf{x}) - \varphi_*(\mathbf{x})| \, d\mathbf{x}$$
$$\le \frac{1}{2} \int_{\mathbf{x} \in \mathcal{X}} w_1 \cdot |\varphi(\mathbf{x}) - \varphi_*(\mathbf{x})| + w_2 \cdot |\varphi_{\mathrm{pro}}(\mathbf{x}) - \varphi_*(\mathbf{x})| - |\varphi(\mathbf{x}) - \varphi_*(\mathbf{x})| \, d\mathbf{x}$$
$$= \frac{1}{2} \int_{\mathbf{x} \in \mathcal{X}} w_2 \cdot |\varphi_{\mathrm{pro}}(\mathbf{x}) - \varphi_*(\mathbf{x})| - w_2 \cdot |\varphi(\mathbf{x}) - \varphi_*(\mathbf{x})| \, d\mathbf{x}$$
$$= \frac{w_2}{2} \int_{\mathbf{x} \in \mathcal{X}} |\varphi_{\mathrm{pro}}(\mathbf{x}) - \varphi_*(\mathbf{x})| - |\varphi(\mathbf{x}) - \varphi_*(\mathbf{x})| \, d\mathbf{x} \le 0$$

$$\tag{14}$$

where the inequality is obtained with the following formula:

$$|w_1 \cdot \varphi(\mathbf{x}) + w_2 \cdot \varphi_{\mathrm{pro}}(\mathbf{x}) - \varphi_*(\mathbf{x})|$$
$$= |w_1 \cdot \varphi(\mathbf{x}) - w_1 \cdot \varphi_*(\mathbf{x}) + w_2 \cdot \varphi_{\mathrm{pro}}(\mathbf{x}) - w_2 \cdot \varphi_*(\mathbf{x})| \tag{15}$$
$$\le w_1 \cdot |\varphi(\mathbf{x}) - \varphi_*(\mathbf{x})| + w_2 \cdot |\varphi_{\mathrm{pro}}(\mathbf{x}) - \varphi_*(\mathbf{x})|$$

This implies that $\tilde{\varphi}_{\mathrm{pro}}(\mathbf{x})$ has a greater overlap with $\varphi_*(\mathbf{x})$ than $\varphi(\mathbf{x})$ does. $\qquad \square$

A.2 PROOFS OF THEOREM 1

In the context of sequential design, let $\mathcal{F}_N$ denote the $\sigma$-algebra generated by the random variables $\mathbf{x}^1, Z^1, \ldots, \mathbf{x}^N, Z^N$ where $Z^i$ is the observation of $f_l(\mathbf{x}^i)$. Additionally, let $\mathcal{F}_{N,\tilde{\mathbf{x}}}$ be the $\sigma$-algebra generated by $\mathbf{x}^1, Z^1, \ldots, \mathbf{x}^N, Z^N, \tilde{\mathbf{x}}, \tilde{Z}$ with $\tilde{Z}$ the observation of $f_h(\tilde{\mathbf{x}})$. Then, the EI-based sequential design of `Lamda+BO` takes the following form:

$$\mathbf{x}_{N+1} = \arg\max_{\tilde{\mathbf{x}} \in \Omega} \mathbb{E}_N \left[ M_N^0 - M_{N,\tilde{\mathbf{x}}}^\rho \right], \tag{16}$$

in which $y_h^*$ is the threshold of promising solutions in HF problems lower bounded by the $\alpha$ quantile of $\{f_\ell(\mathbf{x}) | \forall \mathbf{x} \in \mathcal{S}\}$ due to the overlapping between HF and LF landscape, and

$$M_{N,\tilde{\mathbf{x}}}^\rho = \min_{\mathbf{x} \in \mathcal{X}, \, \mathbb{P}(f_h(\mathbf{x}) < y_h^*) | \mathcal{F}_{N,\tilde{\mathbf{x}}}) = 1, \, \sigma_f(\mathbf{x} | \mathcal{F}_{N,\tilde{\mathbf{x}}}) = 0} \tilde{f}(\mathbf{x}), \tag{17}$$

$$M_N^0 = \min_{\mathbf{x} \in \mathcal{X}, \, \mathbb{P}(f_h(\mathbf{x}) < y_h^* | \mathcal{F}) = 1, \, \sigma_f(\mathbf{x} | \mathcal{F}_N) = 0} \tilde{f}(\mathbf{x}). \tag{18}$$

When GPs are non-degenerate, i.e., $\sigma_f = 0$ only if $\mathbf{x} \in \mathcal{D}$ equation (16) becomes equivalent to equation (9). Specifically, $M_N^0$ will be equal to $f_{\mathcal{D}}^*$ in equation (8), while $M_{N,\tilde{\mathbf{x}}}^\rho$ will be the predicted promising HF objective value under threshold $y_h^*$ implicitly defined by equation (6). Next, we present the criteria for *asymptotic convergence* of `Lamda+BO`. The proof of the first statement, i.e., the convergence of the acquisition function, compromises three steps.

**Step 1. `Lamda+BO` serves as a stepwise uncertainty reduction (SUR) sequential design.** For $N \geq 2$, a minimization version of equation (16) can be given as

$$\mathbf{x}_{N+1} = \arg\min_{\tilde{\mathbf{x}} \in \mathcal{X}} \mathbb{E}_N \left[ H_{N,\tilde{\mathbf{x}}} \right], \tag{19}$$

in which

$$H_{N,\tilde{\mathbf{x}}} = M_{N,\tilde{\mathbf{x}}}^\rho - M_N^0 = \mathbb{E}_{N,\tilde{\mathbf{x}}} \left[ M_{N,\tilde{\mathbf{x}}}^\rho - \min_{\mathbf{x} \in \mathcal{X}, \, \mathbb{P}(f_h(\mathbf{x}) < y_h^*)} \tilde{f}(\mathbf{x}) \right].$$

The above equation holds since: $i)$ $M_N^0$ is independent from $\tilde{\mathbf{x}}$, and $ii)$ $\mathbb{E}_{N,\tilde{\mathbf{x}}} \left[ M_{N,\tilde{\mathbf{x}}}^\rho \right] = M_{N,\tilde{\mathbf{x}}}^\rho$ for minimum operation. Therefore this strategy can be transformed into an equivalent SUR sequential design strategy for $H_{N,\tilde{\mathbf{x}}}$. Likewise, we define

$$H_N = \mathbb{E}_N \left[ M_N^\rho - \min_{\mathbf{x} \in \mathcal{X}, \, f_h(\mathbf{x}) < y_h^*} \tilde{f}(\mathbf{x}) \right]. \tag{20}$$

**Step 2.** $(H_N)$ **is a supermartingale.** For well-structured GP models and well-defined smooth functions $\rho^i$, we have: $i)$ $\sigma_f(\mathbf{x} | \mathcal{F}_{N+1}) \leq \sigma_f(\mathbf{x} | \mathcal{F}_N)$ (based on the definition of GP predicted variance), and $ii)$ $\mathbb{P}(f_h(\mathbf{x}) < y_h^*(\mathbf{x}) | \mathcal{F}_N) = 1$ is sufficient for $\mathbb{P}(f_h(\mathbf{x}) < y_h^*(\mathbf{x}) | \mathcal{F}_{N+1}) = 1$ based on the non-increasing property of density estimation on an evaluated solution $\mathbf{x}^N$. Therefore, the following inequality holds:

$$H_N - \mathbb{E}_N[H_{N+1}] = \mathbb{E}_N \left[ M_N^\rho - M_{N+1}^\rho \right] \geq 0, \tag{21}$$

which implies that $(H_N)_{N \in \mathbb{N}}$ is a supermartingale. Consequently, there is $H_N - \mathbb{E}_N[H_{N+1}] \to 0$ as $N \to \infty$, and also

$$\sup_{\tilde{\mathbf{x}} \in \mathcal{X}} \left[ H_N - \mathbb{E}_N[H_{N,\tilde{\mathbf{x}}}] \right] \to 0. \tag{22}$$

**Step 3. The acquisition function of `Lamda+BO` converges to zero almost surely.** Due to the lower bound, according to equation (21), as evaluations of HF objective increase, $\tilde{\varphi}_{\text{pro}}$ tends to converge to $\varphi$. Note that for $N \to \infty$, $M_N^\rho = M_N^0$ as this convergence appears. Additionally, we also have

$$\sup_{\tilde{\mathbf{x}} \in \mathcal{X}} \mathbb{E}_N \left[ M_N^\rho - M_{N,\tilde{\mathbf{x}}}^\rho \right] \geq \sup_{\tilde{\mathbf{x}} \in \mathcal{X}} \mathbb{E}_N \left[ M_N^0 - M_{N,\tilde{\mathbf{x}}}^\rho \right] \geq \sup_{\tilde{\mathbf{x}} \in \mathcal{X}} \tilde{\varphi}_{\text{pro}}(\mathbf{x}) \text{EI}(\tilde{\mathbf{x}} | \mathcal{D}). \tag{23}$$

Therefore, with the same proof as that of Proposition 2.9 (Bect et al., 2019), for $N \to \infty$, equation 22 and equation 23 yield $\tilde{\varphi}_{\text{pro}}(\mathbf{x}) \text{EI}(\tilde{\mathbf{x}} | \mathcal{D}) \to 0$. This completes the proof for the first statement.

The second statement stands according to the global search ability of EI and corresponding dense evaluated solutions in $\mathcal{X}$. We complete the proof by providing the following facts: $i)$ $\tilde{\varphi}_{\mathrm{pro}}(\mathbf{x})\sigma_f(\tilde{\mathbf{x}}) \rightarrow 0$ holds from the first statement; $ii)$ the lower bound $y_h^*$ will be tight when $N \rightarrow \infty$; and $iii)$ $\sigma_f(z|\mathcal{F}_N) \rightarrow 0$ for all sequences accordingly. Based on these facts, the sequence is almost surely dense in $\mathcal{X}$. As a result, $f_{\mathcal{D}}^*$ from any sequence converges to $f_{\mathcal{X}}^*$ almost surely when $N \rightarrow \infty$.

## B. RELATED WORKS

### B.1 MULTI-FIDELITY BAYESIAN OPTIMIZATION

In the realm of MFBO, previous research primarily leverages LF to construct an accurate MF model for guiding the sampling process (Swersky et al., 2013; Poloczek et al., 2017; Kandasamy et al., 2019; Mikkola et al., 2022; Li et al., 2020b). One challenge in these methods is to model performance using data from various fidelities. Solutions include employing Gaussian process regression (GPR) models tailored to each fidelity level (Kandasamy et al., 2019), multi-task GPR models for discrete fidelity levels (Swersky et al., 2013; Poloczek et al., 2017; Mikkola et al., 2022; Li et al., 2020b), and GPR models for a continuous fidelity space (Klein et al., 2017; Kandasamy et al., 2017). While Gaussian processes are commonly used for modelling the surrogate function, Li et al. (Li et al., 2020b; 2021a) have implemented deep neural networks to represent the relationships across different fidelities. In terms of sampling, entropy search methods (Swersky et al., 2013; Poloczek et al., 2017; Takeno et al., 2020) are utilized, which take into account both the information gain and the associated costs of each fidelity level. These techniques enable the sampling of new solutions at either low or high fidelity levels, gradually leading to improved hyperparameters in the HF space.

### B.2 BANDIT BASED METHOD

Bandit-based methods employ LF for identifying promising solutions for HF evaluations (Falkner et al., 2018; Li et al., 2022b; Awad et al., 2021). In pioneering work in the field, Li et al. (2017) introduced Hyperband, a method that improves upon the Successive Halving (SH) algorithm by integrating various early stopping strategies across multiple SH brackets. Each bracket starts with a different number of solutions at varied fidelity levels. However, Hyperband's approach of randomly sampling solutions doesn't utilize previous sample information (Falkner et al., 2018). To enhance this, Falkner et al. (2018) developed BOHB, which combines BO with Hyperband, using the tree Parzen estimator (TPE) (Bergstra et al., 2011) to build surrogate models for each fidelity level. While BOHB is effective, it struggles with discrete dimensions and scaling to high-dimensional problems. Addressing these challenges, Awad et al. (2021) enhanced BOHB with differential evolution for better candidate sampling. Additionally, Li et al. (2021b) developed an MF ensemble model that integrates information from all fidelity levels to more accurately estimate the highest fidelity. Nevertheless, despite their utilization of LF data, these approaches still demand extensive exploration throughout the entire search space.

### B.3 USING PRIORS FOR OPTIMIZATION

These methods concentrate on promising regions to reduce the consumed resources by leveraging prior information. A line of work uses prior information about locations of optimal solutions to accelerate the process of HPO. For instance to reduce evaluations on bad regions, Souza et al. (2021) injected priors about which parts of the input space will yield the best performance into BO's standard probabilistic model to form a pseudo-posterior, which was shown to be more sample-efficient than BO baselines. Further, Li et al. (2020a) incorporated Gaussian distributions of optimal solutions into the posterior distribution of observed data and used Thompson sampling to obtain the next solution. Ramachandran et al. (2020) used the prior distribution of optimal solutions to warp the search space, expanding around high-probability regions of optimal solutions and shrinking around low-probability regions. Truncated normal and gamma distributions were used to form the prior distributions. Additionally, Hvarfner et al. (2022) incorporated prior Gaussian distributions about locations of optimal solutions into the acquisition function, achieving competitive results across a wide range of benchmarks. Mallik et al. (2023) also integrated prior knowledge of optimal hyperparameters to enhance the efficiency of Hyperband. Although using prior distributions of locations of

optimal solutions can accelerate optimization, accessing the prior for a specific task may not always be accessible.

## B.4 TRANSFER SEARCH SPACE

In addition to leveraging experts' prior, this method used information from previous tasks to reduce search space. For instance, Wistuba et al. (2015) pruned the bad regions of search space according to the results from previous tasks. Perrone & Shen (2019) and Li et al. (2022a) utilized previous tasks to design a sub-region of the entire search space for a new task. While these approaches have demonstrated efficiency in using promising regions instead of the entire space, they require the preparation of source tasks and the evaluation of task similarities to effectively select relevant tasks for learning promising regions, which can be challenging (Perrone & Shen, 2019).

## C. BENCHMARKS

### C.1 TABULAR BENCHMARKS

**FCNet**: We utilized benchmarks for FCNet from Yahpo-Gym (Pfisterer et al., 2022), detailed in Table 3. The selected tasks include FCNet-Naval-Propulsion, FCNet-Protein-Structure, FCNet-Slice-Localization, and FCNet-Parkinsons-Telemonitoring.

**NAS-Bench-301**: Our NAS-Bench-301 benchmarks, also sourced from Yahpo-Gym (Pfisterer et al., 2022), focus on CIFAR-10. For hyperparameter details, refer to (Pfisterer et al., 2022).

**NAS-Bench-201**: This benchmark encompasses 6 hyperparameters for neural architecture search. It includes statistics from 15,625 CNN models across three datasets: CIFAR-10-valid, CIFAR-100, and ImageNet16-120 (Eggensperger et al., 2021). We simplify their name as CIFAR-10, CIFAR-100, and ImageNet in this paper.

Table 3: Hyperparameter ranges for FCNet.

| Parameter | Name | Type | Range |
|---|---|---|---|
| Fidelity | epoch | int | $[1, 100]$ |
| Hyperparameter | batch size | int | $[8, 64]$ (log-scale) |
| | initial learning rate | con | $[5e - 04, 0.1]$ (log-scale) |
| | dropout 1 | con | $[0.0, 0.6]$ |
| | dropout 2 | con | $[0.0, 0.6]$ |
| | number of units 1 | int | $[16, 512]$ |
| | number of units 2 | int | $[16, 512]$ |

### C.2 SURROGATE BENCHMARKS

We utilized four problems from the PD1 benchmarks and three from the JAHSBench surrogate benchmarks. Detailed information about these benchmarks is available in (Mallik et al., 2023).

### C.3 RAW PROBLEMS

The hyperparameters for the deep neural networks, LeNet and ResNet-18, are detailed in Table 4.

Table 4: Hyperparameter ranges for LeNet and ResNet-18.

| Parameter | Name | Type | Range |
|---|---|---|---|
| Fidelity | datasize | con | $[0.3, 1]$ |
| Hyperparameter | batch size | int | $[64, 512]$ (log-scale) |
| | initial learning rate | con | $[5e - 3, 0.1]$ (log-scale) |
| | momentum | con | $[0.5, 0.99]$ |
| | weight decay | con | $[1e - 5, 1e - 2]$ |
| | nesterov | cat | $\{$True, False$\}$ |

## C.4 PARAMETERS OF LF PROBLEMS

Table 5 shows parameters of LF problems used in the first phase and the corresponding OVL between high- and low-fidelity problems.

Table 5: Parameters of LF problems used in the first phase and the corresponding OVL between high- and low-fidelity problems.

| Tasks | LF parameter | OVL | Tasks | LF parameter | OVL |
|-------|-------------|-----|-------|-------------|-----|
| MFH3 | epoch=4 | 0.713 | MFH6 | epoch=4 | 0.728 |
| XGBoost-40981 | datasize=0.3 | 0.885 | FCNet-Naval-Propulsion | epoch=4 | 0.901 |
| XGBoost-41146 | datasize=0.3 | 0.933 | FCNet-Protein-Structure | epoch=4 | 0.694 |
| XGBoost-1489 | datasize=0.3 | 0.805 | FCNet-Slice-Localization | epoch=4 | 0.731 |
| XGBoost-1067 | datasize=0.3 | 0.883 | FCNet-Parkinsons-Telemonitoring | epoch=4 | 0.854 |
| rpart-40981 | datasize=0.3 | 0.167 | NAS-Bench-301-CIFAR-10 | epoch=20 | 0.712 |
| rpart-41146 | datasize=0.3 | 0.488 | NAS-Bench-201-CIFAR-10 | epoch=20 | 0.760 |
| rpart-1089 | datasize=0.3 | 0.89 | NAS-Bench-201-CIFAR-100 | epoch=20 | 0.727 |
| rpart-1067 | datasize=0.3 | 0.642 | NAS-Bench-201-ImageNet | epoch=20 | 0.696 |
| ranger-40981 | datasize=0.3 | 0.711 | JAHS-CIFAR-10 | epoch=4 | 0.574 |
| ranger-41146 | datasize=0.3 | 0.782 | JAHS-Colorectal-Histology | epoch=4 | 0.523 |
| ranger-1489 | datasize=0.3 | 0.77 | JAHS-Fashion-MNIST | epoch=4 | 0.532 |
| ranger-1067 | datasize=0.3 | NAN | PD1-LM1B | epoch=30 | 0.934 |
| glmnet-40981 | datasize=0.3 | 0.294 | PD1-WMT | epoch=4 | 0.926 |
| glmnet-41146 | datasize=0.3 | 0.962 | PD1-CIFAR-100 | epoch=45 | 0.797 |
| glmnet-1489 | datasize=0.3 | 0.918 | PD1-ImageNet | epoch=20 | 0.727 |
| glmnet-1067 | datasize=0.3 | 0.648 | NAN | NAN | NAN |

## D. OVL BETWEEN LOW AND HIGH FIDELITY

Figure 11 presents the overlapping coefficients between good solutions in high- and low-fidelity settings across various HPO tasks. The overlapping coefficients are computed using equation (3) and are visualized over epochs or datasets for different benchmarks. It can be observed that the overlap generally increases with the number of iterations (e.g., epochs or dataset size), especially for FCNet, NAS-Bench-201, and RecNet-18. However, for NAS-Bench-301 and JAHS, the overlap exhibits a trend of decreasing in the middle stages before rising again. Specifically:

- FCNet and RecNet-18 show an overlap that is already close to or greater than $0.7$ even at smaller iteration parameters (e.g., fewer epochs or smaller datasets), indicating strong LF-HF consistency and stability at earlier stages.

- For PD1 and NAS-Bench-201, the overlap reaches or exceeds $0.7$ at specific points, such as epoch = 10 for PD1 and epoch = 20 for NAS-Bench-201, suggesting that these tasks achieve good LF-HF agreement relatively early in the optimization process.

- NAS-Bench-301 and JAHS, on the other hand, maintain relatively low overlap in the initial and middle stages. The overlap only increases significantly as the settings approach the HF level, indicating that these tasks require longer training or higher fidelity to achieve substantial LF-HF alignment.

## E. DISTRIBUTION OF SOLUTIONS IN THE HF AND LF.

In this section, we present the distribution of solutions in the HF and LF settings. Figure 12 and Figure 13 shows good solutions identified by `Lamda+BOHB` at the first phase for FCNet, NAS-Bench-301 and JAHS. It can be observed that there is an overlap between the good solutions in the HF and LF settings, and the good solutions found in the first phase are close to these overlapping regions.

Moreover, Figure 14, Figure 15, Figure 16 and Figure 17 show the distribution of good solutions in the HF and LF. Across all $15$ problems, a clear overlap is observed between the good solutions identified in both fidelity levels.

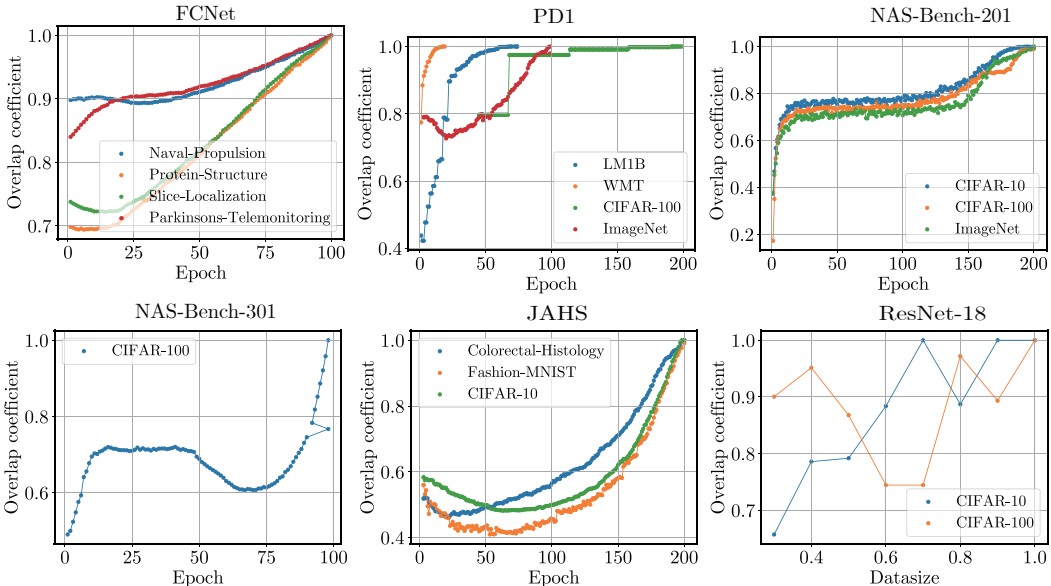

Figure 11: The overlapping coefficient of the HPO tasks.

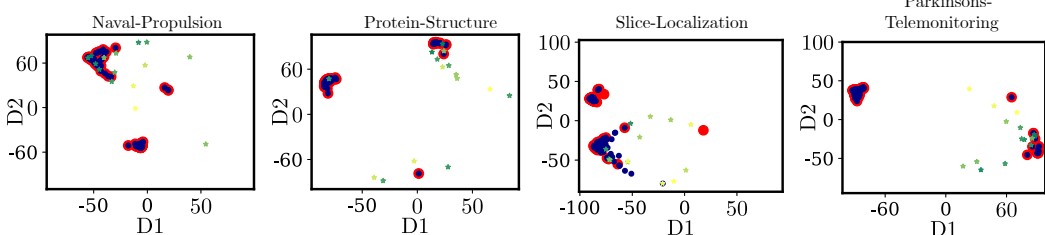

Figure 12: Visualization of good solutions identified by `Lamda+BOHB` at the first stage for FCNet. The solutions were mapped to a 2D representation for clarity. The red and blue points are the real top 10% high and low fidelity solutions from 10,000 sampling points. Stars are obtained by `Lamda+BOHB`, with a more yellow color indicating later sampling.

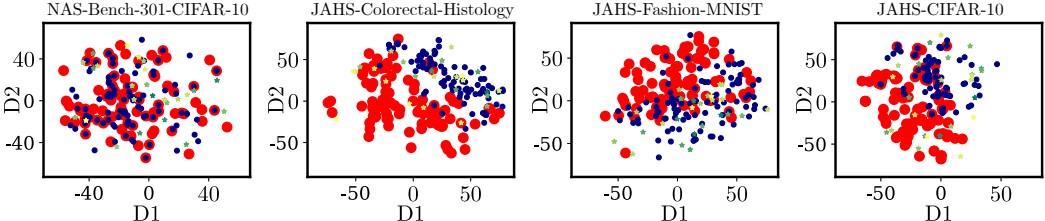

Figure 13: Visualization of good solutions identified by `Lamda+BOHB` at the first phase for NAS-Bench-301 and JAHS. The solutions were mapped to 2D. The red and blue points are the real top 10% high and low fidelity solutions from 10,000 sampling points. Stars are obtained by `Lamda+BOHB`, with a more yellow color indicating later sampling.

## F. EXPERIMENTAL RESULTS

### F.1 CONVERGENCE CURVES

This section presents the convergence curves of the proposed algorithm and its peer methods. The convergence performance of `Lamda` under different algorithms, including `BOHB`, `MUMBO`, `BO`, and `RS`, is illustrated in Figure 18, Figure 19, Figure 20, Figure 21, Figure 22, Figure 23, Figure 24 and Figure 25. The experimental results demonstrated that using `Lamda` can enhance peformanc

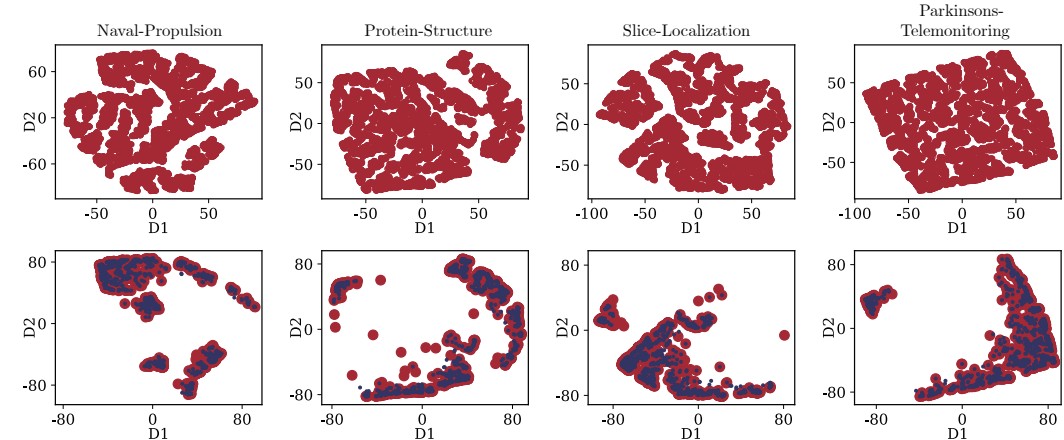

Figure 14: (Top) The visualization of 10,000 hyperparameter configurations in the 2D space for FCNet. (Bottom) The visualization of the top 10% solutions at the high and low fidelity, with high fidelity solutions marked in red. In this case, LF is obtained by setting the epoch number as four.

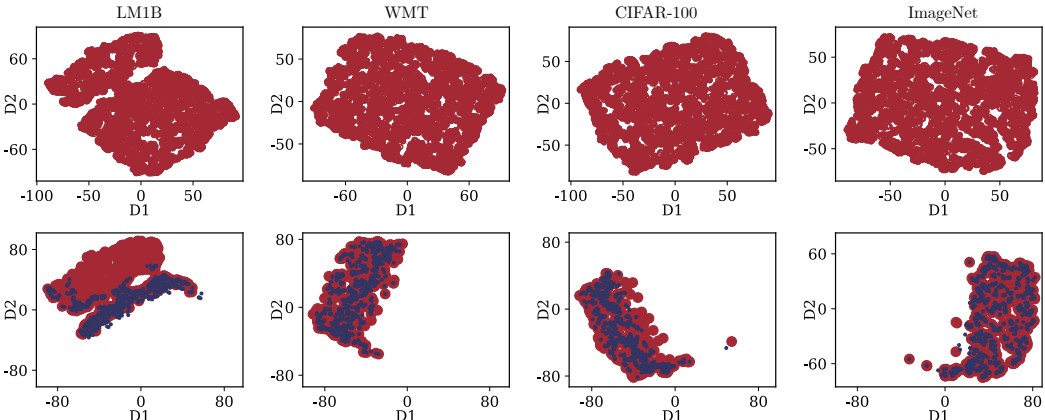

Figure 15: (Top) The visualization of the 10,000 hyperparameter configurations in the 2D space for PD1. (Bottom) The visualization of the top 10% solutions at the high and low fidelity, with high fidelity solutions marked in red. In this case, LF is obtained by setting the epoch number as 30, 4, 45, and 20 for the four tasks.

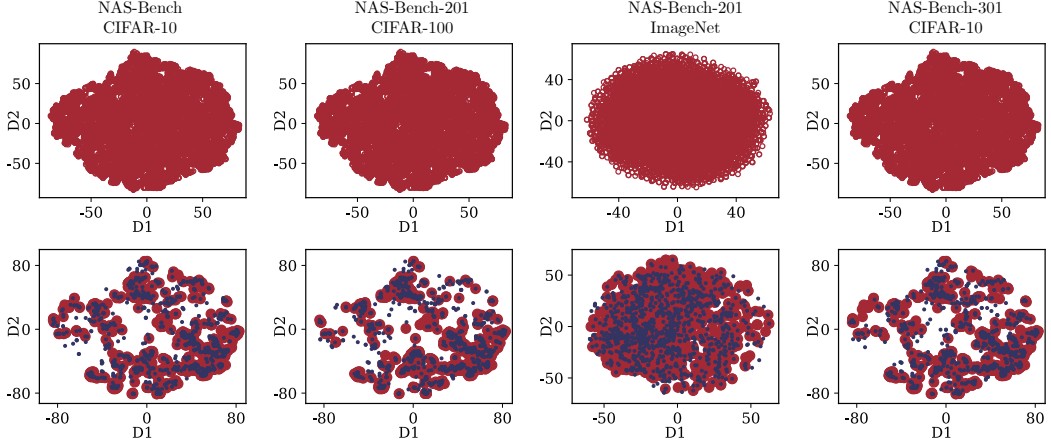

Figure 16: (Top) The visualization of 10,000 hyperparameter configurations in the 2D space for NAS-Bench-201 and NAS-Bench-301. (Bottom) The visualization of the top 10% solutions at the high and low fidelity, with high fidelity solutions marked in red. In this case, LF is obtained by setting the epoch number as 20.

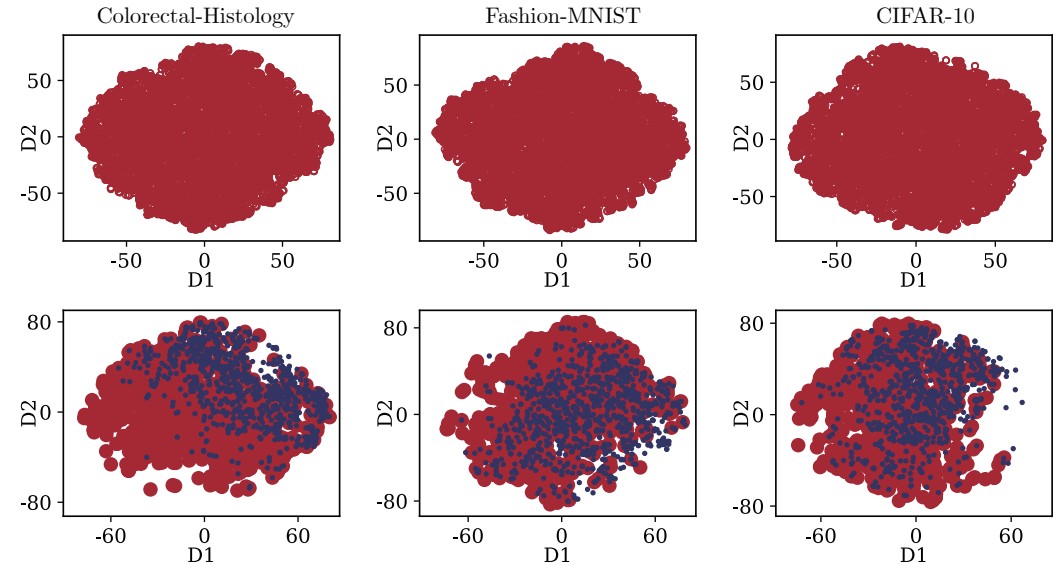

Figure 17: (Top) The visualization of $10,000$ hyperparameter configurations in the 2D space for JASH. (Bottom) The visualization of the top 10% solutions at the high and low fidelity, with high fidelity solutions marked in red. In this case, LF is obtained by setting the epoch number as 20.

of the baseline algorithms. The convergence curves of peer algorithms are presented in Figure 26 and Figure 27. `Lamda` also performs good compared with peer algorithms.

Figure 28, Figure 29, Figure 30, Figure 31 and Figure 32 depict the results of parameter analysis including budget ($B$), thresholds ($\gamma$) for stopping the first phase, the interval for calculating the overlapping coefficient ($\Delta$), the quantile ($\alpha$) used in the TPE, and the initial weight ($w_0$). These results indicate that the algorithm's performance is robust to the hyperparameter settings.

### F.2 TABLE RESULTS OF PEER ALGORITHMS

Table 6 shows the peer algorithms' final validation errors of the current incumbent at 100 HF evaluations horizons.

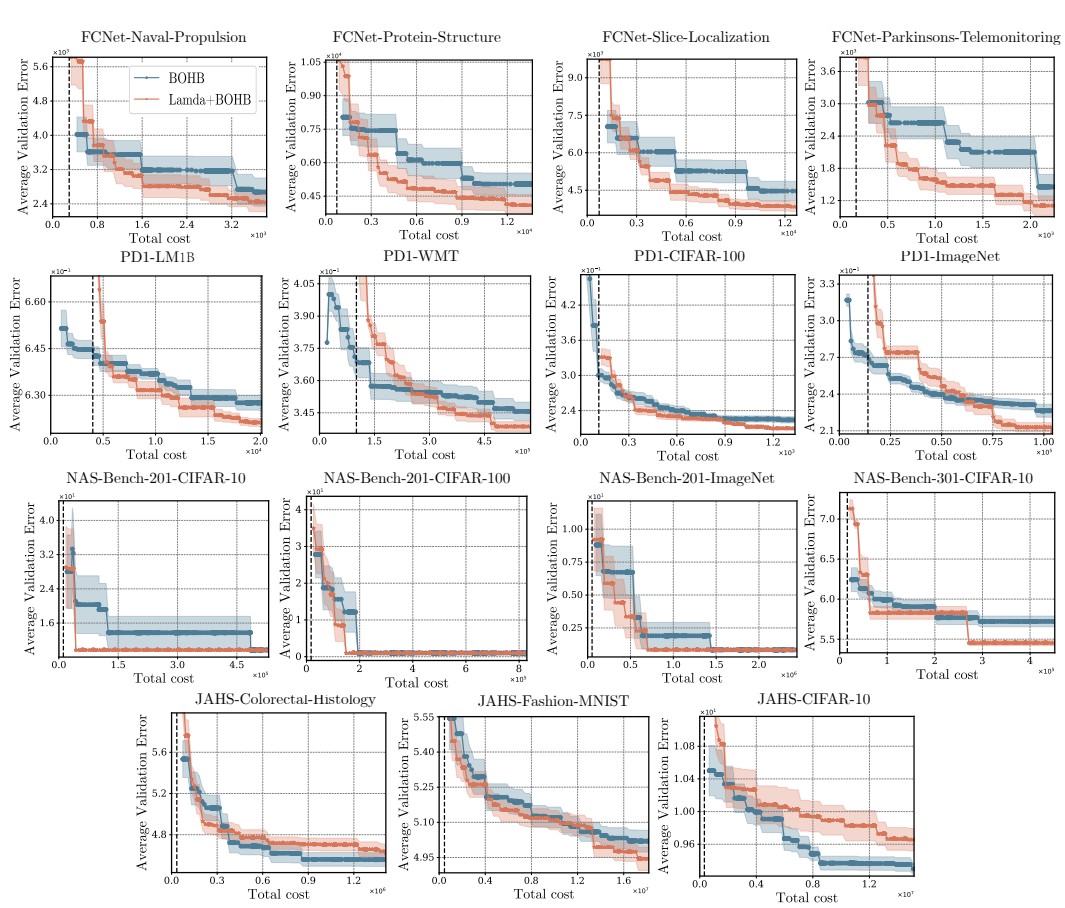

Figure 18: Validation error observed in tuning 15 HPO tasks, using BOHB as the baseline.

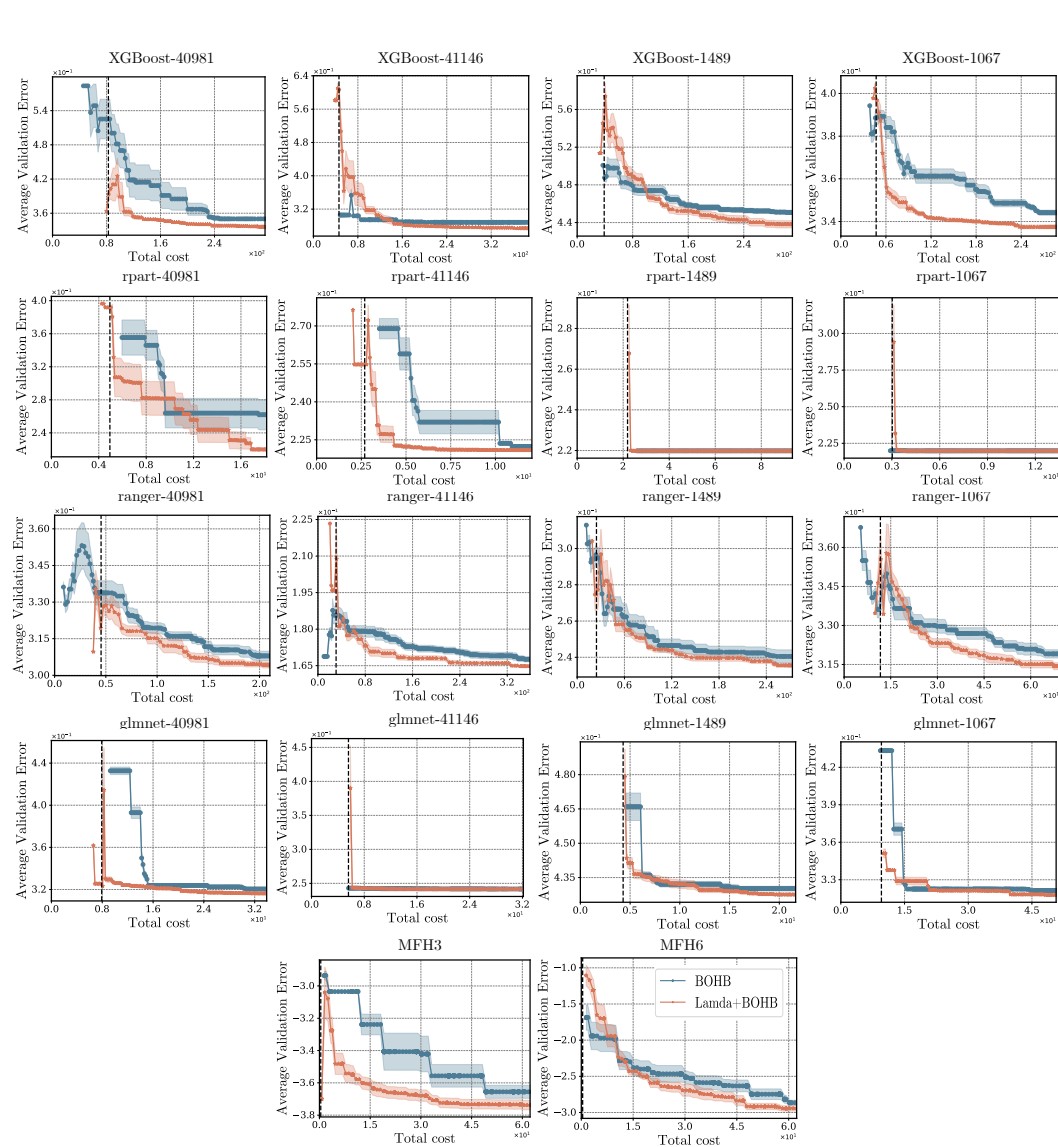

Figure 19: Validation error observed in tuning 18 HPO tasks, using BOHB as the baseline.

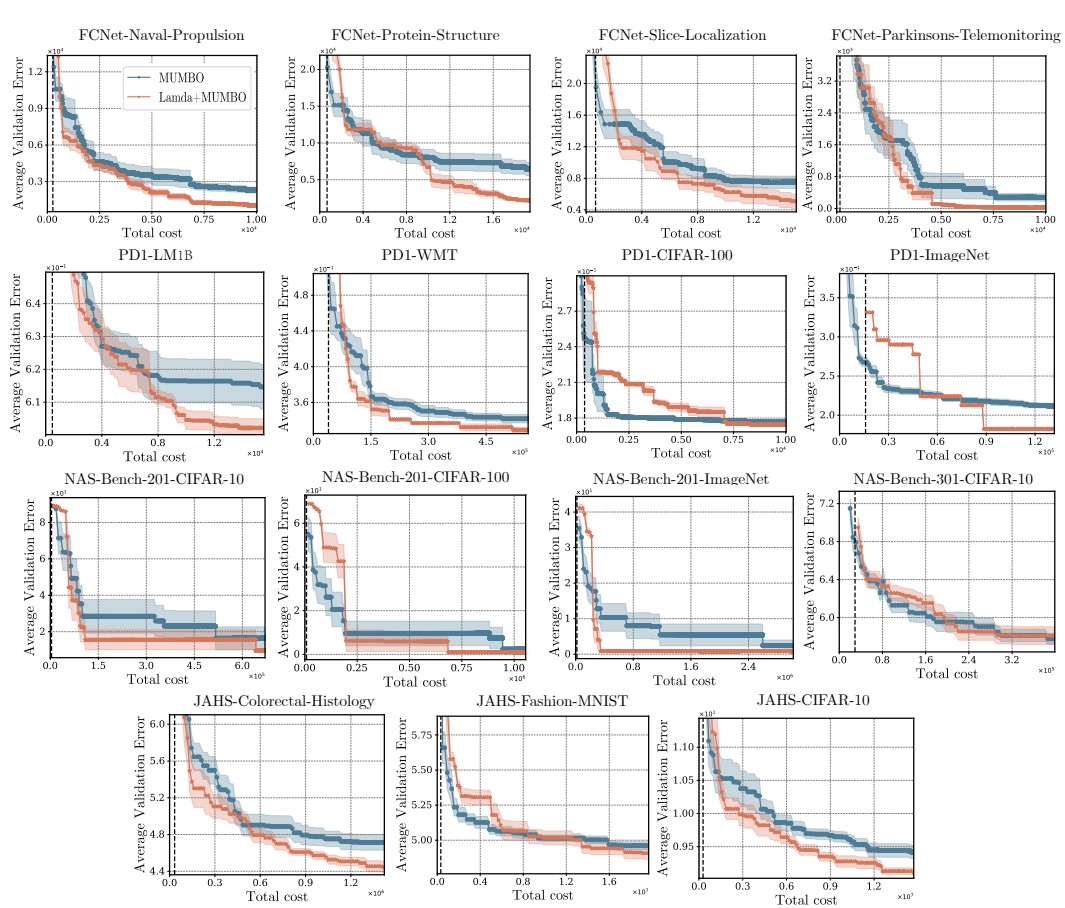

Figure 20: Validation error observed in tuning 15 HPO tasks, using MUMBO as the baseline.

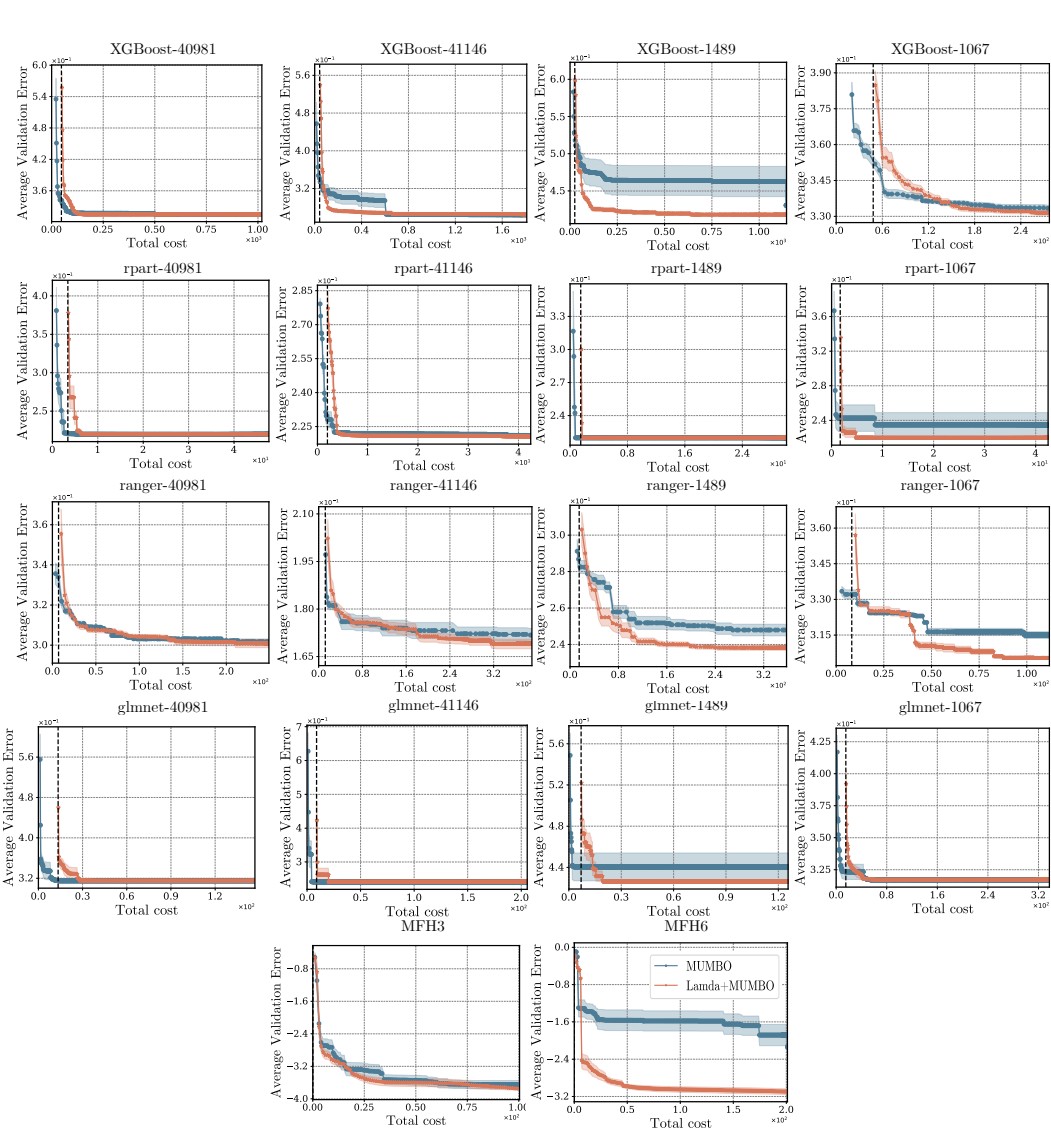

Figure 21: Validation error observed in tuning 18 HPO tasks, using MUMBO as the baseline.

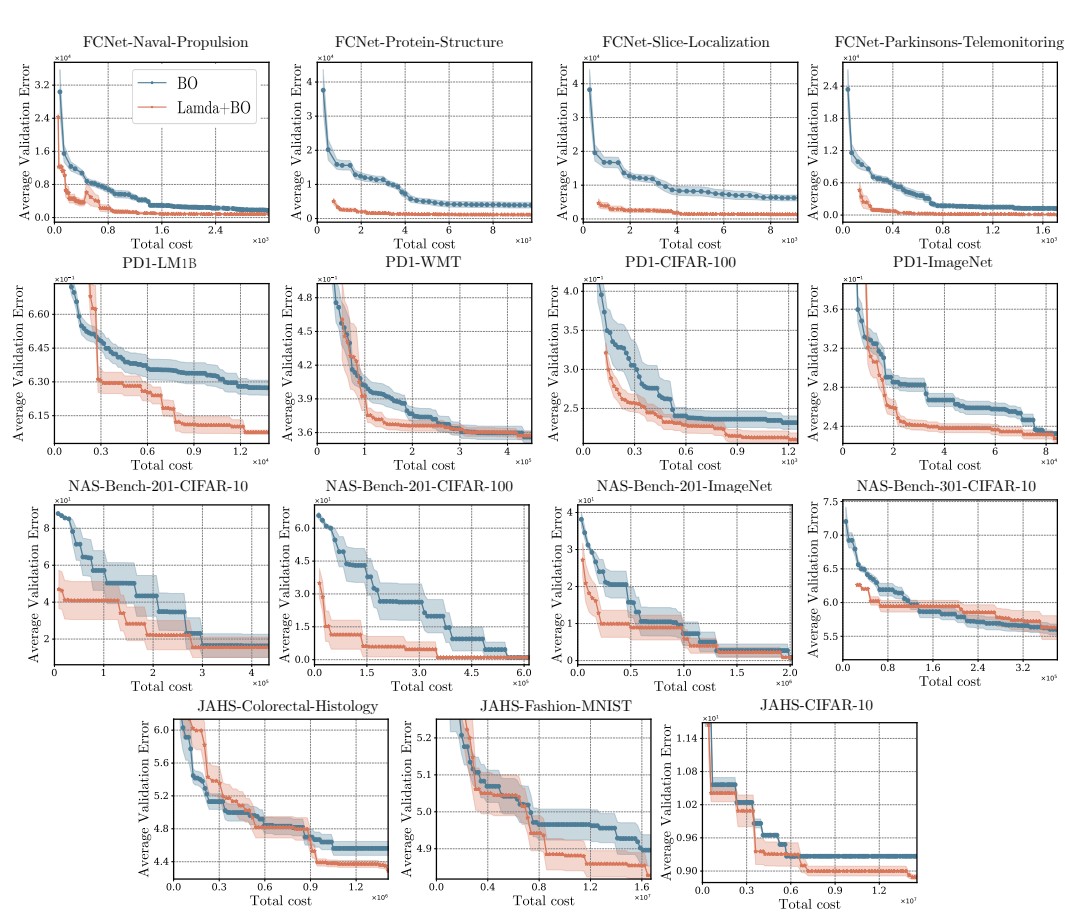

Figure 22: Validation error observed in tuning 15 HPO tasks, using `BO` as the baseline.

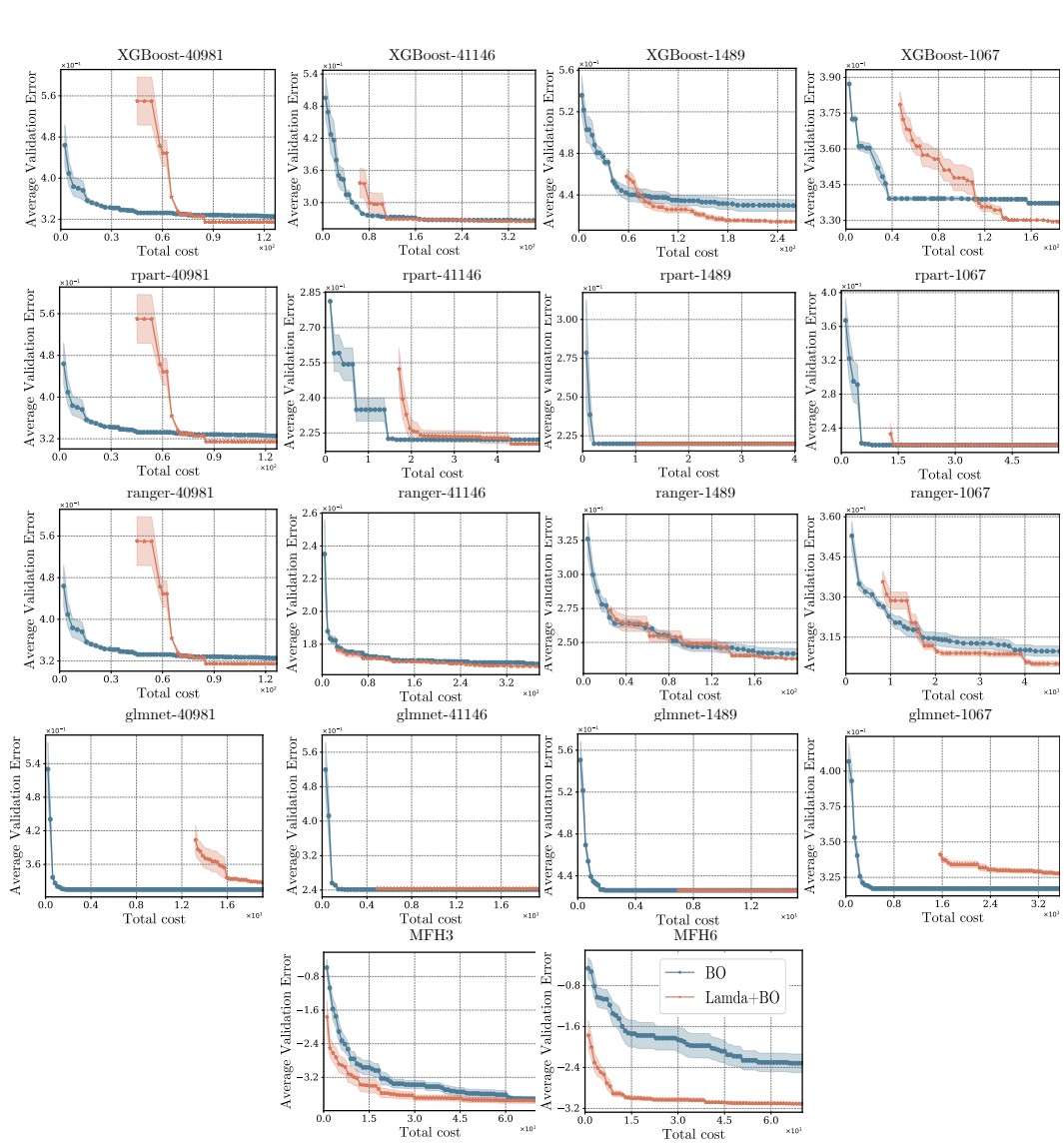

Figure 23: Validation error observed in tuning 18 HPO tasks, using BO as the baseline.
.

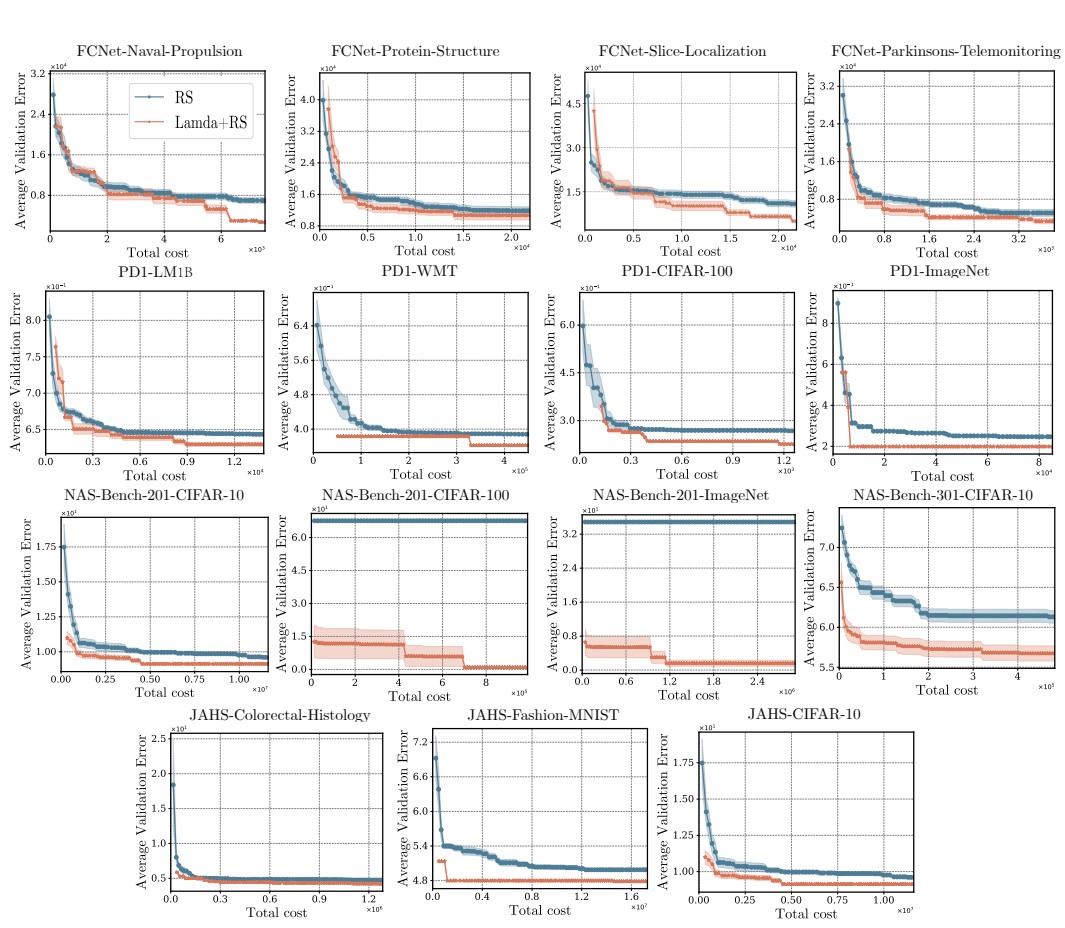

Figure 24: Validation error observed in tuning 15 HPO tasks, using RS as the baseline.
.

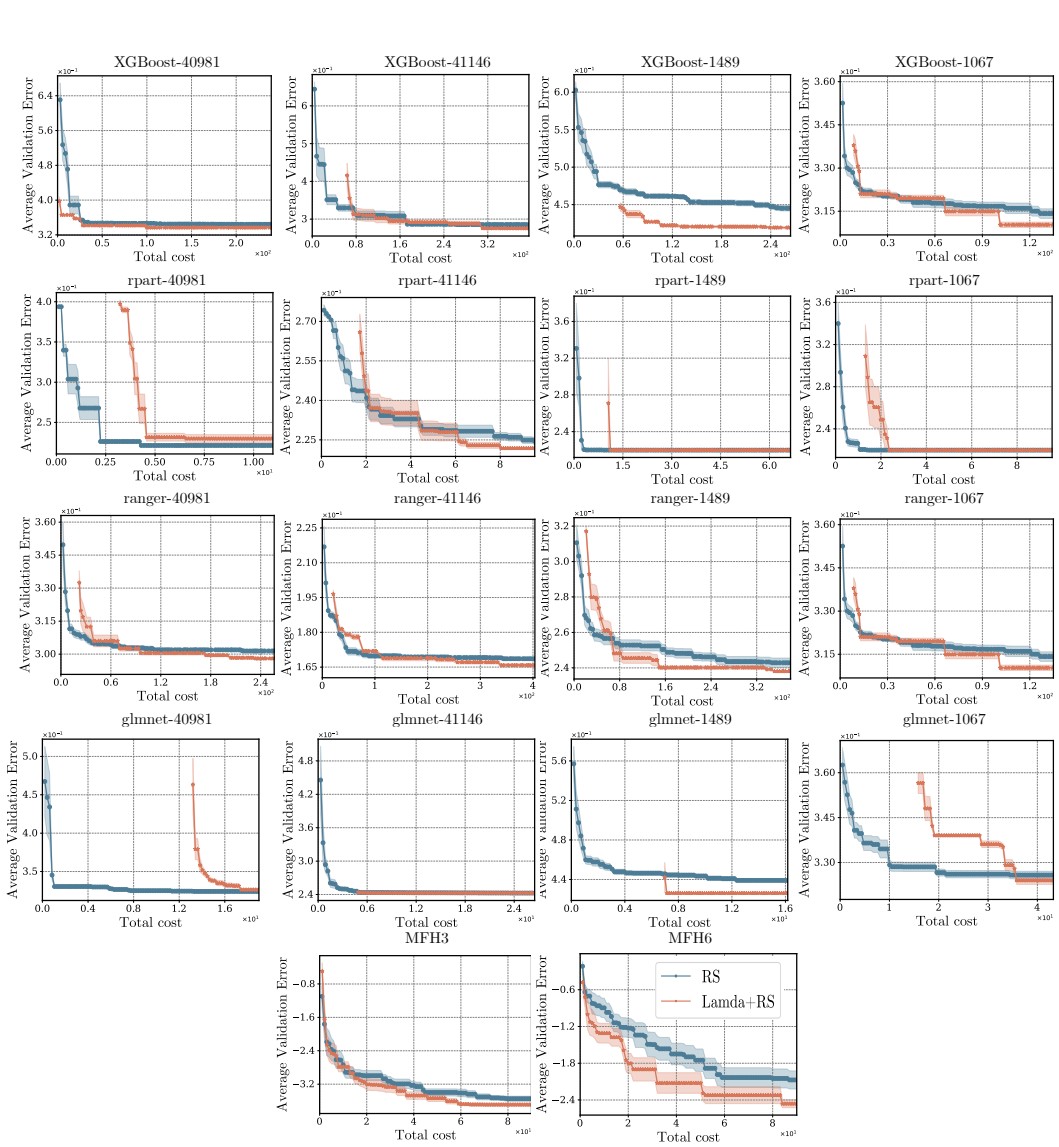

Figure 25: Validation error observed in tuning 18 HPO tasks, using RS as the baseline.
.

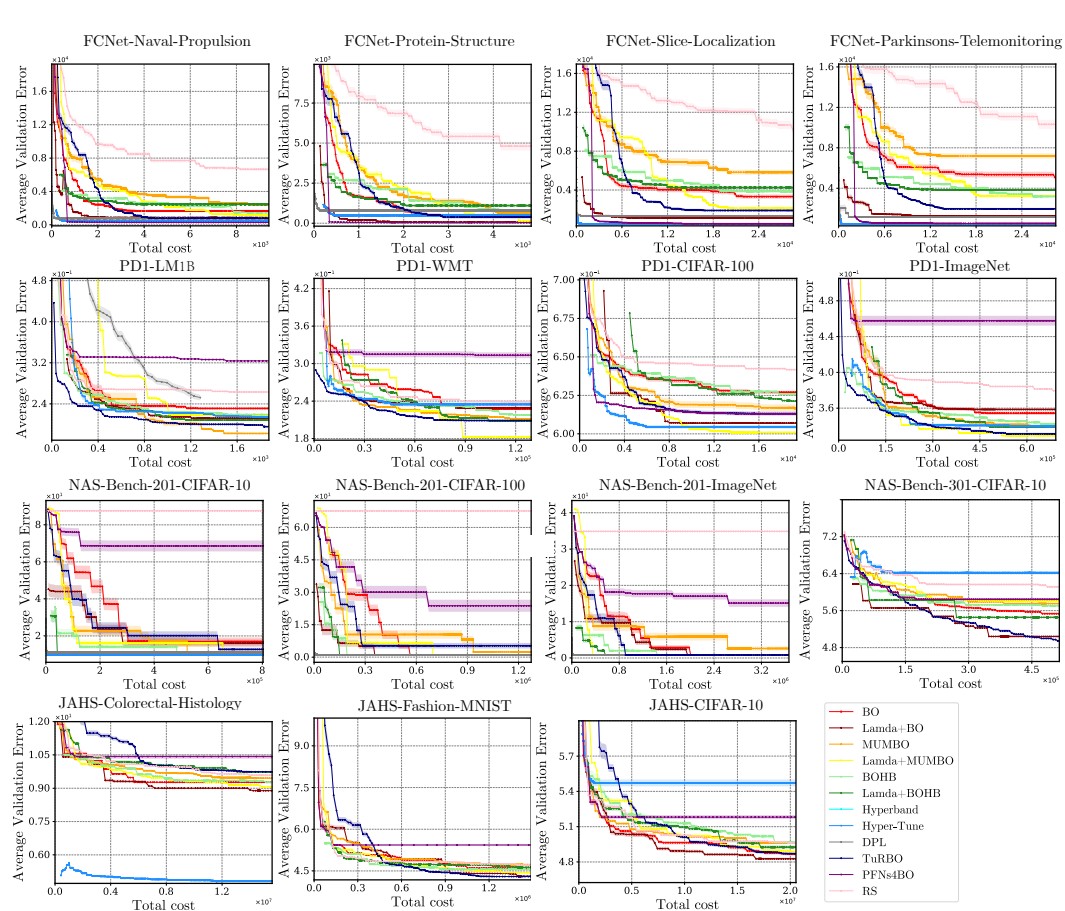

Figure 26: Validation error observed in tuning 15 HPO tasks, comparing peer algorithms.
.

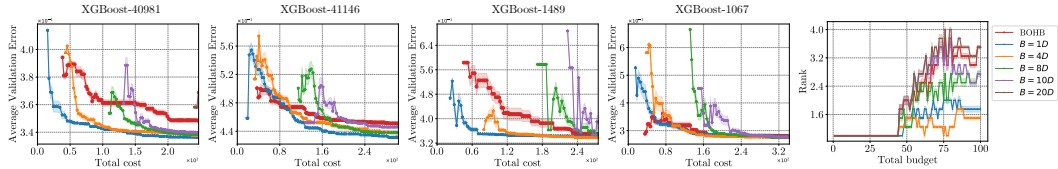

Figure 27: Validation error observed in tuning 18 HPO tasks, comparing peer algorithms.

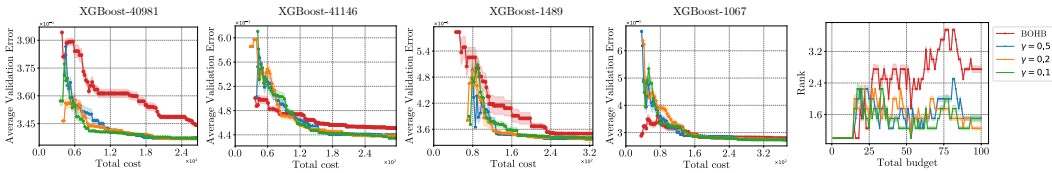

Figure 28: Validation errors of `Lamda+BOHB` under different parameters $B$ used at the first phase.

Figure 29: Validation error observed of `Lamda+BOHB` under different parameter $\gamma$ at the first phase.

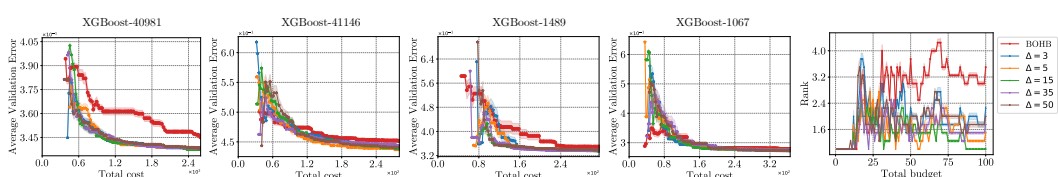

Figure 30: Validation error observed of `Lamda+BOHB` under different parameter $\Delta$ at the first phase.

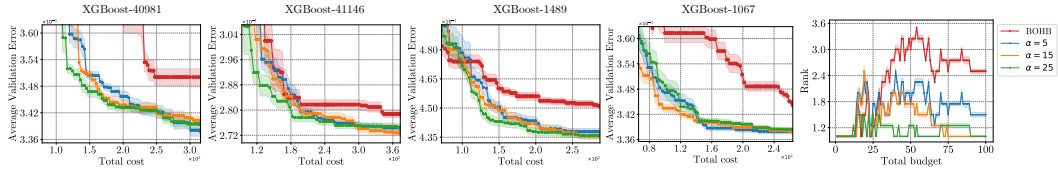

Figure 31: Validation error observed of `Lamda+BOHB` under different parameter $\alpha$ at the first phase.

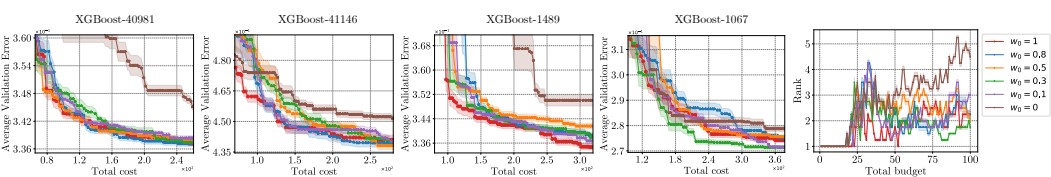

Figure 32: Validation error observed of `Lamda+BOHB` under different parameter $w$ at the first phase.

Table 6: Comparing peer algorithms' final validation errors of the current incumbent at 100 HF evaluations horizons. Runs are averaged over 31 seeds. (The bolded part indicates: "under the Wilcoxon rank-sum test, methods incorporating priors significantly outperform their baselines.")

| Tasks | BOHB | Lamda+BOHB | MUMBO | Lamda+MUMBO | BO | Lamda+BO | RS | Lamda+RS | PriorBand | Lamda+PriorBand |
|---|---|---|---|---|---|---|---|---|---|---|
| MFH3 | -3.655e+00(1.339e-01) | -3.716e+00(1.056e-01) | -3.860e+00(3.569e-01) | -3.846e+00(2.174e-01) | -3.822e+00(2.172e-01) | -3.843e+00(5.508e-02) | -3.572e+00(2.529e-01) | -3.718e+00(1.323e-01) | -3.857e+00(3.553e-01) | -3.852e+00(2.401e-03) |
| MFH6 | -2.864e+00(1.364e-01) | -2.935e+00(1.427e-01) | -1.872e+00(0) | -3.178e+00(2.752e-01) | -2.138e+00(9.286e-01) | -3.112e+00(8.129e-02) | -2.260e+00(6.577e-01) | -2.396e+00(2.627e-01) | -3.187e+00(1.506e-01) | -3.167e+00(5.198e-02) |
| XGBoost-41981 | 3.456e-01(9.817e-03) | 3.337e-01(9.490e-03) | 3.147e-01(2.235e-07) | 3.147e-01(2.235e-07) | 3.337e-01(4.953e-03) | 3.147e-01(1.028e-05) | 3.442e-01(1.143e-03) | 3.367e-01(1.528e-03) | NA | NA |
| XGBoost-41146 | 2.883e-01(1.238e-03) | 2.783e-01(6.558e-03) | 2.634e-01(1.694e-03) | 2.635e-01(1.063e-03) | 2.660e-01(1.620e-03) | 2.646e-01(1.823e-03) | 2.824e-01(8.990e-03) | 2.768e-01(4.922e-04) | NA | NA |
| XGBoost-1489 | 4.477e-01(8.205e-03) | 4.330e-01(2.073e-02) | 4.616e-01(5.269e-02) | 4.116e-01(1.450e-02) | 4.164e-01(1.462e-02) | 4.116e-01(5.029e-03) | 4.415e-01(1.642e-02) | 4.187e-01(5.514e-03) | NA | NA |
| XGBoost-1067 | 3.453e-01(5.331e-03) | 3.377e-01(3.775e-03) | 3.345e-01(5.656e-03) | 3.301e-01(5.028e-03) | 3.357e-01(7.962e-04) | 3.298e-01(1.182e-04) | 3.399e-01(0) | 3.345e-01(0) | NA | NA |
| rpart-41981 | 2.200e-01(6.758e-02) | 2.199e-01(8.859e-06) | 2.199e-01(7.525e-06) | 2.199e-01(1.937e-06) | 2.199e-01(3.759e-06) | 2.200e-01(2.866e-04) | 2.213e-01(8.945e-04) | 2.199e-01(1.214e-05) | NA | NA |
| rpart-41146 | 2.222e-01(1.529e-03) | 2.210e-01(9.219e-04) | 2.210e-01(4.312e-04) | 2.205e-01(3.331e-04) | 2.210e-01(1.799e-03) | 2.204e-01(1.356e-04) | 2.233e-01(3.626e-03) | 2.216e-01(1.541e-03) | NA | NA |
| rpart-1489 | 2.198e-01(0) | 2.198e-01(0) | 2.198e-01(0) | 2.198e-01(0) | 2.198e-01(0) | 2.198e-01(0) | 2.198e-01(0) | 2.198e-01(0) | NA | NA |
| rpart-1067 | 2.198e-01(0) | 2.198e-01(0) | 2.388e-01(0) | 2.198e-01(0) | 2.198e-01(0) | 2.198e-01(0) | 2.198e-01(0) | 2.198e-01(0) | NA | NA |
| ranger-41981 | 3.087e-01(4.186e-03) | 3.038e-01(6.079e-03) | 2.993e-01(2.547e-03) | 3.022e-01(3.181e-03) | 3.024e-01(7.710e-03) | 2.835e-01(1.074e-02) | 3.019e-01(1.860e-03) | 2.967e-01(2.828e-03) | NA | NA |
| ranger-41146 | 1.645e-01(6.803e-03) | 1.645e-01(2.341e-03) | 1.683e-01(2.645e-03) | 1.679e-01(4.282e-03) | 1.670e-01(3.795e-03) | 1.673e-01(1.883e-03) | 1.685e-01(1.981e-03) | 1.641e-01(2.814e-03) | NA | NA |
| ranger-1489 | 2.328e-01(5.845e-03) | 2.358e-01(2.925e-03) | 2.440e-01(1.350e-03) | 2.372e-01(2.984e-03) | 2.336e-01(2.404e-03) | 2.316e-01(1.231e-03) | 2.464e-01(4.325e-03) | 2.381e-01(1.180e-03) | NA | NA |
| ranger-1067 | 3.182e-01(6.926e-03) | 3.156e-01(6.902e-03) | 3.138e-01(5.727e-03) | 3.055e-01(5.693e-04) | 3.142e-01(5.990e-03) | 3.061e-01(5.558e-03) | 3.180e-01(2.325e-03) | 3.089e-01(3.449e-03) | NA | NA |
| glmnet-41981 | 3.200e-01(2.952e-03) | 3.161e-01(4.786e-04) | 3.149e-01(0) | 3.149e-01(0) | 3.149e-01(0) | 3.269e-01(6.331e-03) | 3.241e-01(4.395e-04) | 3.247e-01(1.148e-03) | NA | NA |
| glmnet-41146 | 2.415e-01(3.053e-04) | 2.413e-01(5.809e-04) | 2.411e-01(0) | 2.411e-01(0) | 2.411e-01(0) | 2.430e-01(3.797e-04) | 2.425e-01(4.839e-04) | 2.429e-01(5.945e-04) | NA | NA |
| glmnet-1489 | 4.290e-01(2.869e-03) | 4.274e-01(5.247e-04) | 4.392e-01(0) | 4.262e-01(0) | 4.262e-01(0) | 4.262e-01(0) | 4.392e-01(6.759e-03) | 4.262e-01(0) | NA | NA |
| glmnet-1067 | 3.214e-01(4.097e-04) | 3.178e-01(7.455e-05) | 3.171e-01(0) | 3.171e-01(0) | 3.171e-01(0) | 3.282e-01(6.786e-03) | 3.232e-01(5.589e-03) | 3.210e-01(4.693e-03) | NA | NA |
| FCNet-Naval-Propulsion | 2.537e+03(1.014e+03) | 2.411e+03(6.557e+02) | 2.458e+03(7.802e+02) | 1.002e+03(5.606e+02) | 2.627e+02(2.871e+02) | 2.121e+02(2.884e+02) | 6.766e+03(6.833e+02) | 3.370e+03(1.043e+03) | 4.461e+03(8.544e+02) | 3.956e+03(4.427e+02) |
| FCNet-Protein-Structure | 5.030e+03(2.183e+03) | 4.334e+03(2.098e+03) | 6.529e+03(4.761e+03) | 2.015e+03(2.819e+03) | 2.764e+03(3.452e+03) | 1.156e+03(1.424e+02) | 1.367e+04(6.222e+03) | 1.097e+04(5.827e+03) | 6.273e+03(4.760e+03) | 3.501e+03(1.508e+02) |
| FCNet-Slice-Localization | 4.418e+03(5.756e+02) | 4.006e+03(3.853e+02) | 8.197e+03(4.281e+03) | 3.475e+03(2.819e+03) | 4.825e+03(5.299e+03) | 1.378e+03(1.161e+02) | 1.368e+04(1.960e+03) | 5.048e+03(3.287e+03) | 5.275e+03(1.709e+03) | 4.269e+03(5.377e+03) |
| FCNet-Parkinsons-Telemonitoring | 1.580e+04(4.104e+02) | 1.139e+03(7.311e+02) | 1.364e+02(2.477e+02) | 1.197e+01(9.573e+00) | 4.236e+02(7.323e+02) | 2.623e-05(6.821e+01) | 5.333e+03(5.352e+03) | 4.275e+03(3.180e+03) | 3.997e+02(8.256e+02) | 4.041e+02(1.419e+03) |
| NAS-Bench-301-CIFAR-10 | 5.760e+00(2.620e-01) | 5.454e+00(1.091e-01) | 5.761e+00(1.368e-01) | 5.787e+00(3.964e-01) | 5.492e+00(2.995e-01) | 5.629e+00(5.881e-01) | 6.201e+00(4.134e-01) | 5.538e+00(4.347e-01) | NA | NA |
| NAS-Bench-201-CIFAR-10 | 9.712e+00(8.138e-10) | 9.712e+00(2.543e-10) | 9.792e+00(6.103e-10) | 9.712e+00(1.272e-10) | 9.712e+00(1.424e-09) | 9.712e+00(9.664e-10) | 8.760e+01(0) | 9.712e+00(1.017e-09) | NA | NA |
| NAS-Bench-201-CIFAR-100 | 1.000e+00(0) | 1.000e+00(0) | 1.000e+00(0) | 1.000e+00(0) | 1.000e+00(0) | 1.000e+00(0) | 6.764e+01(0) | 1.000e+00(0) | NA | NA |
| NAS-Bench-201-ImageNet | 8.333e-01(3.709e-10) | 8.333e-01(2.914e-10) | 8.333e-01(1.854e-10) | 8.333e-01(3.444e-10) | 8.333e-01(2.384e-10) | 8.333e-01(3.444e-10) | 3.487e+01(0) | 8.333e-01(4.239e-10) | NA | NA |
| JAHS-CIFAR-10 | 9.252e+00(1.828e-01) | 9.495e+00(5.682e-01) | 9.340e+00(5.540e-01) | 9.002e+00(1.398e-01) | 9.268e+00(1.037e-01) | 9.118e+00(2.973e-01) | 9.536e+00(3.553e-01) | 9.201e+00(1.504e-01) | NA | NA |
| JAHS-Colorectal-Histology | 4.467e+00(3.072e-01) | 4.542e+00(1.801e-01) | 4.596e+00(5.287e-01) | 4.451e+00(2.517e-01) | 4.576e+00(2.848e-01) | 4.288e+00(5.159e-02) | 4.940e+00(3.295e-01) | 4.209e+00(5.453e-02) | NA | NA |
| JAHS-Fashion-MNIST | 4.966e+00(9.998e-02) | 4.960e+00(2.068e-01) | 5.006e+00(1.478e-01) | 4.904e+00(2.414e-01) | 4.850e+00(1.387e-01) | 4.864e+00(8.280e-02) | 4.980e+00(0) | 4.788e+00(0) | NA | NA |
| PD1-LM1B | 6.255e-01(6.187e-03) | 6.209e-01(4.692e-03) | 6.092e-01(3.626e-02) | 6.042e-01(1.818e-02) | 6.277e-01(1.818e-02) | 6.063e-01(2.018e-04) | 6.459e-01(1.125e-02) | 6.315e-01(9.575e-03) | 9.905e+00(0) | 9.905e+00(3.879e-03) |
| PD1-WMT | 3.438e-01(1.379e-02) | 3.394e-01(1.243e-02) | 3.357e-01(4.336e-03) | 3.326e-01(8.931e-03) | 3.578e-01(1.671e-02) | 3.579e-01(2.453e-02) | 3.762e-01(0) | 3.624e-01(0) | 9.907e+01(2.832e-03) | 9.906e+01(0) |
| PD1-CIFAR-100 | 2.141e-01(3.115e-02) | 2.097e-01(1.186e-02) | 1.767e-01(2.363e-03) | 1.743e-01(4.328e-04) | 2.345e-01(4.867e-02) | 1.946e-01(4.658e-02) | 2.722e-01(5.905e-03) | 2.269e-01(8.842e-03) | NA | NA |
| PD1-ImageNet | 2.282e-01(3.206e-02) | 2.050e-01(1.184e-02) | 1.942e-01(0) | 1.824e-01(0) | 2.263e-01(1.753e-03) | 2.145e-01(1.641e-02) | 2.360e-01(0) | 1.986e-01(0) | NA | NA |

### F.3 ANALYZING THE IMPACT OF PROMISING REGIONS OVERLAP IN LF AND HF LANDSCAPES ON ALGORITHM PERFORMANCE

To better illustrate the solution distributions across low- and high-fidelity problems, we visualized the distributions in a 2D space. We sampled $10,000$ hyperparameter configurations and computed their fitness values under both low- and high-fidelity settings. The data was then compressed into 2D using the UMAP McInnes & Healy (2018) algorithm. To further enhance interpretability, we applied linear interpolation to generate a continuous landscape surface. Figure 33, Figure 34, Figure 35, and Figure 36 present the landscapes of FCNet, PD1, NAS-Bench-201, NAS-Bench-301, and JAHS under both high- and low-fidelity settings. The overlap between high- and low-fidelity problems is also shown in the figures.

From these visualizations, we observe that FCNet and PD1 exhibit a high overlap (ranging from 0.7 to 0.9) in their promising regions across fidelity levels, with the promising regions being relatively concentrated. In contrast, while NAS-Bench-201 and NAS-Bench-301 have moderate OVL values (ranging from 0.67 to 0.76), their solution distributions are more dispersed. JAHS demonstrates lower OVL values (ranging from 0.52 to 0.57) and also shows a more scattered solution distribution.

The results indicate that for problems where the promising regions of LF and HF landscape have varying levels of overlaps, our method (`Lamda`) achieves better performance than the original algorithms under the frameworks of `BOHB`, `MUMBO`, `BO`, and `RS`, as illustrated in the convergence curves (Figure 18, Figure 20, Figure 22, and Figure 24). In particular, we would like highlight the results on JAHS-CIFAR-10, JAHS-Colorectal-Histology, and JAHS-Fashion-MNIST, whose overlaps are low. `Lamda` consistently enhance the baseline algorithms.

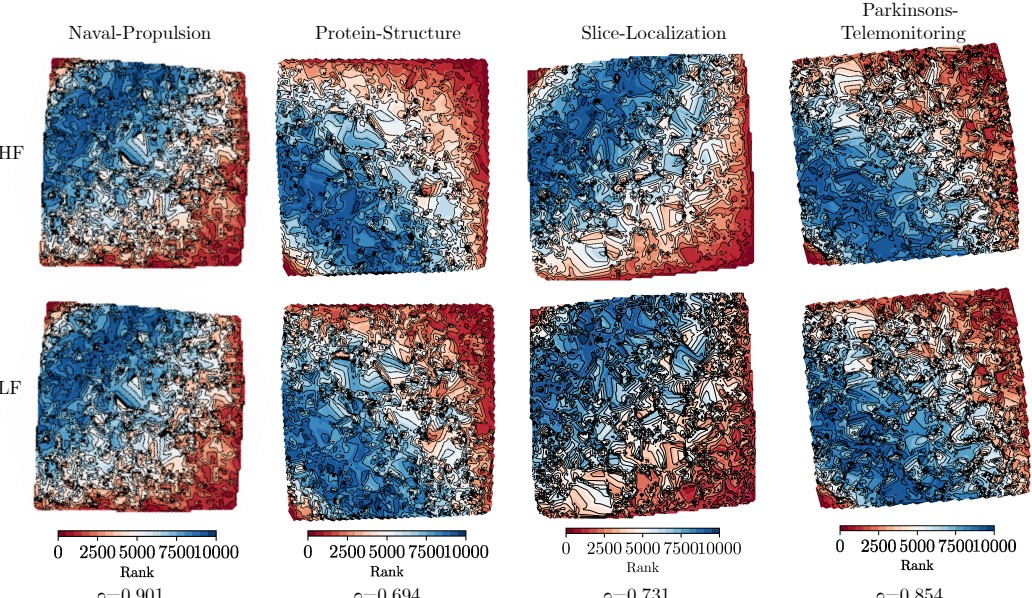

Figure 33: 2D visualization of landscapes of FCNet at high and low fideliteis.

### F.4 PEER COMPARISON ON HYPERPARAMETER OPTIMIZATION FOR FINE-TUNING PRETRAINED IMAGE CLASSIFICATION MODELS

In this section, we additionally adopt the hyperparameter optimization from (Pineda-Arango et al., 2024) for fine-tuning pretrained image classification models on different datasets. A total of 20 problems are used to evaluate the performance of the algorithms, and the results are presented in Table 7. It can be observed that textttLamda+BOHB achieves the best performance across all problems. The overall ranking of the algorithms during the optimization process is illustrated in the Figure 38, showing that `Lamda+BOHB` consistently ranks first after consuming a portion of the resources. The convergence curves in Figure 39 further highlight its superiority in the second phase,

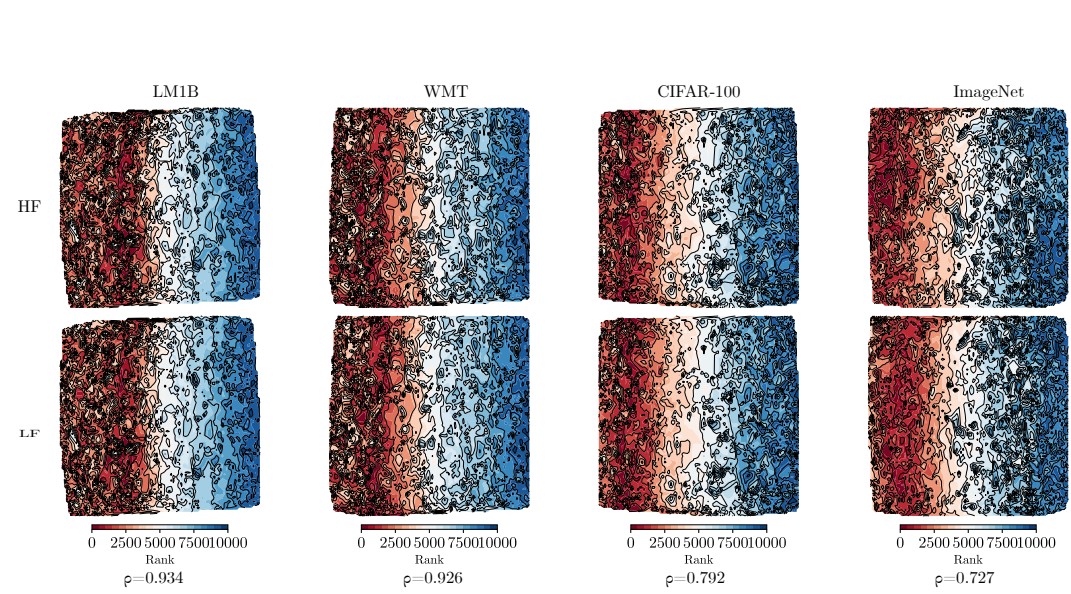

Figure 34: 2D visualization of landscapes of PD1 at high and low fideliteis.

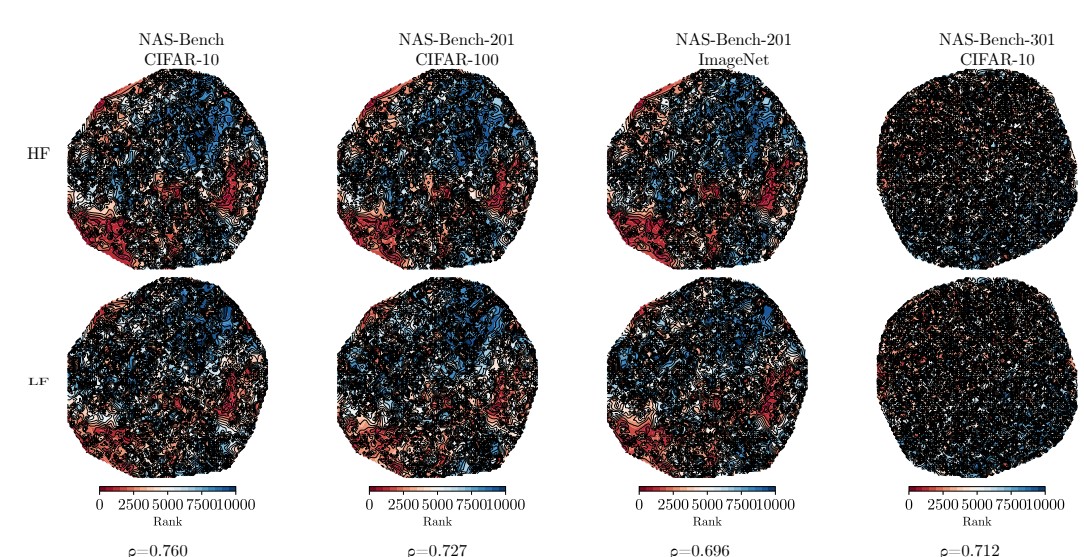

Figure 35: 2D visualization of landscapes of NAS-Banch-201 and 301 at high and low fideliteis.

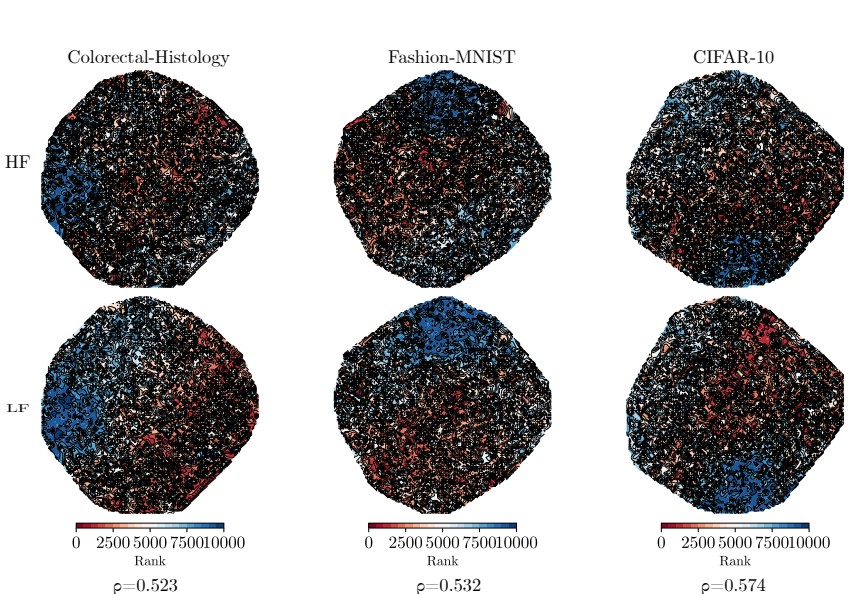

Figure 36: 2D visualization of landscapes of JAHS at high and low fideliteis.

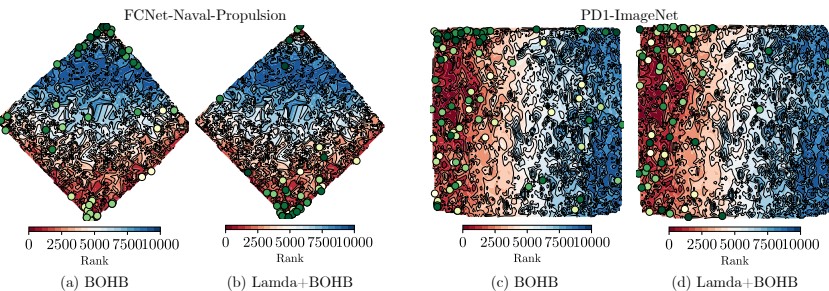

Figure 37: 2D visualization of the sampling points during the optimization process of `Lamda+MUMBO` and `MUMBO` on FCNet-Naval-Propulsion and PD1-ImageNet. The contour represents the dimensionality-reduced landscape of the HF problem, while the points indicate the HF samples collected during optimization. The color gradient from green to yellow indicates the order of sampling, with yellow representing later sampling stages.

where it quickly identifies high-quality solutions and accelerates the optimization process. The experimental results further demonstrate that `Lamda` consistently enhances `BOHB`.

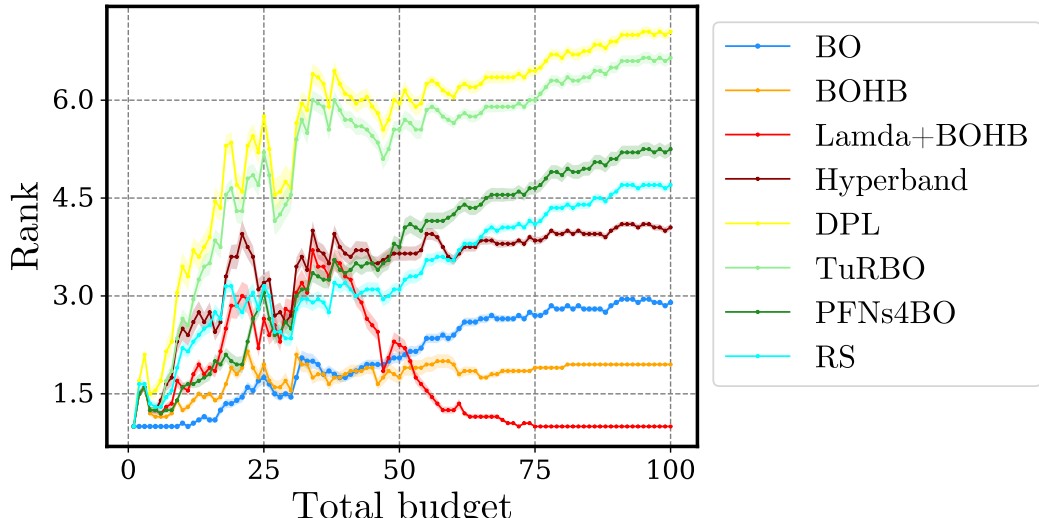

Figure 38: Comparing average relative ranks of peer algorithms across 20 HPO tasks for fine-tuning pretrained image classification models.

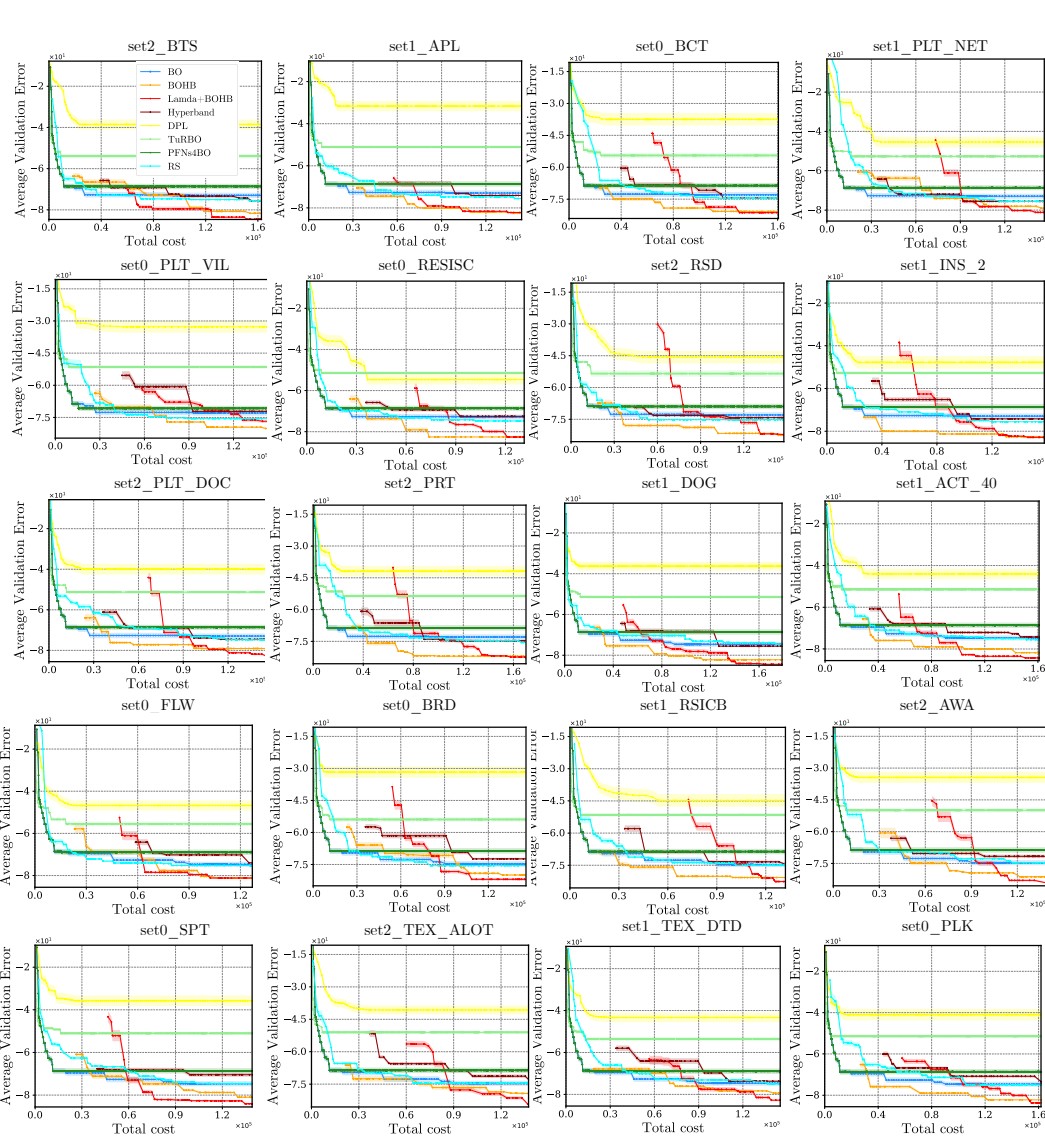

Figure 39: Validation error observed in tuning 20 HPO tasks for fine-tuning pretrained image classification models, comparing peer algorithms.

Table 7: Comparing peer algorithms' final validation errors of the current incumbent at 100 HF evaluations horizons (20 HPO tasks for fine-tuning pretrained image classification models). Runs are averaged over 31 seeds. (The bolded part indicates: "under the Wilcoxon rank-sum test, methods incorporating priors significantly outperform their baselines.")

| Data Name | BO | BOHB | Lamda+BOHB | Hyperband | DPL | TuRBO | PFNs4BO | RS |
|---|---|---|---|---|---|---|---|---|
| mtlbm_extended_set2_BTS | -7.295e+01(1.282e+01) | -8.256e+01(0.000e+00) | **-8.466e+01(2.709e+00)** | -7.556e+01(0.000e+00) | -3.857e+01(2.394e+01) | -5.392e+01(1.042e+01) | -6.862e+01(1.273e+01) | -7.556e+01(0.000e+00) |
| mtlbm_extended_set1_APL | -7.295e+01(1.282e+01) | -8.256e+01(0.000e+00) | **-8.556e+01(0.000e+00)** | -7.556e+01(0.000e+00) | -3.164e+01(2.416e+01) | -5.105e+01(7.422e+00) | -6.862e+01(1.273e+01) | -7.556e+01(0.000e+00) |
| mtlbm_extended_set0_BCT | -7.295e+01(1.282e+01) | -8.256e+01(0.000e+00) | **-8.409e+01(3.032e+00)** | -7.556e+01(0.000e+00) | -3.747e+01(2.900e+01) | -5.446e+01(1.274e+01) | -6.862e+01(1.273e+01) | -7.455e+01(3.034e+00) |
| mtlbm_extended_set1_PLT_NET | -7.295e+01(1.282e+01) | -8.166e+01(2.709e+00) | **-8.556e+01(0.000e+00)** | -7.556e+01(0.000e+00) | -4.541e+01(2.757e+01) | -5.270e+01(8.711e+00) | -6.862e+01(1.273e+01) | -7.556e+01(0.000e+00) |
| mtlbm_extended_set0_PLT_VIL | -7.295e+01(1.282e+01) | -8.256e+01(0.000e+00) | **-8.466e+01(2.709e+00)** | -7.556e+01(0.000e+00) | -3.273e+01(2.761e+01) | -5.142e+01(8.522e+00) | -7.078e+01(1.197e+01) | -7.556e+01(0.000e+00) |
| mtlbm_extended_set0_RESISC | -7.295e+01(1.282e+01) | -8.256e+01(0.000e+00) | **-8.556e+01(0.000e+00)** | -7.556e+01(0.000e+00) | -5.465e+01(2.450e+01) | -5.142e+01(8.522e+00) | -6.862e+01(1.273e+01) | -7.478e+01(2.349e+00) |
| mtlbm_extended_set2_RSD | -7.295e+01(1.282e+01) | -8.256e+01(0.000e+00) | **-8.556e+01(0.000e+00)** | -7.556e+01(0.000e+00) | -4.570e+01(2.934e+01) | -5.342e+01(1.467e+01) | -6.896e+01(1.288e+01) | -7.523e+01(9.984e-01) |
| mtlbm_extended_set1_INS_2 | -7.295e+01(1.282e+01) | -8.256e+01(0.000e+00) | **-8.556e+01(0.000e+00)** | -7.556e+01(0.000e+00) | -4.778e+01(2.946e+01) | -5.270e+01(8.980e+00) | -6.862e+01(1.273e+01) | -7.556e+01(0.000e+00) |
| mtlbm_extended_set2_PLT_DOC | -7.295e+01(1.282e+01) | -8.223e+01(9.984e-01) | **-8.556e+01(0.000e+00)** | -7.556e+01(0.000e+00) | -3.996e+01(2.678e+01) | -5.131e+01(8.203e+00) | -6.862e+01(1.273e+01) | -7.478e+01(2.349e+00) |
| mtlbm_extended_set2_PRT | -7.295e+01(1.282e+01) | -8.256e+01(0.000e+00) | **-8.556e+01(0.000e+00)** | -7.556e+01(0.000e+00) | -4.190e+01(2.468e+01) | -5.356e+01(1.065e+01) | -6.862e+01(1.273e+01) | -7.556e+01(0.000e+00) |
| mtlbm_extended_set1_DOG | -7.467e+01(1.215e+01) | -8.256e+01(0.000e+00) | **-8.489e+01(1.331e+00)** | -7.556e+01(0.000e+00) | -3.633e+01(2.196e+01) | -5.142e+01(8.522e+00) | -6.862e+01(1.273e+01) | -7.462e+01(2.825e+00) |
| mtlbm_extended_set1_ACT_40 | -7.467e+01(1.215e+01) | -8.256e+01(0.000e+00) | **-8.556e+01(0.000e+00)** | -7.556e+01(0.000e+00) | -4.404e+01(2.534e+01) | -5.142e+01(8.392e+00) | -6.862e+01(1.273e+01) | -7.556e+01(0.000e+00) |
| mtlbm_extended_set0_FLW | -7.467e+01(1.215e+01) | -8.256e+01(0.000e+00) | **-8.556e+01(0.000e+00)** | -7.556e+01(0.000e+00) | -4.677e+01(2.420e+01) | -5.547e+01(1.106e+01) | -6.896e+01(1.288e+01) | -7.556e+01(0.000e+00) |
| mtlbm_extended_set0_BRD | -7.467e+01(1.215e+01) | -8.256e+01(0.000e+00) | **-8.500e+01(1.694e+00)** | -7.556e+01(0.000e+00) | -3.176e+01(2.669e+01) | -5.392e+01(1.042e+01) | -6.862e+01(1.273e+01) | -7.556e+01(0.000e+00) |
| mtlbm_extended_set1_RSICB | -7.467e+01(1.215e+01) | -8.256e+01(0.000e+00) | **-8.556e+01(0.000e+00)** | -7.556e+01(0.000e+00) | -4.478e+01(2.931e+01) | -5.146e+01(8.506e+00) | -6.862e+01(1.273e+01) | -7.500e+01(1.694e+00) |
| mtlbm_extended_set2_AWA | -7.467e+01(1.215e+01) | -8.256e+01(0.000e+00) | **-8.556e+01(0.000e+00)** | -7.556e+01(0.000e+00) | -3.442e+01(2.638e+01) | -4.993e+01(9.759e+00) | -6.862e+01(1.273e+01) | -7.466e+01(2.709e+00) |
| mtlbm_extended_set0_SPT | -7.467e+01(1.215e+01) | -8.178e+01(2.349e+00) | **-8.556e+01(0.000e+00)** | -7.523e+01(9.984e-01) | -3.574e+01(2.475e+01) | -5.105e+01(7.422e+00) | -6.862e+01(1.273e+01) | -7.462e+01(2.825e+00) |
| mtlbm_extended_set2_TEX_ALOT | -7.467e+01(1.215e+01) | -8.256e+01(0.000e+00) | **-8.556e+01(0.000e+00)** | -7.556e+01(0.000e+00) | -4.067e+01(2.077e+01) | -5.105e+01(7.422e+00) | -6.862e+01(1.273e+01) | -7.462e+01(2.825e+00) |
| mtlbm_extended_set1_TEX_DTD | -7.467e+01(1.215e+01) | -8.223e+01(9.984e-01) | **-8.556e+01(0.000e+00)** | -7.556e+01(0.000e+00) | -4.332e+01(2.276e+01) | -5.356e+01(1.065e+01) | -6.896e+01(1.288e+01) | -7.500e+01(1.694e+00) |
| mtlbm_extended_set0_PLK | -7.467e+01(1.215e+01) | -8.256e+01(0.000e+00) | **-8.556e+01(0.000e+00)** | -7.556e+01(0.000e+00) | -4.097e+01(2.802e+01) | -5.146e+01(8.506e+00) | -6.862e+01(1.273e+01) | -7.500e+01(1.694e+00) |

# G. ALGORITHM DETAILS

In this section, we outline the workflow of our `Lamda` framework, which operates in two phases: the first-phase low-fidelity search (dubbed as `Lamda-1`) and the second-phase optimization (dubbed as `Lamda-2`). The overall workflow is depicted in Algorithm 1.

The first phase, `Lamda-1`, focuses on identifying promising regions in the search space using LF evaluations. The pseudocode for `Lamda-1` is provided in Algorithm 2. In particular, the sampling strategy in `Lamda-1` is algorithm-agnostic and can be incorporated with most HPO algorithms.

The second phase, `Lamda-2`, allows the use of different algorithms to guide the search. We provide multiple pseudocode to demonstrate how `Lamda-2` can be adapted to various algorithms, including `PriorBand`, `BOHB`, `MUMBO`, vanilla `BO`, and `random search`. The details for each integration are outlined below, with modifications from the original algorithms highlighted in red:

- For `PriorBand`, `Lamda-2` replaces the expert prior $p_\pi(\mathbf{x})$ from (Mallik et al., 2023) with the learned prior $\varphi_{\text{pro}}(\mathbf{x})$ obtained in `Lamda-1`.

- For `BOHB`, `Lamda-2` modifies the uniform sampling distribution in Steps 7 and 8 of Algorithm 3 into the incumbent distribution determined by $\tilde{\varphi}_{\text{pro}}(\mathbf{x})$.

- For `MUMBO` and `BO`, `Lamda-2` combines $\tilde{\varphi}_{\text{pro}}(\mathbf{x})$ with their acquisition functions. The pseudocode for these adaptations is shown in Algorithm 4 and Algorithm 5, respectively.

- For `Random Search`, `Lamda-2` replaces the uniform sampling strategy by the incumbent sampling strategy defined by $\tilde{\varphi}_{\text{pro}}(\mathbf{x})$, as illustrated in Algorithm 6.

---

**Algorithm 3:** Pseudocode for sampling in `Lamda+BOHB`

---

**Input:** Observations $D$, fraction of random runs $\rho$, percentile $q$, number of samples $N_s$,
minimum number of points $N_{\min}$ to build GP models, and bandwidth factor $b_w$
**Output:** next configuration to evaluate

1 Initilize $b \leftarrow \arg\max\{D_b : |D_b| \geq N_{\min} + 2\}$, $\tilde{\rho} \leftarrow \text{Rand}(0, 1)$;
2 **if** $\tilde{\rho} < \rho$ *or* $b = \emptyset$ **then**
3      **return** randomly sampled configuration;
4 **else**
5      Compute $l(\mathbf{x})$ and $g(\mathbf{x})$ as Eqs. (2) and (3) in Falkner et al. (2018);
6      Draw $N_s$ configurations according to $\tilde{\varphi}_{\text{pro}}(\mathbf{x})$ in equation (6);
7      **return** configuration with highest ratio $l(\mathbf{x})/g(\mathbf{x})$

---

---

**Algorithm 4:** Second-phase search with BO

---

**Input:** Input space $\mathcal{X}$, $\varphi_{\text{pro}}(\mathbf{x})$, $w$, $N$ solution for the initial design of GPs, budget $\Lambda_r$, fidelity level $h$, budget function $\lambda_z$.

**Output:** Optimized design $x^*$.

/* Initialization                                     */;

1 Sample $\{\mathbf{x}^i\}_{i=1}^n$ from distribution given by $\varphi_{\text{pro}}(\mathbf{x})$;
2 $y^i \leftarrow f_h(\mathbf{x}^i) + \epsilon^i$, where $\epsilon^i \sim \mathcal{N}(0, \sigma^2)$; $\lambda^i \leftarrow \lambda_z(\mathbf{x}^i, h)$;
3 $\Lambda_r \leftarrow \Lambda_r - \sum_{i=1}^n \lambda^i$;
4 $D \leftarrow \{(\mathbf{x}^i, y^i)\}_{i=1}^n$;
5 **while** $\Lambda_r > 0$ **do**
6     $\varphi(\mathbf{x}) \leftarrow p(\mathbf{x}|D)$;
7     $\tilde{\varphi}_{\text{pro}}(\mathbf{x}) \leftarrow (1 - w) \cdot \varphi(\mathbf{x}) + w \cdot \varphi_{\text{pro}}(\mathbf{x})$;
8     $\mathbf{x}^{n+1} \leftarrow \arg\max_{\mathbf{x} \in \mathcal{X}} \tilde{\varphi}_{\text{pro}}(\mathbf{x}) \text{AF}(\mathbf{x}, \mathcal{D})$ ;
9     $y^{n+1} \leftarrow f_h(\mathbf{x}^{n+1}) + \epsilon$;
10     Update $D \leftarrow D \cup \{(\mathbf{x}^{n+1}, y^{n+1})\}$;
11     $\Lambda_r \leftarrow \Lambda_r - \lambda_z(\mathbf{x}^{n+1}, h)$;
12     $n \leftarrow n + 1$;
13 **return** $x^* \leftarrow \arg\min_{(\mathbf{x}^i, y^i) \in D} y^i$

---

**Algorithm 5:** Second-phase search with MUMBO

---

**Input:** Input space $\mathcal{X}$, prior obtained in the first phase $\varphi_{\text{pro}}(\mathbf{x})$, prior confidence parameter $w$, size $n$ of the initial design, budget for first phase $\Lambda_r$.

**Output:** Optimized design $x^*$.

1 Sample $\{\mathbf{x}^i\}_{i=1}^n \sim \varphi_{\text{pro}}(\mathbf{x})$ and randomly assign fidelity levels $\{z^i\}_{i=1}^n$ with $z^i \sim \text{Uniform}(\{\ell, \ell + 1, \ldots, h\})$;
2 Compute $y^i \leftarrow f_z(\mathbf{x}^i, z^i) + \epsilon^i$, where $\epsilon^i \sim \mathcal{N}(0, \sigma^2)$; $\lambda^i \leftarrow \lambda_z(\mathbf{x}^i, z^i)$;
3 $\Lambda_r \leftarrow \Lambda_r - \sum_{i=1}^n \lambda^i$;
4 Initialize $D \leftarrow \{(\mathbf{x}^i, z^i), y^i\}_{i=1}^n$;
5 **while** $\Lambda_r > 0$ **do**
6     Fit GP to the collected observations $D$, $\varphi(\mathbf{x}) \leftarrow p(\mathbf{x}|D)$;
7     Simulate $N$ samples of $g^*|D$;
8     Prepare $\alpha_{n-1}^{\text{MUMBO}}(\mathbf{x}, z)$ as given by Eq. (5) in Moss et al. (2020);
9     Update $\tilde{\varphi}_{\text{pro}}(\mathbf{x}) \leftarrow (1 - w) \cdot \varphi(\mathbf{x}) + w \cdot \varphi_{\text{pro}}(\mathbf{x})$;
10     Find the next point and fidelity to query

$$(\mathbf{x}^{n+1}, z^{n+1}) \leftarrow \arg\max_{(\mathbf{x}, z)} \tilde{\varphi}_{\text{pro}}(\mathbf{x}) \frac{\alpha_{n-1}^{\text{MUMBO}}(\mathbf{x}, z)}{\lambda_z(\mathbf{x}, z)}$$

11     Collect the new evaluation $y^{n+1} \leftarrow f_z(\mathbf{x}^{n+1}, z^{n+1}) + \epsilon^{n+1}$, $\epsilon^{n+1} \sim \mathcal{N}(0, \sigma^2)$;
12     Append new evaluation to observation set $D \leftarrow D \cup \{(\mathbf{x}^{n+1}, z^{n+1}), y^{n+1}\}$;
13     Update spent budget $\Lambda_r \leftarrow \Lambda_r - \lambda_z(\mathbf{x}^{n+1}, z^{n+1})$;
14 **return** $x^* \leftarrow \arg\min_{\{((\mathbf{x}^i, z^i), y^i) \in D, z^i = h\}} y^i$

**Algorithm 6:** Second-phase search with Random Search

---

**Input:** Input space $\mathcal{X}$, prior obtianed in the first phase $\varphi_{\text{pro}}(\mathbf{x})$, prior confidence parameter $w$, budget for first phase $\Lambda_r$, uniform distribution $p_U$.

**Output:** Optimized design $x^*$.

1  $\varphi(\mathbf{x}) \leftarrow p_U(\mathbf{x})$;

2  **while** $\Lambda_r > 0$ **do**

3      $\tilde{\varphi}_{\text{pro}}(\mathbf{x}) \leftarrow (1 - w) \cdot \varphi(\mathbf{x}) + w \cdot \varphi_{\text{pro}}(\mathbf{x})$;

4      Sample $\mathbf{x}^{n+1} \sim \tilde{\varphi}_{\text{pro}}(\mathbf{x})$;

5      $y^{n+1} \leftarrow f_h(\mathbf{x}^{n+1}) + \epsilon$;

6      Update $D \leftarrow D \cup \{(\mathbf{x}^{n+1}, y^{n+1})\}$;

7      $\Lambda_r \leftarrow \Lambda_r - \lambda_z(\mathbf{x}^{n+1}, h)$;

8  **return** $x^* \leftarrow \arg\min_{(\mathbf{x}^i, y^i) \in D} y^i$

---

