# OpenReview forum: "LAMDA: Two-Phase Multi-Fidelity HPO via Learning Promising Regions from Data"
_ICLR.cc/2025/Conference — Submitted to ICLR 2025_

### Official Review · Reviewer_RsFh · 2024-11-03

**Soundness:** 3
**Presentation:** 3
**Contribution:** 3
**Rating:** 5
**Confidence:** 2

**Summary:**

The paper performs multi-fidelity hyperparameter optimization by identifying promising regions in the low-fidelity landscape followed by searching the regions in the high-fidelity landscape.

**Strengths:**

The original idea seems to be novel.

The paper is well-written.

**Weaknesses:**

The paper lacks experiments comparing the proposed method with existing multi-fidelity HPO frameworks.

**Questions:**

Could this framework be applied to gradient-based hyperparaemter optimization methods?

---

> ### Author Response · Authors · 2024-11-23
> **Responses to Reviewer RsFh**
>
> **W1: Lack experiments comparing the proposed method with existing multi-fidelity HPO frameworks**
>
> We respectfully think this is a misunderstanding. We address your concern from two aspects.
>
> - Our experiments have compared against $9$ state-of-the-art peer HPO algorithms including $7$ specific multi-fidelity HPO algorithms (see Section 3.2 for a detailed description of the peer algorithms).
> - Further, we still would like to highlight that Lamda is not a specific multi-fidelity HPO algorithm. Instead, it is an algorithm-agnostic framework plays as a booster for any baseline HPO algorithms, no matter whether it is for multi-fidelity settings or not. Our results have demonstrated that the performance of five baseline algorithms (PriorBand, BOHB, MUMBO, BO and random search) has been consistently enhanced by using Lamda (see Figure 10, and Appendix F).
>
> **Q1: How can the LAMDA framework be applied to gradient-based hyperparameter optimization methods**
>
> We thank the reviewer for this question. In short, we think Lamda can be applicable for gradient-based HPO methods, and we justify our stance from two aspects.
>
> - First of all, given the non-differentiable nature of hyperparameters, many HPO algorithms, including the ones considered in our work, often leverage zero-order optimization which do not need to access the hyperparameters’ gradients.
> - We are aware of some gradient-based HPO, such as [1] and neural architecture search [2]. They often involve the hyperparameter update into the back-propagation process. For these cases, we have two initial ideas.
>     - The first one is to warm-start good hyperparameters by sampling from the distribution of the promising region obtained in the first-phase search of Lamda.
>     - The other possible solution is to help the loss function design. Similar to the L-1 regularization $\Vert \bf{x} \Vert_1$ introduced to regularize hyperparameters updating within a small region, we can add a term into the loss function, such as the log probability $\tilde\varphi_{\text{pro}}(x)$  to regularize hyperparameters optimize towards the promising regions with higher probability.
>
> [1] Dougal Maclaurin, David Duvenaud, Ryan P. Adams: Gradient based Hyperparameter Optimization through Reversible Learning. ICML 2015: 2113-2122.
>
> [2] Thomas Elsken, Jan Hendrik Metzen, Frank Hutter:Neural Architecture Search: A Survey. J. Mach. Learn. Res. 20: 55:1-55:21 (2019).

---

> ### Author Response · Authors · 2024-11-29
> **Discussion deadline is approaching**
>
> Dear reviewer,
>
> First of all, thank you very much for your comments and constructive suggestions on our paper. They are all really helpful.
>
> Would you please kindly let us know whether our responses are satisfactory? Or if you have further concerns, we are more than to engage in further discussions.
>
> Thank you very much!
>
> Authors of paper #7012

---

> > ### Author Response · Authors · 2024-12-01
> > **We are keen on engaging in further discussions**
> >
> > Dear reviewer,
> >
> > Sorry for chasing this. However, since the discussion deadline is approaching, your kind help, comments, constructive suggestions on our work and our responses will be very much important and appreciated. We are keen on engaging in further discussions.
> >
> > Thank you very much!
> >
> > Authors of paper #7012

---

### Official Review · Reviewer_6ipG · 2024-11-03

**Soundness:** 3
**Presentation:** 3
**Contribution:** 2
**Rating:** 6
**Confidence:** 4

**Summary:**

The paper introduces LAMDA, a two-phase multi-fidelity hyperparameter optimization framework that enhances efficiency by identifying promising regions in low-fidelity landscapes and focusing high-fidelity searches within these areas. This approach reduces computational costs and can integrate with existing methods, improving their performance. While it offers flexibility and data-driven insights, it relies on accurate low-fidelity approximations and may require complex integration. Key questions include its performance with poor LF approximations and sensitivity to parameter choices.

**Strengths:**

1. By concentrating on promising regions, Lamda effectively reduces both the computational cost and time required for hyperparameter optimization.

2. The framework can be integrated with various existing methods, like Prior-Based, Bandit-Based and MFBO methods, enhancing their effectiveness.

3. Unlike methods that depend on expert knowledge, Lamda learns to identify promising regions from data, allowing it to adapt to diverse scenarios. This underscores its data-driven nature.

**Weaknesses:**

1. The performance of Lamda may hinge on the accuracy with which the low-fidelity (LF) landscape (first phase) reflects the high-fidelity (HF) landscape, owing to the two-phase search framework.

2. Section 2.3 states that "Lamda plays as a booster." Consequently, integrating Lamda with existing multi-fidelity hyperparameter optimization methods may necessitate substantial modifications and fine-tuning.

3. The computational overhead associated with implementing the two-phase search strategy, as well as the method's sensitivity to parameters such as the overlapping coefficient and the weights utilized in the second phase of the search, have not been thoroughly addressed.

**Questions:**

1. In line 186, from a mathematical accuracy perspective, should the definitions of $\mathcal{S}_{pro/inf}$ include the case where $f_l(x) = y^*$

2. In line 294, how does the definition $f_\mathcal{D}^*=\min\limits_{\langle x, f_h(x)\rangle\in\mathcal{D}}f_h(x)$ contribute to the overall framework? Does this consideration potentially overlook the significance of $x$?

3. Could you provide a detailed explanation of the results presented in Table 2, as referenced in line 402?

There seem to be some typos:

1. In line 289, $\sigma_f^2(x)$ should be corrected to $\sigma_f^2(\tilde{x})$.

2. In line 298, there seems to be a pair of unnecessary parentheses $( )$ in $\langle(x^i,f_h(x^i))\rangle$.

3. In line 291 and 296, "equation 7" should be formatted as $(7)$.

4. In Theorem 2, what does $2_B^k$ refer to in line 322?

---

> ### Author Response · Authors · 2024-11-23
> **Responses to Reviewer 6ipG**
>
> **W1: Performance of Lamda depends on overlapping regions**
>
> We respectfully think this is a misunderstanding. Our justifications are the following two aspects.
>
> - Our theoretical analysis of the convergence properties  (see Theorem 1 from line 296 and Theorem 2 from line 318) does not rely on the any assumption about the overlapping rate of promising regions within the LF and HF landscapes.
> - Our experiments have considered different scenarios where the promising regions of LF and HF landscape have varying levels of overlaps (Table 5 in Appendix C.4 provides statistics of the overlapping rates). In particular, we would like highlight the results on JAHS-CIFAR-10, JAHS-Colorectal-Histology, and JAHS-Fashion-MNIST, whose overlaps are low. It is demonstrated that Lamda consistently enhance the baseline algorithms (see Figures 17, 19, 21, and 23) in most scenarios.
>
> **W2: Integrating Lamda with existing HPO methods may necessitate substantial modifications and fine-tuning**
>
> We address your concerns from the three aspects.
>
> - First, we respectfully argue that Lamda is an algorithm-agnostic framework. Integrating Lamda  into existing HPO methods only involves replacing the sampling strategies of the baseline optimizer with  $\tilde\varphi_{\text{pro}}$. Therefore, it does not need any modifications and fine-tuning. We have discussed how to integrate Lamda into different types of HPO algorithms, where we use PriorBand, BOHB, MUMBO, vanilla BO and random search as the baselines (see lines 261-278).
> - Additionally, we have added pseudocode to illustrate the integration process for each of these algorithms (see Appendix G in the uploaded PDF). The required modifications are minimal and specifically focus on the sampling step (highlighted in red in the pseudocode for clarity).
> - Our experimental results demonstrated that using Lamda consistently enhance the baseline algorithms ( Figures 17 to Figure 24  in Appendix F.1). We also kindly refer the reviewer to the second bullet point of our responses to the reviewer **gCCh**’s **Q7** for further justifications about this.
>
> **W3: Method's sensitivity to parameters**
>
> We respectfully argue that the algorithm’s performance is not sensitive to the reviewer concerned hyperparameter settings. This assertion is empirically validated by our parameter sensitivity study (see Section 4.3). Specifically, we validated this by conducting a sensitivity study on $B$ (computational overhead), $\gamma$, $\alpha$, $ \Delta$ and $w$, using Lamda+BOHB as a case. The experimental results demonstrated that our performance is not sensitive to those parameter settings (see lines 491, 498, 502 and Figures 27, 28, 29, 30, and 31 for convergence curves).
>
> **Q1: The definition of $\mathcal{S}_{\text{pro}}$**
>
> Thank you for the correction. We have modified this by including the case where $f_l(x)=y^∗$ when calculating $\mathcal{S}_{\text{pro}}$.
>
> **Q2: The definition of  $f^\ast_\mathcal{D}$**
>
> Thank you for your question. We address your concerns from the following two aspects.
>
> - The  definition of  $f^\ast_\mathcal{D}$ is leveraged in the design of expected improvement (EI), which is a classical acquisition function of Bayesian optimization. In Bayesian optimization, the surrogate model such as Gaussian processes will be utilized in the calculation of EI to involve $\tilde{\mathbf{x}}$, see Equation (8) in our manuscript.
> - In our work, we further consider the importance of $\bf{x}$ by adding $\tilde{\varphi}_\mathrm{pro}(\mathbf{x})$ into the design of acquisition function, as seen in equations (7) and (9). The `Lamda-BO` used the **priors** from low-fidelity promising region to better guide the search direction of the underlining HPO algorithms.
>
> **Q3: Explanation of the results presented in Table 2**
>
> We apologize for causing the reviewer’s confusion. We clarify the meaning of Table 2 as follows.
>
> - Due to the large amount of comparisons considered in our experiments, we try to use a succinct format to show the comparison results of Lamda+x against the corresponding baseline algorithm (x represents the baseline algorithm, i.e., PriorBand, BOHB, MUMBO, vanilla Bayesian optimization, and random search, used in our experiments).
> - Specifically, if Lamda+x significantly outperforms the baseline by using Wilcoxon singed-rank test, we count it as ‘Win’. Otherwise, if Lamda+x is defeated by the baseline, we count it as ‘Lose’. Further, if the performance of Lamda+x is not significantly different from the baseline, we count it as ‘Tie’.
>
> **T1:**
>  We have corrected it to $\sigma^2_f(\tilde{\mathbf{x}})$ in our upload PDF version.
>
> **T2:**
> We have corrected it to $\left\lbrace\langle\mathbf{x}^i,f_h(\mathbf{x}^i)\rangle\right\rbrace_{i=2}^n$ in our upload PDF version.
>
> **T3:**
> We have corrected it to  "equation (7)" in our upload PDF version.
>
> **T4:**
> Sorry for the typos. It should be $2^{k_B}$.

---

> ### Comment · Reviewer_6ipG · 2024-11-26
>
> Thanks for your patient response. Due to some misunderstanding on my part regarding the performance of your algorithm, after carefully considering your answer, I believe the rating of your paper could be 7. However, since there is no rating of 7 on this scale, I will maintain the current score of 6. Nevertheless, I have increased my confidence and now strongly believe your paper deserves this rating.

---

> > ### Author Response · Authors · 2024-11-29
> > **Follow-up responses to Reviewer 6ipG**
> >
> > We sincerely thank the reviewer for accepting our justifications for your concerns. May we respectfully ask whether your mentioned rating $7$ is “Weak Accept” or “Accept” in other AI conference system or not?
> >
> > In addition, as per requested by the reviewer **gCCh**, we have further revised our manuscript during the extended rebuttal period.
> >
> > - We have rewritten the Introduction section to rephrase the motivation and contributions of this paper.
> > - As per requested by the reviewer **gCCh**, we have moved two pseudo codes (Algorithms 1 and 2) from the Appendix G to the Section 2 of the updated manuscript.
> > - We have consolidated the experimental section.
> >     - Figures 7 and 37 provide two examples to demonstrate that how Lamda+BOHB and Lamda+MUMBO boost the performance of the corresponding baseline BOHB and MUMBO by helping them focus more on exploring the promising regions.
> >     - As per requested by the reviewer **gCCh**, we have added 20 additional HPO tasks on large-scale pretrained models with 305M parameters [1]. The experimental results show that Lamda+BOHB not only enhances the baseline BOHB but also outperforms all other peer algorithms (see Table 7, Figures 38 and 39).
> >
> > [1] Arango S P, Ferreira F, Kadra A, et al. Quick-Tune: Quickly Learning Which Pretrained Model to Finetune and How, ICLR, 2024.

---

### Official Review · Reviewer_gCCh · 2024-11-04

**Soundness:** 2
**Presentation:** 2
**Contribution:** 2
**Rating:** 5
**Confidence:** 3

**Summary:**

This work aims to overcome the drawbacks of popular hyperparameter optimization (HPO) methods that lead to wasted computational resources, namely the exhaustive exploration of the entire search space and expensive high-fidelity (HF) evaluations. The paper proposes a two-phase multi-fidelity (MF) framework for HPO called LAMDA (Learning promising regions from data) to address these issues. The framework can be integrated into existing HPO techniques.

Based on the observation of overlapping high-performing regions between low-fidelity (LF) and HF landscapes, in the first phase, the algorithm learns promising regions from evaluations of the LF landscape. In the second phase, these learned promising regions are leveraged to more efficiently explore the HF landscape in the actual HPO process, thereby avoiding the waste of resources in less promising regions.

The paper contains an empirical evaluation of the proposed method combined with several state-of-the-art HPO methods (PriorBand, BOHB, MUMBO, BO) and random search as a baseline. On a total 36 benchmarks (tabular, surrogate and real) the paper demonstrates that the proposed method leads to substantial improvement.

Overall, I think the paper leaves more open questions than it convinces me that LAMDA is a great method that should be used in practice. While the idea is surprisingly simple, the paper does not show why previous multi-fidelity methods do not work as advertised (e.g., conduct too much exploration, or require evaluations on the highest fidelity), it is unclear how to implement the method (substantial details and the actual implementation are missing), and the experiments also lack detail and should be extended by a few recent benchmarks.

**Strengths:**

* The paper empirically demonstrates the effectiveness of the proposed method across various HPO tasks and HPO techniques. Additionally, the paper offers a theoretical analysis for Bayesian optimization and Hyperband.
* The proposed method can be integrated into existing HPO techniques, such as prior-based and bandit-based methods, as well as Bayesian optimization (BO).
* Although the approach introduces several hyperparameters that need to be tuned, the paper proposes specific values and demonstrates in experiments that the impact of the hyperparameters on performance is minor.

**Weaknesses:**

* The work claims that it overcomes the limitations of existing multi-fidelity methods, which require exhaustive searches across the entire search space (Reference: Table 1). However, the proposed method (LAMDA) still necessitates an exhaustive search of the LF landscape, using the learned promising regions only for searches in the HF landscape. Approaches like MFBO or bandit-based methods similarly leverage LF evaluations to focus HF evaluations on promising areas. For instance, BOCA already proposes using cheap fidelities to identify promising regions for HF experiments (“Therefore, one may use cheap low fidelity experiments with small (N, T ) to discard bad hyper-parameters and deploy expensive high fidelity experiments with large (N, T ) only in a small but promising region.” [1]).
While LAMDA may offer greater efficiency with this approach, the claim that the overall idea would be new (Reference: ”To address these challenges, we propose leveraging LF problems to identify promising regions for the HF problem.” [Row 93-94]) and that it would eliminate the need for exhaustive searches across the entire search space (Reference: Table 1) is not true.
* The paper implies to have initially observed that promising solutions in HPO tend to overlap between LF and HF evaluations. (References: “We have observed that high-quality solutions in HPO exhibit some overlapping between high- and low-fidelity problems.” [Rows: 14-15], “This strategy is inspired by our observation of overlapping promising regions between high- and low-fidelity HPO problems.” [Rows: 96-97]). However, this observation is not new at all and serves as the basis for many existing publications on MF-HPO [3 (Figure 1), 1, 2]. Also, if this is proposed as a novel, key observation, the paper should dedicate a section to describe and analyze this behavior.
* The paper leaves many open questions regarding the experimental setup (see questions below).

## Minor criticisms (putting them here as there is no field for extra comments):
* There is a typo on row 103: "quirky."
* Figure 10 is quite small and displays multiple overlapping curves, making it difficult to read clearly.
* Lines 204–205 are ambiguously phrased, suggesting that the overlapping coefficient could only be 0 or 1. This should be rephrased to clarify that it can also take intermediate values.
* The paper Swersky et al. (2013) is about multi-tasking Bayesian optimization. It contains a two-fidelity setting, but are you sure that you did not mean "Freeze-Thaw Bayesian Optimization" (Swersky et al. (2014))?

## Further references

* [1]: Kandasamy, K., Dasarathy, G., Schneider, J., & Póczos, B. (2017, July). Multi-fidelity bayesian optimisation with continuous approximations. In International conference on machine learning (pp. 1799-1808). PMLR.
* [2]: Falkner, S., Klein, A., & Hutter, F. (2018, July). BOHB: Robust and efficient hyperparameter optimization at scale. In International conference on machine learning (pp. 1437-1446). PMLR.
* [3]: Klein, A., Falkner, S., Bartels, S., Hennig, P., & Hutter, F. (2017, April). Fast bayesian optimization of machine learning hyperparameters on large datasets. In Artificial intelligence and statistics (pp. 528-536). PMLR.

**Questions:**

* It is stated that the framework can be integrated with existing HPO techniques. However, especially for the multi-fidelity techniques used in the paper, such as BOHB, it is not clearly explained how LAMDA is integrated. It is very unclear how this could be done, as BOHB itself is a multi-fidelity method that can make use of low and high fidelities. The paper should be extended to contain a per-method discussion on how LAMDA is integrated to make it easier for the reader to understand this.
* Following up on the previous question, why does the proposed method outperform existing multi-fidelity methods? The existing multi-fideliy method are made to **not** explore the high fidelity in much detail (but this paper claims they do), and I think this warrants further discussion, as I do not understand why BOHB etc would not perform well on these problems. Source code of the method would increase my trust in the proposed method and its evaluation.
* Again, following up on the previous two questions: How does the proposed method compare against https://proceedings.mlr.press/v89/song19b.html ?The method by Song et al. appears to be closely related and targets the same problem.
* The experiment section (4.) does not specify the fidelity levels used in the experiments. While Appendix C (Tables 3 and 4) provides a fidelity range for some experiments, it does not clearly indicate the specific fidelity levels applied.
* Have you observed settings where the computational expenses introduced in the first phase do not outweigh the rewards in the second phase? If so, were you able to characterize those settings to indicate when the application of the proposed method is less meaningful?
* Why not use more recent benchmarks, such as the ones used in DYHPO, Quicktune, DPL? Also, since the paper mentions DPL, please make sure to cite it, as well as other BO methods you mention (e.g., TURBO). It is great that you are using 33 different benchmark tasks, but using a decision tree benchmark (rpart) to demonstrate a multi-fidelity hyperparameter optimization method at ICLR appears to be out of place.
* Have you thought about or observed failure modes in the proposed probability distribution used to guide the HPO process (Equation 6)? For instance, in complex HF landscapes with several local minima, augmenting the density learned on the fly with the density from the first phase search might shift the sampling distribution away from sufficiently good but not optimal solutions in the early phases, deteriorating anytime performance.
* What Bayesian Optimization library is used in the experiments? It would be great if you could add references to the other baselines, too (There it is less ambiguous, but for BO, it is really ambiguous what library you used). In any case, it would be important to use a state-of-the-art library, for example, HEBO.
* How much is the lower fidelity explored in practice? I think it would be important to add a figure on when the method moves from the lower to the higher fidelity.
* Why would it be sufficient to only update the PDF of promising regions? Is it because in the TPE sampler new hyperparameter settings are only sampled from the KDE describing the promising regions?
* How does the proposed overlap coefficient compare to the one proposed in https://arxiv.org/abs/2212.06751?

---

> ### Author Response · Authors · 2024-11-23
> **Responses to Reviewer gCCh (1/3)**
>
> **W1:**
> We thank the reviewer’s detailed comments. Here we justify our stance from the following four aspects.
>
> - First, we respectfully argue that Lamda is not a specific multi-fidelity hyperparameter optimization (HPO) algorithm. Instead, it is an algorithm-agnostic framework that can be integrated with any existing HPO algorithms no matter whether they consider multiple fidelities or not. Please also kindly refer to our response to the **W1** of the reviewer **EWdW**.
> - As for the bandit-based HPO algorithms (e.g., Hyperband [1]), they always sample the entire space in both low- and high-fidelity levels. In [2], this ‘global (or random) sampling strategy’ has been criticized to be inefficient. Accordingly, they proposed PriorBand that uses expert belief as a prior to constrain the sampling within an incumbent area. However, eliciting an appropriate expert belief is non-trivial and will incur extensive expert’s cognitive burden. Differently, Lamda does not rely on any expert crafted prior but to learn a reasonably reliable prior by exploring the low-fidelity (LF) landscape of the underlying HPO task.
> - As for BOCA, its core idea is a unified surrogate model that integrates the search domain and the fidelity space. When searching for the next point of merit including its fidelity level, the optimization of the acquisition function is still conducted within the entire search domain (see the equations (6) and (7) in [3]). Differently, our philosophy is to strategically constrain the search/sampling within the promising region learned from exploring the LF landscape of the underlying HPO task. If we consider Bayesian optimization as the baseline, we can directly integrate the learned promising region (represented as a PDF in equation (6)) into the acquisition function (see equation (7) of our manuscript).
> - We understand the reviewer’s concern about our statement of ‘exhaustive search’ while our first-phase search still considers searching within the LF landscape. We have modified our statement in the updated PDF (see lines 90-94, highlighted in red).
>
> [1] Lisha Li, et al.: Hyperband: A Novel Bandit-Based Approach to Hyperparameter Optimization. J. Mach. Learn. Res. 18: 185:1-185:52(2017).
>
> [2] Neeratyoy Mallik: PriorBand: Practical Hyperparameter Optimization in the Age of Deep Learning. NeurIPS 2023.
>
> [3] Kirthevasan Kandasamy, et al.:Multifidelity Bayesian Optimisation with Continuous Approximations. ICML 2017: 1799-1808.
>
> **W2:**
> We justify our stance from the following three aspects.
>
> - The reviewer listed references [1] and [2] only made an assumption about the overlap between low- and high-fidelity landscapes, without providing any (empirical) evidence. As for [3], the authors merely used SVM on MNIST as the example to explore the potential overlap of the validation error landscapes across different dataset sizes, which can be seen as fidelity levels.
> - In this paper, we have conducted large-scale experiments (see Figure 11 in Appendix D and Figures 13 to 16 in Appendix E) which support our observation, as picked up by the reviewer. To consolidate our assertion, we have added another perspective from visualizing the entire landscape (see Figures 32 and 33 in Appendix F.4 in our upload PDF version) during this rebuttal.
> - Further, we would like to reiterate the differences w.r.t. [1-3] and our key novelty as a framework that plays as a booster for the baseline algorithm, rather than a specific multi-fidelity HPO algorithm. The potentially overlapped low- and high-fidelity landscapes support our rationale of learning a reasonably reliable prior (i.e., promising region represented by PDF) by exploring the LF landscape.  Nevertheless, it is worth noting that we do not strongly rely on the significant overlaps of the promising regions in low- and high-fidelity landscapes (see our responses to **W2** and **Q1** of the reviewer **EWdW**).
>
> [1] Kandasamy, K., Dasarathy, G., Schneider, J., & Póczos, B. (2017, July). Multi-fidelity bayesian optimisation with continuous approximations. In International conference on machine learning (pp. 1799-1808). PMLR.
>
> [2] Falkner, S., Klein, A., & Hutter, F. (2018, July). BOHB: Robust and efficient hyperparameter optimization at scale. In International conference on machine learning (pp. 1437-1446). PMLR.
>
> [3] Klein, A., Falkner, S., Bartels, S., Hennig, P., & Hutter, F. (2017, April). Fast bayesian optimization of machine learning hyperparameters on large datasets. In Artificial intelligence and statistics (pp. 528-536). PMLR.
>
> **W3:**
> We have provided detailed responses to the open questions regarding the experimental setup in the corresponding sections below. Please refer to our replies to the specific questions for clarity and further elaboration.

---

> ### Author Response · Authors · 2024-11-23
> **Responses to Reviewer gCCh (2/3)**
>
> **Q1:**
> We have discussed how to integrate Lamda into different types of HPO algorithms, where we use PriorBand, BOHB, MUMBO, vanilla BO and random search as the baselines (see lines 261-278).  Additionally, we have added pseudocode to illustrate the integration process for each of these algorithms (see Section G in Appendix).
>
> **Q2:**
> We address your concerns from the following four aspects.
>
> - For the concern about “why […] outperform existing multi-fidelity methods”, please refer to our responses to your **W1** and the reviewer **EWdW**’s **W1**.
> - For the concern about “The existing multi-fidelity method are made to not explore the high fidelity in much detail […]”, we apologize for this over-claim. Please kindly refer to our response to your **W1** (the fourth bullet point).
> - For the concern about “[…] why BOHB etc would not perform well […]”, we respectfully think this might be a misunderstanding. As our responses to your **W1** and the reviewer **EWdW**’s **W1**, Lamda plays as a booster for the baseline. In other words, by integrating Lamda, BOHB becomes better (see the convergence trajectories comparisons of Lamda+BOHB and BOHB in Figures 17 and 18 in Appendix F.1, and also the solution distributions comparisons in Figure 36 in Appendix F.2).
> - We have prepared a demo code for proof-of-concept verification on Lamda+BOHB in https://doi.org/10.5281/zenodo.14203431. We are committed to promote open science and promise to release the entire code after the acceptance of this paper.
>
> **Q3:**
> We address the reviewer’s concern from two aspects.
>
> - First, the reviewer mentioned paper, MF-MI-Greedy,  is a specific multi-fidelity HPO algorithm. While the proposed MF-MI-Greedy algorithm also introduced the concept of two-phase search, it needs to switch between LF and HF across different rounds. Differently, Lamda is an algorithm-agnostic framework (see our responses to your **W1** and the reviewer **EWdW**’s **W1**), and it uses the search within the LF landscape to learn a reasonably reliable prior to jump-start the baseline algorithm on the underlying HPO task.
> - Furthermore, we believe Lamda can be integrated into this MF-MI-Greedy algorithm. For example, the learned prior can be used to inform the mutual information calculation in its first phase, and/or its acquisition function in the second phase (line 4 and line 5 in Algorithm 1 of the mentioned paper).
>
> **Q4:**
> We apologize for this oversight. In this rebuttal, we have added a table summarizing the settings of the low-fidelity levels (see Table 5 in the Appendix C.4).
>
> **Q5:**
> We appreciate the reviewer’s concern. However, our experiments on 33 test problems (including fully connected networks, transformers, ResNet, neural architecture search with NAS-Bench-201 and NAS-Bench-301, as well as joint architecture and hyperparameter search with JAHSBench cases ) do not show the reviewer mentioned scenarios. In short, our experimental results have demonstrated that Lamda consistently enhances the baseline algorithms.
>
> **Q6:**
> We have conducted additional experiments as per suggested by the reviewer (20 HPO tasks for fine-tuning pretrained image classification models). Experimental results (see Appendix F.4) have also demonstrated that Lamda consistently enhance the baseline algorithms. In addition, we have added references to relevant methods mentioned in the paper.
>
> **Q7:**
> We address the reviewer’s concerns from two aspects.
>
> - We have tested Lamda on problems with complex landscapes with multiple local optima, including NAS-Bench-201, NAS-Bench-301, and JAHS (see Figures 34 and 35 in Appendix F.3 of the uploaded PDF for visualizations of these landscapes under both high- and low-fidelity settings). Our experimental results have demonstrated that Lamda consistently enhances the baseline algorithms (see Figures 17, 19, 21, and 23).
> - We thank the reviewer for picking anytime performance. To address this concern, we would like to reiterate the key novelty of Lamda as an algorithm-agnostic framework that can boost any baseline HPO algorithms.
>     - Note that the learned prior in the first-phase search is not a hard constraint for the second-phase search (i.e., the underlying HPO task). Instead, as shown in our equation (6), it plays as a probabilistic aggregation to the corresponding search strategy of the baseline algorithm (e.g., acquisition function or bandit sampling).
>     - Furthermore, because the prior is learned from the low-fidelity landscape of the underlying HPO task with limited budget, it mitigates the risk of running with an erroneous prior crafted by expert’s experience (see the second bullet point of our responses to your **W1**).
>
> These constitute the key reasons that why the baseline algorithm’s performance will be enhanced, or at least will not be deteriorated, by using Lamda. We hope these justifications can further address the your concerns in **Q2** and **Q5**.

---

> ### Author Response · Authors · 2024-11-23
> **Responses to Reviewer gCCh (3/3)**
>
> **Q8:**
> We utilized the GPyOpt [1] package for our Bayesian Optimization implementation, while MUMBO was implemented using the Emukit [2] framework. Other peer algorithms were implemented using the packages provided in their original papers.
>
> [1] https://github.com/SheffieldML/GPyOpt
>
> [2] https://github.com/EmuKit/emukit/blob/main/notebooks/Emukit-tutorial-multi-fidelity-MUMBO-Example.ipynb
>
> **Q9:**
> We have updated the convergence plots of Lamda under the BOHB and MUMBO frameworks (see Figures 17, 18, 19, and 20 in Appendix F.1) to indicate the actual resources consumed during the first phase. Since these resources vary across different seeds, we repeated the experiments multiple times and reported the average values in the updated figures.
>
> **Q10:**
> We respectfully think the reviewer’s concern might be caused by a misunderstanding of our idea.
>
> - In the first phase, both the promising region $\varphi_{\text{pro}}$ and the inferior region $\varphi_{\text{inf}}$ are updated with the increasingly collected data (see line 185 to 189). In the second phase, the promising region $\tilde\varphi_{\text{pro}}$ that guides the underlining HPO algorithms is updated on-the-fly with the search process (see line 233).
> - As justified in our responses to your **W1** and the reviewer **EWdW**’s **W1**, Lamda plays as a booster that can be integrated with any other HPO algorithms. That is to say all components of the baseline will be updated accordingly as it is.
>
> **Q11:**
> We thank the reviewer for bringing this new reference, which we considered not directly related to our work. Hereby, we clarify our differences from two aspects.
>
> - From the definition perspective, the overlap coefficient (OVL) defined in our paper is inspired by a common measure of agreement in statistics [1]. It quantifies the similarity between **probability distributions** of promising regions in low- and high-fidelity landscapes. In contrast, the $\gamma$ -set similarity introduced in [2] was performed on the **set level**, the Borel body as pointed out. Mathematically, the $\gamma$ -set similarity was calculated based on the density function of two sets, which can be seen as a variant of OVL based on the following facts.
>     - Firstly, the work calculated the difference of two sets as $d_{tv} = \frac{1}{2}\int \| p(x \mid \mathcal{X}\_1) - p(x \mid \mathcal{X}\_2) \| dx$
> (Equation 27 in [2]). Note that $d_{tv}$ is indeed a variant of OVL since $\text{OVL} = 1 - d_{tv}$. This can be illustrated by the areas of two probability distributions (see the presented figure below).
>
>     - Therefore, the $\gamma$ -set similarity was calculated by $S = \frac{1 - d_{tv}}{1+d_{tv}} = \frac{ \text{OVL}}{2 - \text{OVL}}$ (Equation 26 in [2]). When $\text{OVL} = 1$, $S = 1$. Obviously, $S$ reduces to $0$ when OVL reduces to $0$.
>
>     Based on these derivations, the $\gamma$ -set similarity [2] is a  monotonic transformation of our OVL. We are humble to think the $\gamma$ -set similarity might also work in Lamda. However, we also think this does not compromise the key novelty of Lamda. Further, we believe Lamda, as an algorithm-agnostic framework, is general enough to accommodate other new algorithmic component.
>
> - From the problem setup perspective, [2] focus on meta-learning enabled HPO. Differently, as justified in our responses to your **W1**, Lamda uses a principled prior learning strategy to boost the performance of baseline algorithms in a multi-fidelity optimization paradigm.
>
> **M1:**
>  We have corrected it to  ‘quickly’ in our modified version.
>
> **M2:**
> We have update figure 10 in our modified version.
>
> **M3:**
> We have corrected it in our upload PDF version. Specifically, we rephrased lines 204–205 to:
>
> Note that $\rho\left(\varphi_1(\mathbf{x}),\varphi_2(\mathbf{x})\right)$ ranges from 0 to 1, where $\rho=1$ if and only if the two distributions are fully overlapped, and $\rho=0$ if there is no intersection at all.
>
> **M4:**
> We want to use the Swersky et al. (2013) paper to support the use of Bayesian optimization methods for solving multi-fidelity problems.

---

> > ### Comment · Reviewer_gCCh · 2024-11-26
> > **Thank you very much for the detailed response**
> >
> > Thank you very much for the detailed answer. After carefully reading it, I believe that the overall idea has not been explored yet, but I still have too many open questions before I can recommend acceptance: or to phrase it with the words of the AC: I think the community does not yet benefit from having this paper accepted. Given your thorough response, I think that you strongly believe in your own work, and am very certain that you can improve the manuscript substantially to make your paper easier to understand. Below, you can find some more questions and thoughts I had after reading your answers:
> > * What does it mean that you learn a prior? Is this an analogy using the priorband paper, or is this how you would characterize your method?
> > * Regarding BOCA: I do not understand why optimizing the acquisition function across the whole space would be a problem. If the model or the acquisition function is tailored for this, why not?
> > * I know that summarizing the reviews is not my task, but rather the task of the area chair. Nonetheless, I'm concluding from the fact that multiple reviewers misunderstood the motivation as well as the integration with the base method means that the paper is not ready for publication yet. All explanations and motivation of the method need to go to the main paper, and must not be hidden in the appendix. I think the authors can drastically improve the paper based on the feedback to make it more accessible:
> >     * Better explain that you create a meta-method and that it is easy to integrate with any base method. By this I mean that you should improve your overall text that explains this. In that direction, thank you for adding Appendix G. However, I think there is a mistake in MUMBO, as I would expect the multiplier to be inside the argmax. Also, if possible, could you move Algorithms 1 & 2 to the main paper?
> >     * show that their method samples the high fidelity less.
> >     * show that other methods indeed suffer from sampling the high fidelity too often.
> > * I appreciate the authors providing link to code. However, I think this needs to be done with the submission, not towards the end of the rebuttal period. I do not have sufficient time left to check the code submission. Also, it would increase my trust to have the full integration into all tools available.

---

> > > ### Author Response · Authors · 2024-11-29
> > > **Follow-up responses to Reviewer gCCh (1/2)**
> > >
> > > >**Q1:** What does it mean that you learn a prior? Is this an analogy using the priorband paper, or is this how you would characterize your method?
> > >
> > > We thank the reviewer for this question. First of all, we agree that the concept of prior is the same as the one used in PriorBand. However, we would like to respectfully argue the fundamental differences from the following two aspects.
> > >
> > > - In PriorBand, the prior is elicited as an expert belief. In practice, as stated in the PriorBand paper’s Appendix D.3 [1], for a given machine learning algorithm, the priors are generated by taking different configurations’ performance recorded in HPO benchmarks (e.g., PD1, JAHS-Bench-201, and LCBench). For example, the near-optimal priors are generated from the $50,000$ top-ranked configurations. Note that the computational cost for obtaining these configurations’ performance data is not counted in the PriorBand algorithm. Moreover, this prior setting is rigid where it is not clear what happens if the new HPO tasks are completely different from the benchmarks used to generate the priors.
> > > - Differently, in our Lamda framework, we try to learn a reasonably reliable prior by exploring the low-fidelity (LF) landscape of the underlying HPO task. This is implemented as the first-phase search, which is also restricted to limited computational budgets. By doing so, we equip the underlying HPO methods with a ‘bespoke’ prior without relying on expert belief. Further, our prior is represented as a probability density function (PDF) of the promising region, rather than a rigid domain. This strategy provides flexibility to HPO task in the high-fidelity landscape. Meanwhile, it also mitigates the risk of being trapped by a poorly represented promising region. Given these justifications, you can tell that it is not necessary to leverage the data external to the underlying HPO task in Lamda. Meanwhile, it does  not incur any additional computational cost  (please see our response to **W1** of reviewer **EWdW**).
> > >
> > > [1] Neeratyoy Mallik, et al.: PriorBand: Practical Hyperparameter Optimization in the Age of Deep Learning. NeurIPS 2023.
> > >
> > > >**Q2:** Regarding BOCA: I do not understand why optimizing the acquisition function across the whole space would be a problem. If the model or the acquisition function is tailored for this, why not?
> > >
> > > We apologize for causing this confusion and we further clarify the comparison with BOCA from three aspects.
> > >
> > > - First, we do not intend to criticize BOCA’s acquisition function optimization across the entire search domain, while we appreciate its core contribution is on building surrogate model tailored for multi-fidelity HPO.
> > > - On the other hand, the third bullet point of our response to your **W1** aims to highlight the difference in terms of the search behavior. In other words, Lamda strategically re-weights different regions of the search space (see Equation (7)). By doing so, we enable the baseline HPO algorithm to focus more on the promising areas while still maintaining a certain level of global exploration capability. As our response to your **Q1**, this flexibility mitigates the risk of being trapped by a poorly represented promising region.
> > > - Last but not least, we think Lamda is not orthogonal to BOCA. Instead, as an algorithm-agnostic framework, we think Lamda can also be used to boost the performance of BOCA.

---

> > > > ### Author Response · Authors · 2024-11-29
> > > > **Follow-up responses to Reviewer gCCh (2/2)**
> > > >
> > > > >**Q3:** All explanations and motivation of the method need to go to the main paper, and must not be hidden in the appendix. I think the authors can drastically improve the paper based on the feedback to make it more accessible:
> > > > >- Better explain that you create a meta-method and that it is easy to integrate with any base method. By this I mean that you should improve your overall text that explains this. In that direction, thank you for adding Appendix G. However, I think there is a mistake in MUMBO, as I would expect the multiplier to be inside the argmax. Also, if possible, could you move Algorithms 1 & 2 to the main paper?
> > > > >- show that their method samples the high fidelity less.
> > > > >- show that other methods indeed suffer from sampling the high fidelity too often.
> > > >
> > > > We appreciate the reviewer’s constructive suggestions on improving our presentation. We address your concerns in the following five aspects.
> > > >
> > > > - For the concern about [… explanations and motivation of the method …], we have revised the motivation statement in Section 1 of our uploaded PDF (see lines 76–123, highlighted in red color). We sincerely hope this will be clearer to the reviewer.
> > > > - For the concern about [… explain that you create a meta-method and that it is easy to integrate with any base method …], we have rewritten the Introduction section. In particular, we have added a statement to highlight that Lamda is an algorithm-agnostic framework (lines 121-123). Section 2.3 provides a conceptual description of how Lamda can be integrated into existing baseline HPO algorithms with minor adaptation.
> > > > - We apologize for the error in MUMBO and we have corrected it (see line 2216 and Algorithm 5).
> > > > - For the suggestion about [… move Algorithms 1 & 2 to the main paper …],  we have included two new pseudo-code (Algorithms 1 and 2 in Section 2).
> > > > - For the suggestions about [show that their method samples the high fidelity less;  show that other methods indeed suffer from sampling the high fidelity too often], we have added two experiments onFCNet-Naval-Propulsion and PD1-ImageNet. From the 2D visualization of the high-fidelity loss landscapes (see Figure 7), we can see that Lamda+BOHB has less high-fidelity samples. Without using Lamda, BOHB explores more on those less promising regions (in blue color). We see this as an inefficient use of limited computational budget. In contrast, Lamda+BOHB focus more on the promising regions (in red color).  These empirically support our assertion and also explains how Lamda helps the baseline HPO algorithm (BOHB as an example), under the same computational budget.
> > > >
> > > > >**Q4:** I appreciate the authors providing link to code. However, I think this needs to be done with the submission, not towards the end of the rebuttal period. I do not have sufficient time left to check the code submission. Also, it would increase my trust to have the full integration into all tools available.
> > > >
> > > > We fully understand the reviewer’s concern about the reproducibility. We tried our best to tidy up two demos amid the tight rebuttal period. Nevertheless, we promise to release the full source code including illustrative demos after the acceptance of this paper.

---

> > > > > ### Comment · Reviewer_gCCh · 2024-12-02
> > > > > **Rating Increase and updated response**
> > > > >
> > > > > * Thank you for improving the introduction, this really makes it easier to get into the paper. Also, using the analogy of a prior helps me to understand the paper better (might be due to my background, not sure if this applies to all reviewrs).
> > > > > * The figure on Page 1 might be very helpful, but I don't understand it yet. Why are there small boxes around the evaluations in panel (a)? And what is method (d), is it the proposed method?
> > > > > * BO in Section 3.2 does not have a reference for the exact implementation. What implementation do you use here?
> > > > > * Figure 10: even after zooming in, it is really hard to match the lines and legend. How about using different markers for the different methods (in addition to the different colors)?
> > > > > * Figure 38 looks like there is an error in the baseline DPL. Why would DPL perform so bad on this task, despite being a multi-fidelity method that is designed for tasks like finetuning models? Is it because the paper's setup only gives access to two fidelities? If this is the case, this strongly misrepresents the idea of DPL, which can work with quasi-continuous fidelities, and this should be strongly mentioned in the paper. After seeing this issue in Figure 38, I also see it in Figure 10.
> > > > > * The fact that no code is available yet is a substantial problem for me, and the promise to release code is not sufficient. For this reason, I will not consider this further in my evaluation.
> > > > >
> > > > > Overall, I think the paper has improved a lot, and my previous rating as reject is no longer justified. I increased it to borderline reject, mostly due to the things above, which prevent me from giving an even stronger rating.
> > > > >
> > > > > One last thing: please iterate the abstract a few more times, I think its readability can be further improved.

---

> > > > > > ### Author Response · Authors · 2024-12-02
> > > > > > **Follow-up response to the Reviewer gCCh**
> > > > > >
> > > > > > First of all, we thank the reviewer for the positive feedback and recognition on our revision. As for your additional questions, please find our point-by-point responses as follows.
> > > > > >
> > > > > > - **[Confusion on Figure 1]** We apologize for causing this confusion. The indexed small boxes in Figure 1(a) represent the locations of sampled points at each iteration, where the index represents the iteration number (see lines 53-54), e.g., $t_1$ indicates the first iteration and so on so forth. Figure 1(d) illustrates the progression of the sampling areas of the prior-based methods (see lines 66-70), where Lamda falls into this category. Note that as our **follow-up response to your Q1**, the priors used in Lamda are strategically learned from the HPO in the low-fidelity landscape.
> > > > > >
> > > > > > - **[Better layout in Figure 10]** We thank the reviewer for this suggestion and we like the idea of using different markers. We will revise all trajectory plots in the camera-ready version.
> > > > > >
> > > > > > - **[Poor performance of DPL in Figure 38]** We address the reviewer’s concerns from the following two aspects.
> > > > > >    - Firstly, instead of only using two fidelities, our experiments consider continuous fidelities. For example, the experiments in Figure 38 are about FCNet, where the lowest fidelity is epoch=1 while the highest fidelity is epoch=100.
> > > > > >    - We used the source code provided by the DPL paper [1] directly. We think the primary reason for DPL’s poor performance is its requirement to evaluate the quality of each new sample (see Equation (7) in [2]) across all fidelity levels to determine the most suitable one (see Equation (8) in [2]). This makes DPL inefficient and wastes significant resources on intermediate fidelity evaluations.
> > > > > >
> > > > > > - **[Improvement on abstract]** We thank the reviewer for this suggestion. We will improve the writing for the abstract in the camera-ready version.
> > > > > >
> > > > > > - **[Source code]** We apologize for not being able to provide the full code in our previous response, as the experimental code is messy. During this rebuttal period, we have been working hard to tidy it up. That’s why we provided the demo for Lamda+BOHB in our last response. Can we kindly ask for your patience as we complete this process, and we expect to have it ready **by the end of today** or **early tomorrow at the latest**.
> > > > > >
> > > > > > In summary, hope our responses can be helpful to clarify the reviewer's further concerns, and finally lead to an additional rating increase.
> > > > > >
> > > > > > ---
> > > > > > [1] https://github.com/machinelearningnuremberg/DPL
> > > > > >
> > > > > > [2] Kadra A, Janowski M, Wistuba M, et al. Scaling laws for hyperparameter optimization[J]. Advances in Neural Information Processing Systems, 2024, 36.

---

> > > > > > > ### Author Response · Authors · 2024-12-02
> > > > > > > **Update the source code**
> > > > > > >
> > > > > > > As promised, please kindly find our demo source from this link https://doi.org/10.5281/zenodo.14261996. If you need any further clarification, please feel free to let us know. Thank you very much again for engaging in this very interesting discussion.

---

> > > > > > > > ### Comment · Reviewer_gCCh · 2024-12-03
> > > > > > > > **Thank you**
> > > > > > > >
> > > > > > > > Thank you very much for uploading the source code. Unfortunately, I cannot check the source any more given the upcoming deadline. However, for the future, it would be good if you add comments and a readme to your code, but I understand that the time was tight now.

---

> > > > > > > > > ### Author Response · Authors · 2024-12-03
> > > > > > > > > **Response to issues in the evaluation**
> > > > > > > > >
> > > > > > > > > >**Q1:** However, this also requires so-called freeze-thaw capabilities: if one continues training a configuration, the previous state needs to be reloaded from disk. This allows DPL to make use of the full learning curve, and efficiently figure out which configurations are promising. Is this how you use DPL in your experiments?
> > > > > > > > >
> > > > > > > > > About the **"training process of DPL"**, we indeed use the **"reloaded previous state"** approach as described.  This method aligns with how the original DPL algorithm trains its learning curve surrogate model. Specifically, in the DPL code [1], within the file `power_law_surrogate.py`, line 222 to 247, the `_prepare_dataset` function shows that all historical solutions are reloaded and utilized to train the surrogate model effectively.
> > > > > > > > >
> > > > > > > > > [1] https://github.com/machinelearningnuremberg/DPL
> > > > > > > > >
> > > > > > > > > ```python
> > > > > > > > > def _prepare_dataset(self) -> TabularDataset:
> > > > > > > > >         """This method is called to prepare the necessary training dataset for training a model.
> > > > > > > > >
> > > > > > > > >         Returns:
> > > > > > > > >             train_dataset: A dataset consisting of examples, labels, budgets and learning curves.
> > > > > > > > >         """
> > > > > > > > >         train_examples, train_labels, train_budgets, train_curves = self.history_configurations()
> > > > > > > > >         train_curves = self.prepare_training_curves(train_budgets, train_curves)
> > > > > > > > >         train_examples = np.array(train_examples, dtype=np.single)
> > > > > > > > >         train_labels = np.array(train_labels, dtype=np.single)
> > > > > > > > >         train_budgets = np.array(train_budgets, dtype=np.single)
> > > > > > > > >
> > > > > > > > >         # scale budgets to [0, 1]
> > > > > > > > >         train_budgets = train_budgets / self.max_benchmark_epochs
> > > > > > > > >
> > > > > > > > >         train_dataset = TabularDataset(
> > > > > > > > >             train_examples,
> > > > > > > > >             train_labels,
> > > > > > > > >             train_budgets,
> > > > > > > > >             train_curves,
> > > > > > > > >         )
> > > > > > > > >         return train_dataset
> > > > > > > > >  ```
> > > > > > > > > > **Q2:** Also, if you are using continuous fidelities, how does LAMDA handle this? I thought LAMDA can only handle two fidelity levels, and needs to be extended to handle more than that?
> > > > > > > > >
> > > > > > > > > As our initial response to your **W1**, Lamda is an algorithm-agnostic framework that can be incorporated into any baseline HPO methods, no matter whether it considers multiple fidelity levels or not. Its first phase search uses the HPO within the low-fidelity landscape to learn a reasonably good prior for the second phase search. If the underlying baseline HPO algorithm uses continuous fidelities (e.g., BOCA, DPL), it just go ahead with this under the remaining computational budget.

---

> > > > > > > > > > ### Author Response · Authors · 2024-12-03
> > > > > > > > > > **Follow-up response to issues in the evaluation**
> > > > > > > > > >
> > > > > > > > > > We would like to further clarify the reviewer’s concern about using continuous fidelities in our experiments. In the Lamda’s first-phase search, we specify a fidelity level if the underlying problem has continuous fidelity levels. For example, the fidelity level is set to epoch=$4$ for FCNet in Figure 38. Please kindly refer to Table 5 of the Appendix C.4 which gives the fidelity level used in our experiments.

---

> > > > > > > ### Comment · Reviewer_gCCh · 2024-12-03
> > > > > > > **Issues in the evaluation**
> > > > > > >
> > > > > > > > [Poor performance of DPL in Figure 38] We address the reviewer’s concerns from the following two aspects.
> > > > > > > >
> > > > > > > > Firstly, instead of only using two fidelities, our experiments consider continuous fidelities. For example, the experiments in Figure 38 are about FCNet, where the lowest fidelity is epoch=1 while the highest fidelity is epoch=100.
> > > > > > > We used the source code provided by the DPL paper [1] directly. We think the primary reason for DPL’s poor performance is its requirement to evaluate the quality of each new sample (see Equation (7) in [2]) across all fidelity levels to determine the most suitable one (see Equation (8) in [2]). This makes DPL inefficient and wastes significant resources on intermediate fidelity evaluations.
> > > > > > >
> > > > > > > I think the understanding is correct, that DPL requires the evaluation of a new sample across all fidelity levels. However, this also requires so-called freeze-thaw capabilities: if one continues training a configuration, the previous state needs to be reloaded from disk. This allows DPL to make use of the full learning curve, and efficiently figure out which configurations are promising. Is this how you use DPL in your experiments?
> > > > > > >
> > > > > > > Also, if you are using continuous fidelities, how does LAMDA handle this? I thought LAMDA can only handle two fidelity levels, and needs to be extended to handle more than that?

---

### Official Review · Reviewer_EWdW · 2024-11-09

**Soundness:** 3
**Presentation:** 2
**Contribution:** 2
**Rating:** 5
**Confidence:** 4

**Summary:**

The paper introduces Lamda, a two-phase multi-fidelity hyperparameter optimization (HPO) framework designed to improve the efficiency of HPO by leveraging low-fidelity (LF) evaluations to identify promising regions in the search space. In the first phase, Lamda conducts a search in the LF landscape to locate regions where high-quality solutions are likely to exist. This is achieved using the Tree-structured Parzen Estimator (TPE) method to model the probability density functions (PDFs) of promising and inferior solutions, with the Overlapping Coefficient (OVL) used to measure the convergence of the promising region distribution.

In the second phase, the promising regions identified from the LF evaluations are transferred to guide the search in the high-fidelity (HF) landscape. This is done by modifying the sampling distribution to focus more on these promising regions, thereby reducing the need to explore the entire search space exhaustively at HF, which is computationally expensive.

**Strengths:**

Efficiency: Lamda reduces the computational cost of HPO by focusing HF evaluations on promising regions identified through LF evaluations, avoiding unnecessary exploration of less promising areas.

Versatility: The framework is versatile and can be integrated with a variety of existing HPO methods.
Empirical Validation: Extensive experiments on diverse benchmarks show that Lamda outperforms baseline methods, indicating its practical effectiveness across different domains and tasks.

**Weaknesses:**

Lack of Significant Novelty: The approach primarily combines existing techniques in a straightforward manner. The idea of using LF evaluations to guide HF searches is not entirely new in the field of multi-fidelity HPO.

Dependence on Overlapping Regions: The effectiveness of Lamda hinges on the assumption that promising regions in LF and HF landscapes overlap significantly.

**Questions:**

Overlap Assumption Validity:

Question: How does Lamda perform in scenarios where the promising regions of LF and HF landscapes do not significantly overlap? Have you tested the method on tasks where this assumption is invalid?

Parameter Sensitivity Analysis:

Question: Can you provide more insight into the sensitivity of Lamda's performance to its hyperparameters, such as the weight
𝑤 w and threshold 𝛾? Are there guidelines or adaptive strategies for selecting these parameters?

---

> ### Author Response · Authors · 2024-11-23
> **Responses to Reviewer EWdW**
>
> **W1: Lack of Significant Novelty**
>
> We respectfully think this is a misunderstanding of our core idea. We provide our justifications from the following two aspects.
>
> - In essence, Lamda is not a specific multi-fidelity hyperparameter optimization (HPO) algorithm. Instead, it is an algorithm-agnostic framework that plays as a booster (see line 254) for any existing HPO algorithms, no matter whether it considers multiple fidelities or not. As in lines 279-282, we provided five instances of incorporating Lamda into existing multi-fidelity HPO algorithms, vanilla Bayesian optimization (BO) and random search. Our experimental results demonstrated that using Lamda indeed enhance the baseline algorithms.
> - On the other hand, we admit that using prior to boost or jump-start HPO has been investigated in the literature (see Table 1). However, we argue that Lamda is unique in two aspects.
>     - First, the prior used in these methods is either crafted by expert experience or via transfer learning (see Sections B.3 and B.4 in Appendix). Differently, Lamda leverages the low-fidelity (LF) evaluations of the underlying HPO task to learn a reasonably reliable **prior**.
>     - Further, the current methods do not consider the cost incurred by obtaining such priors (e.g., experts’ cognitive burden or data collection budgets from other sources for transfer learning) in the underlying HPO task. In contrast, the computational cost of learning prior is accounted in when using Lamda to boost the baseline HPO algorithm (see Figures 2(c) and 2(f)).
>
> **W2: Dependence on Overlapping Regions**
>
> We respectfully think this is a misunderstanding. Our justifications are the following two aspects.
>
> - Our theoretical analysis of the convergence properties  (see Theorem 1 from line 296 and Theorem 2 from line 318) does not rely on the any assumption about the overlapping rate of promising regions within the LF and HF landscapes.
> - Our experiments have already considered a variety of scenarios with varying levels of overlaps. Please kindly refer to our reply to your **Q1**.
>
> **Q1: Overlap Assumption Validity**
>
> We thank the reviewer for this question. Our experiments have considered different scenarios where the promising regions of LF and HF landscape have varying levels of overlaps (Table 5 in Appendix C.4 provides statistics of the overlapping rates). In particular, we would like highlight the results on JAHS-CIFAR-10, JAHS-Colorectal-Histology, and JAHS-Fashion-MNIST, whose overlaps are low. Lamda consistently enhance the baseline algorithms (see Figures 17, 19, 21, and 23).
>
> **Q2: Parameter Sensitivity Analysis**
>
> We have conducted the parameter sensitivity study on $B$, $\gamma$, $\alpha$, $ \Delta$ and $w$ in our submitted manuscript (see Section 4.3). The experimental results demonstrated that Lamda is not sensitive to those parameter settings (see lines 491, 498, 502). Therefore, instead of introducing an adaptive parameter setting strategy, which often involves new hyperparameters (no-free-lunch), we think it might be more practical to use a constant parameter setting within our suggested range (see our discussion in Section 4.3).

---

> ### Author Response · Authors · 2024-11-29
> **Discussion deadline is approaching**
>
> Dear reviewer,
>
> First of all, thank you very much for your comments and constructive suggestions on our paper. They are all really helpful.
>
> Would you please kindly let us know whether our responses are satisfactory? Or if you have further concerns, we are more than to engage in further discussions.
>
> Thank you very much!
>
> Authors of paper #7012

---

> > ### Author Response · Authors · 2024-12-01
> > **We are keen on engaging in further discussions**
> >
> > Dear reviewer,
> >
> > Sorry for chasing this. However, since the discussion deadline is approaching, your kind help, comments, constructive suggestions on our work and our responses will be very much important and appreciated. We are keen on engaging in further discussions.
> >
> > Thank you very much!
> >
> > Authors of paper #7012

---

### Author Response · Authors · 2024-11-23
**General response**

Dear Reviewers and Meta-Reviewers,

We sincerely thank you for your thoughtful and constructive feedback. Your suggestions have been invaluable in refining our work, and we deeply appreciate the time and effort you dedicated to reviewing our paper. We have carefully addressed all points and incorporated the necessary improvements in the revised version.

We are pleased that reviewers appreciated:

- The **efficiency** and **versatility** of Lamda framework  across various HPO tasks and HPO techniques. [EWdW, gCCh, 6ipG]
- The **novelty** of Lamda framework for HPO. [RsFh]
- The **theoretical** **soundness** of Lamda for Bayesian optimization and Hyperband. [gCCh]
- The **strong empirical evidence and impact of the hyperparameters on performance is minor**, demonstrating Lamda outperforms baseline methods, indicating its practical effectiveness across different domains and tasks. [EWdW,gCCh]

We also sincerely thank the reviewers for their valuable suggestions, which helped us identify areas for improvement. In response to this feedback, we have made several major revisions and additions (marked as red in paper):

- **Clarified the novelty** of Lamda which learn a reasonably reliable prior by exploring the low-fidelity (LF) landscape of the underlying HPO task . *(Section 1)* [gCCh]
- **Enhanced explanations** of how Lamda is integrated into existing HPO methods. (pseudocode in Appendix G) [gCCh]
- Added **detailed explanation** to highlight the effectiveness of Lamda on different scenarios where the promising regions of LF and HF landscape have varying levels of overlaps . *(Appendix D and F.3)* [EWdW, gCCh,6ipG]
- Provided **additional experiments** and analysis of Lamda on hyperparameter optimization for fine-tuning pretrained image classification models, demonstrating its efficacy on larger models. *(Appendix F.4)* [gCCh]

We believe these revisions address the reviewers' concerns and further strengthen our work. Thank you for your constructive feedback and recognition of our contributions.

Best regards,

The Authors

---

### Author Response · Authors · 2024-11-25
**We are keen on engaging in further discussions**

Dear reviewers,

Sorry for chasing this. However, since the discussion deadline is approaching, your kind help, comments, constructive suggestions on our work and our responses will be very much important and appreciated. We are keen on engaging in further discussions.

Thank you very much!

Authors of paper #7012

---

### Meta-Review · Area_Chair_eQ3D · 2024-12-20

**Metareview:**

The paper considers the problem of multi-fidelity hyperparameter optimization and presents a two-phase framework titled LAMDA. The key idea is to first identify promising regions in the low-fidelity (LF) landscape, and then use these regions to guide the search in the high-fidelity (HF) landscape. Similar ideas have already been proposed in the literature (for e.g., BOCA) and the proposed contribution is slightly below par for ICLR. The motivation of the paper describing challenges with existing approaches for multi-fidelity HPO is unclear and not justified properly. Similarly, the point about natural integration with other approach appears ad-hoc. I request the authors to update the paper based on comments from the reviewers for the next cycle.

**Additional Comments On Reviewer Discussion:**

Multiple reviewers (EWdW, gCCh) had concerns about the novelty of the approach. This is not the main reason to reject a paper but it is useful to specify what new challenges are being addressed in the paper. Reviewers also had concerns about calling the approach as a "framework" when the integration with other approaches like BOHB wasn't clear. Most reviewers engaged with the author response and score the paper to be below par.

---

### Decision · Program_Chairs · 2025-01-22

Reject